# The spiked matrix model with generative priors

**Benjamin Aubin**[†], **Bruno Loureiro**[†], **Antoine Maillard**[⋆],
**Florent Krzakala**[⋆], **Lenka Zdeborová**[†]

## Abstract

Using a low-dimensional parametrization of signals is a generic and powerful way to enhance performance in signal processing and statistical inference. A very popular and widely explored type of dimensionality reduction is sparsity; another type is generative modelling of signal distributions. Generative models based on neural networks, such as GANs or variational auto-encoders, are particularly performant and are gaining on applicability. In this paper we study spiked matrix models, where a low-rank matrix is observed through a noisy channel. This problem with sparse structure of the spikes has attracted broad attention in the past literature. Here, we replace the sparsity assumption by generative modelling, and investigate the consequences on statistical and algorithmic properties. We analyze the Bayes-optimal performance under specific generative models for the spike. In contrast with the sparsity assumption, we do not observe regions of parameters where statistical performance is superior to the best known algorithmic performance. We show that in the analyzed cases the approximate message passing algorithm is able to reach optimal performance. We also design enhanced spectral algorithms and analyze their performance and thresholds using random matrix theory, showing their superiority to the classical principal component analysis. We complement our theoretical results by illustrating the performance of the spectral algorithms when the spikes come from real datasets.

## 1 Introduction

A key idea of modern signal processing is to exploit the structure of the signals under investigation. A traditional and powerful way of doing so is via sparse representations of the signals. Images are typically sparse in the wavelet domain, sound in the Fourier domain, and sparse coding [1] is designed to search automatically for dictionaries in which the signal is sparse. This compressed representation of the signal can be used to enable efficient signal processing under larger noise or with fewer samples leading to the ideas behind compressed sensing [2] or sparsity enhancing regularizations. Recent years brought a surge of interest in another powerful and generic way of representing signals – generative modeling. In particular the generative adversarial networks (GANs) [3] provide an impressively powerful way to represent classes of signals. A recent series of works on compressed sensing and other regression-related problems successfully explored the idea of replacing the traditionally used sparsity by generative models [4–10]. These results and performances conceivably suggest that [11]:

Generative models are the new sparsity.

Next to compressed sensing and regression, another technique in statistical analysis that uses sparsity in a fruitful way is sparse principal component analysis (PCA) [12]. Compared to the standard PCA,

⋆ Laboratoire de Physique de l'École Normale Supérieure, PSL University & CNRS & Sorbonne Universités, Paris, France.

in sparse-PCA the principal components are linear combinations of a few of the input variables, specifically $k$ of them. This means (for rank-one) that we aim to decompose the observed data matrix $Y \in \mathbb{R}^{n \times p}$ as $Y = \mathbf{u}\mathbf{v}^\mathsf{T} + \xi$ where the spike $\mathbf{v} \in \mathbb{R}^p$ is a vector with only $k \ll p$ non-zero components, and $\mathbf{u}, \xi$ are commonly modelled as independent and identically distributed (i.i.d.) Gaussian variables.

The main goal of this paper is to explore the idea of replacing sparsity of the spike $\mathbf{v}$ by the assumption that the spike belongs to the range of a generative model. Sparse-PCA with structured sparsity inducing priors is well studied, e.g. [13], in this paper we remove the sparsity entirely and in a sense replace it by lower dimensionality of the latent space of the generative model. For the purpose of comparing generative model priors and sparsity we focus on the rich range of properties in the noisy high-dimensional regime (denoted below, borrowing statistical physics jargon, as the *thermodynamic limit*) where the spike $\mathbf{v}$ cannot be estimated consistently, but can be estimated better than by random guessing. In particular we analyze two spiked-matrix models as considered in a series of existing works on sparse-PCA, e.g. [14–20], defined as follows:

**Spiked Wigner model ($\mathbf{vv}^\mathsf{T}$):** Consider an unknown vector (the spike) $\mathbf{v}^\star \in \mathbb{R}^p$ drawn from a distribution $P_v$; we observe a matrix $Y \in \mathbb{R}^{p \times p}$ with a symmetric noise term $\xi \in \mathbb{R}^{p \times p}$ and $\Delta > 0$:

$$Y = \frac{1}{\sqrt{p}}\mathbf{v}^\star\mathbf{v}^{\star\mathsf{T}} + \sqrt{\Delta}\xi\,, \tag{1}$$

where $\xi_{ij} \sim \mathcal{N}(0,1)$ i.i.d. The aim is to find back the hidden spike $\mathbf{v}^\star$ from $Y$ (up to a global sign).

**Spiked Wishart (or spiked covariance) model ($\mathbf{uv}^\mathsf{T}$):** Consider two unknown vectors $\mathbf{u}^\star \in \mathbb{R}^n$ and $\mathbf{v}^\star \in \mathbb{R}^p$ drawn from distributions $P_u$ and $P_v$ and let $\xi \in \mathbb{R}^{n \times p}$ with $\xi_{\mu i} \sim \mathcal{N}(0,1)$ i.i.d. and $\Delta > 0$, we observe

$$Y = \frac{1}{\sqrt{p}}\mathbf{u}^\star\mathbf{v}^{\star\mathsf{T}} + \sqrt{\Delta}\xi\,; \tag{2}$$

the goal is to find back the hidden spikes $\mathbf{u}^\star$ and $\mathbf{v}^\star$ from $Y \in \mathbb{R}^{n \times p}$.

The noisy high-dimensional limit that we consider in this paper (the *thermodynamic limit*) is $p, n \to \infty$ while $\beta \equiv n/p = \Theta(1)$, and the noise $\xi$ has a variance $\Delta = \Theta(1)$. The prior $P_v$ is representing the spike $\mathbf{v}$ via a $k$-dimensional parametrization with $\alpha \equiv p/k = \Theta(1)$. In the sparse case, $k$ is the number of non-zeros components of $\mathbf{v}^\star$, while in generative models $k$ is the number of latent variables.

## 1.1 Considered generative models

The simplest non-separable prior $P_v$ that we consider is the Gaussian model with a covariance matrix $\Sigma$, that is $P_v(\mathbf{v}) = \mathcal{N}(\mathbf{v}; \mathbf{0}, \Sigma)$. This prior is not compressive, yet it captures some structure and can be simply estimated from data via the empirical covariance. We use this prior later to produce Fig. 4.

To exploit the practically observed power of generative models, it would be desirable to consider models (e.g. GANs, variational auto-encoders, restricted Boltzmann machines, or others) trained on datasets of examples of possible spikes. Such training, however, leads to correlations between the weights of the underlying neural networks for which the theoretical part of the present paper does not apply readily. To keep tractability in a closed form, and subsequent theoretical insights, we focus on multi-layer generative models where all the weight matrices $W^{(l)} \in \mathbb{R}^{k_{l+1} \times k_l}$, $l = 1, \ldots, L$ (with $k_1 = k$, $k_{L+1} = p$), are fixed, layer-wise independent, i.i.d. Gaussian with zero mean and unit variance. Let $\mathbf{v} \in \mathbb{R}^p$ be the output of such a generative model

$$\mathbf{v} = \varphi^{(L)}\left(\frac{1}{\sqrt{k_L}}W^{(L)}\ldots\varphi^{(1)}\left(\frac{1}{\sqrt{k_1}}W^{(1)}\mathbf{z}\right)\ldots\right)\,. \tag{3}$$

with $\mathbf{z} \in \mathbb{R}^k$ a latent variable drawn from separable distribution $P_z$, with $\rho_z = \mathbb{E}_{P_z}[z^2]$ and $\varphi^{(l)}$ element-wise activation functions that can be either deterministic or stochastic. In the setting considered in this paper the ground-truth spike $\mathbf{v}^\star$ is generated using a ground-truth value of the latent variable $\mathbf{z}^\star$. The spike is then estimated from the knowledge of the data matrix $Y$, and the known form of the spiked-matrix and of the generative model. In particular the matrices $W^{(l)}$ are known, as are the parameters $\beta, \Delta, P_z, P_u, P_v, \varphi^{(l)}$. Only the spikes $\mathbf{v}^\star, \mathbf{u}^\star$ and the latent vector $\mathbf{z}^\star$ are unknown, and are to be inferred.

For concreteness and simplicity, the generative model that will be analyzed in most examples given in the present paper is the single-layer case of (3) with $L = 1$:

$$\mathbf{v} = \varphi\left(\frac{1}{\sqrt{k}}W\mathbf{z}\right) \quad \Leftrightarrow \quad \mathbf{v} \sim P_{\text{out}}\left(\cdot \Big| \frac{1}{\sqrt{k}}W\mathbf{z}\right). \tag{4}$$

We define the compression ratio $\alpha \equiv p/k$. In what follows we will illustrate our results for $\varphi$ being linear, sign and ReLU functions.

## 1.2 Summary of main contributions

We analyze how the availability of generative priors, defined in section 1.1, influences the statistical and algorithmic properties of the spiked-matrix models (1) and (2). Both sparse-PCA and generative priors provide statistical advantages when the effective dimensionality $k$ is small, $k \ll p$. However, we show that from the algorithmic perspective the two cases are quite different. This is why our main findings are best presented in a context of the results known for sparse-PCA. We draw two main conclusions from the present work:

**(i) No algorithmic gap with generative-model priors:** Sharp and detailed results are known in the thermodynamic limit (as defined above) when the spike $\mathbf{v}^\star$ is sampled from a separable distribution $P_v$. A detailed account of several examples can be found in [21]. The main finding for sparse priors $P_v$ is that when the sparsity $\rho = k/p = 1/\alpha$ is large enough then there exist optimal algorithms [15], while for $\rho$ small enough there is a striking gap between statistically optimal performance and the one of best known algorithms [16]. The small-$\rho$ expansion studied in [21] is consistent with the well-known results for exact recovery of the support of $\mathbf{v}^\star$ [22,23], which is one of the best-known cases in which gaps between statistical and best-known algorithmic performance were described.

Our analysis of the spiked-matrix models with generative priors reveals that in the investigated cases the algorithmic gap disappears and known algorithms are able to obtain (asymptotically) optimal performance even when the dimension is greatly reduced, i.e. $\alpha \gg 1$. Analogous conclusion about the lack of algorithmic gaps was reached for the problem of phase retrieval under a deep generative prior in [9]. This result suggests that plausibly generative priors are better than sparsity as they lead to algorithmically easier problems and give back the hope that the structure can be exploited not only information-theoretically but also tractably.

**(ii) Spectral algorithms reaching statistical threshold:** Arguably the most basic algorithm used to solve the spiked-matrix model is based on the leading singular vectors of the matrix $Y$. We will refer to this as PCA. Previous work on spiked-matrix models [17,21] established that in the thermodynamic limit and for separable priors of zero mean PCA reaches the best performance of all known efficient algorithms in terms of the value of noise $\Delta$ below which it is able to provide positive correlation between its estimator and the ground-truth spike. While for sparse priors positive correlation is statistically reachable even for larger values of $\Delta$ [17,21], no efficient algorithm beating the PCA threshold is known[2].

In the case of generative priors we find in this paper that other spectral methods improve on the canonical PCA. We design a spectral method, called LAMP, that (under certain assumptions, e.g. zero mean of the spikes) reach the statistically optimal threshold, meaning that for larger values of noise variance no other (even exponential) algorithm is able to reach positive correlation with the spike. Again this is a striking difference with the sparse separable prior, making the generative priors algorithmically more attractive. We demonstrate the performance of LAMP on the spiked-matrix model when the spike is taken to be one of the fashion-MNIST images showing considerable improvement over canonical PCA.

## 2 Analysis of information-theoretically optimal estimation

We first discuss the information theoretic results on the estimation of the spike, regardless of the computational cost. A considerable amount of results have been obtained for the spiked-matrix models with separable priors [14,15,18,19,25–29]. Here, we extend these results to the case where the spike $\mathbf{v}^\star \in \mathbb{R}^p$ is generated from a *generic non-separable prior* $P_v$ on $\mathbb{R}^p$.

## 2.1 Mutual Information and Minimal Mean Squared Error

We consider the mutual information between the ground-truth spike $\mathbf{v}^\star$ and the observation $Y$, defined as $I(Y; \mathbf{v}^\star) = D_{\mathrm{KL}}(P_{(v^\star, Y)} \| P_{v^\star} P_Y)$. Next, we consider the best possible value of the mean-squared-error on recovering the spike, commonly called the minimum mean-squared-error (MMSE). The MMSE estimator is computed from marginal-means of the posterior distribution $P(\mathbf{v}|Y)$.

**Theorem 1.** *[Mutual information for the spiked Wigner model with structured spike] Informally (see SM section 3 for details and proof), assume the spikes $\mathbf{v}^\star$ come from a sequence (of growing dimension p) of generic structured priors $P_v$ on $\mathbb{R}^p$, then*

$$\lim_{p \to \infty} i_p \equiv \lim_{p \to \infty} \frac{I(Y; \mathbf{v}^\star)}{p} = \inf_{\rho_v \geq q_v \geq 0} i_{\mathrm{RS}}(\Delta, q_v), \tag{5}$$

$$\text{with} \quad i_{\mathrm{RS}}(\Delta, q_v) \equiv \frac{(\rho_v - q_v)^2}{4\Delta} + \lim_{p \to \infty} \frac{I\left(\mathbf{v}; \mathbf{v} + \sqrt{\frac{\Delta}{q_v}}\boldsymbol{\xi}\right)}{p} \tag{6}$$

*and $\boldsymbol{\xi}$ being a Gaussian vector with zero mean, unit diagonal variance and $\rho_v = \lim\limits_{p \to \infty} \mathbb{E}_{P_v}[\mathbf{v}^\intercal \mathbf{v}]/p$.*

This theorem connects the asymptotic mutual information of the spiked model with generative prior $P_v$ to the mutual information between $\mathbf{v}$ taken from $P_v$ and its noisy version, $I(\mathbf{v}; \mathbf{v} + \sqrt{\Delta/q_v}\boldsymbol{\xi})$. Computing this later mutual information is itself a high-dimensional task, hard in full generality, but it can be done for a range of models. The simplest tractable case is when the prior $P_v$ is separable, then it yields back exactly the formula known from [18, 19, 26]. It can be computed also for the Gaussian generative model, $P_v(\mathbf{v}) = \mathcal{N}(\mathbf{v}; \mathbf{0}, \Sigma)$, leading to $I(\mathbf{v}; \mathbf{v} + \sqrt{\Delta/q_v}\boldsymbol{\xi}) = \mathrm{Tr}\left(\log\left(\mathrm{I}_p + q_v \Sigma/\Delta\right)\right)/2$.

More interestingly, the mutual information associated to the generative prior in eq. (6) can also be asymptotically computed for the multi-layer generative model with random weights, defined in eq. (3). Indeed, for the single-layer prior (4) the corresponding formula for mutual information has been derived and proven in [30]. For the multi-layer case the mutual information formula has been derived in [6] and proven for the case of two layers in [31]. Theorem 1 together with the results from [6, 30, 31] yields the following formula (see SM sec. 3 for details) for the spiked Wigner model (1) with $L$-layer generative prior (3):

$$i_{\mathrm{RS}}(\Delta, q_v) = \frac{\rho_v^2}{4\Delta} + \frac{1}{4\Delta}q_v^2 + \tag{7}$$

$$\frac{1}{\alpha} \underset{\{\hat{q}_l, q_l\}_l}{\mathbf{extr}} \left[ \frac{1}{2} \sum_{l=1}^{L} \alpha_l \hat{q}_l q_l - \sum_{l=2}^{L} \alpha_l \Psi_{\mathrm{out}}^{(l)}(\hat{q}_l, q_{l-1}) - \alpha \Psi_{\mathrm{out}}^{(L+1)}\left(\frac{q_v}{\Delta}, q_L\right) - \Psi_z(\hat{q}_z) \right].$$

where $\alpha_l = k_l/k$ (note that in particular $\alpha_1 = 1$) and the functions $\Psi_z$, $\Psi_{\mathrm{out}}$ are defined by

$$\Psi_z(x) \equiv \mathbb{E}_\xi \left[ \mathcal{Z}_z\left(x^{1/2}\xi, x\right) \log\left(\mathcal{Z}_z\left(x^{1/2}\xi, x\right)\right) \right], \tag{8}$$

$$\Psi_{\mathrm{out}}^{(l)}(x, y) \equiv \mathbb{E}_{\xi, \eta} \left[ \mathcal{Z}_{\mathrm{out}}^{(l)}\left(x^{1/2}\xi, x, y^{1/2}\eta, \rho_l - y\right) \log\left(\mathcal{Z}_{\mathrm{out}}^{(l)}\left(x^{1/2}\xi, x, y^{1/2}\eta, \rho_l - y\right)\right) \right], \tag{9}$$

with $\xi, \eta \sim \mathcal{N}(0, 1)$ i.i.d., $\rho_{l+1}$ the second moment of the hidden variable $\mathbf{h}^{(l+1)} = \varphi^{(l)}\left(\frac{1}{\sqrt{k_l}} W^{(l)} \mathbf{h}^{(l)}\right) \in \mathbb{R}^{k_{l+1}}$ and $\mathcal{Z}_z, \mathcal{Z}_{\mathrm{out}}^{(l)}$ are the normalizations of the following denoising scalar distributions:

$$Q_z^{\gamma, \Lambda}(z) \equiv \frac{P_z(z)}{\mathcal{Z}_z(\gamma, \Lambda)} e^{-\frac{\Lambda}{2}z^2 + \gamma z}, \quad Q_{\mathrm{out}}^{(l), B, A, \omega, V}(v, x) \equiv \frac{P_{\mathrm{out}}^{(l)}(v|x)}{\mathcal{Z}_{\mathrm{out}}^{(l)}(B, A, \omega, V)} e^{-\frac{A}{2}v^2 + Bv} \frac{e^{-\frac{(x-\omega)^2}{2V}}}{\sqrt{2\pi V}}. \tag{10}$$

Result (7) is remarkable in that it connects the asymptotic mutual information of a high-dimensional model with a simple scalar formula that can be easily evaluated. In the SM sec. 2 we show how this formula is obtained using the heuristic replica method from statistical physics and, once we have the formula in hand, we prove it using the interpolation method in SM sec. 3. In SM sec. 2.2 we also give the corresponding formula for the spiked Wishart model.

Beyond its theoretical interest, the main point of the mutual information formula is that it yields the optimal value of the mean-squared error (MMSE). It is well-known [32] that the mean-squared error is minimized by an estimator evaluating the conditional expectation of the signal given the observations. Following generic theorems on the connection between the mutual information and the MMSE [33], one can prove in particular that for the spiked-matrix model [27] the MMSE on the spike $\mathbf{v}^\star$ is asymptotically given by:

$$\text{MMSE}_v = \rho_v - q_v^\star, \tag{11}$$

where $q_v^\star$ is the optimizer of the function $i_{\text{RS}}(\Delta, q_v)$.

## 2.2 Examples of phase diagrams

Taking the extremization over $q_v, \hat{q}_z, q_z$ in eq. (7), we obtain the following fixed point equations:

$$q_v = 2\partial_{q_v}\Psi_{\text{out}}\left(\frac{q_v}{\Delta}, q_z\right), \quad q_z = 2\partial_{\hat{q}_z}\Psi_z(\hat{q}_z), \quad \hat{q}_z = 2\alpha\partial_{q_z}\Psi_{\text{out}}\left(\frac{q_v}{\Delta}, q_z\right). \tag{12}$$

Using (11), analyzing the fixed points of eqs. (12) provides all the informations about the performance of the Bayes-optimal estimator in the models under consideration.

**Phase transition:** A first question is whether better estimation than random guessing from the prior is possible. In terms of fixed points of eqs. (12), this corresponds to the existence of the *non-informative* fixed point $q_v^\star = 0$ (i.e. zero overlap with the spike, or maximum $\text{MSE}_v = \rho_v$). Evaluating the right-hand side of eqs. (12) at $q_v = 0$, we can see that $q_v^\star = 0$ is a fixed point if

$$\mathbb{E}_{P_z}[z] = 0 \quad \text{and} \quad \mathbb{E}_{Q_{\text{out}}^0}[v] = 0, \tag{13}$$

where $Q_{\text{out}}^0(v,x) \equiv Q_{\text{out}}^{0,0,0,\rho_z}(v,x)$ from eq. (10). Note that for a deterministic channel the second condition is equivalent to $\varphi$ being an odd function.

When the condition (13) holds, $(q_v, \hat{q}_z, q_z) = (0,0,0)$ is a fixed point of eq. (12). The numerical stability of this fixed point determines a phase transition point $\Delta_c$, defined as the noise below which the fixed point $(0,0,0)$ becomes unstable. This corresponds to the value of $\Delta$ for which the largest eigenvalue of the Jacobian of the eqs. (12) at $(0,0,0)$, given by

$$2\mathrm{d}(\partial_{q_v}\Psi_{\text{out}}, \alpha\partial_{q_z}\Psi_{\text{out}}, \partial_{\hat{q}_z}\Psi_z)|_{(0,0,0)} = \begin{pmatrix} \frac{1}{\Delta}\left(\mathbb{E}_{Q_{\text{out}}^0}v^2\right)^2 & 0 & \frac{1}{\rho_z^2}\left(\mathbb{E}_{Q_{\text{out}}^0}vx\right)^2 \\ \frac{\alpha}{\Delta}\left(\mathbb{E}_{Q_{\text{out}}^0}vx\right)^2 & 0 & \frac{\alpha}{\rho_z^2}\left(\mathbb{E}_{Q_{\text{out}}^0}x^2 - \rho_z\right)^2 \\ 0 & \left(\mathbb{E}_{P_z}z^2\right)^2 & 0 \end{pmatrix}, \tag{14}$$

becomes greater than one. The details of this calculation can be found in sec. 6 of the SM.

It is instructive to compute $\Delta_c$ in specific cases. We therefore fix $P_z = \mathcal{N}(0,1)$ and $P_{\text{out}}(v|x) = \delta(v - \varphi(x))$ and discuss two different choices of (odd) activation function $\varphi$.

**Linear activation:** For $\varphi(x) = x$ the leading eigenvalue of the Jacobian becomes one at $\Delta_c = 1 + \alpha$. Note that for $L > 1$ the result is derived in SM sec. 2.3 and reads $\Delta_c = 1 + \sum_{l=1}^{L}\frac{\alpha}{\alpha_l}$. Note that in the limit $\alpha = 0$ we recover the phase transition $\Delta_c = 1$ known from the case with separable prior [21]. For $\alpha > 0$, we have $\Delta_c > 1$ meaning the spike can be estimated more efficiently when its structure is accounted for.

**Sign activation:** For $\varphi(x) = \text{sgn}(x)$ the leading eigenvalue of the Jacobian becomes one at $\Delta_c = 1 + \frac{4\alpha}{\pi^2}$. As above it generalizes for $L > 1$ as $\Delta_c = 1 + \sum_{l=1}^{L}\left(\frac{4}{\pi^2}\right)^l\frac{\alpha}{\alpha_l}$. For $\alpha = 0$, $P_v = \text{Bern}(1/2)$, and the transition $\Delta_c = 1$ agrees with the one found for a separable prior distribution [21]. As in the linear case, for $\alpha > 0$, we can estimate the spike for larger values of noise than in the separable case.

In Fig. 1 we solve the fixed point equations (12) and plot the MMSE obtained from the fixed point in a heat map, for the linear, sign and relu activations. The white dashed line marks the above stated threshold $\Delta_c$. The property that we find the most striking is that in these three evaluated cases, for all values of $\Delta$, $\alpha$ and $L$ that we analyzed, we always found that eq. (12) has a unique stable fixed point.

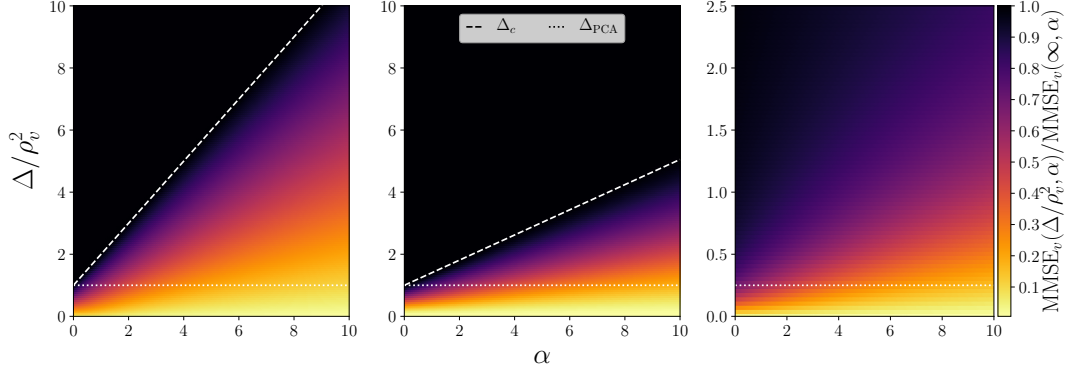

Figure 1: Spiked Wigner model $\mathrm{MMSE}_v$ on the spike as a function of noise to signal ratio $\Delta/\rho_v^2$, and generative prior (4) with compression ratio $\alpha$ for $L = 1$ linear (left, $\rho_v = 1$), sign (center, $\rho_v = 1$), and relu (right, $\rho_v = 1/2$) activations. Dashed white lines mark the phase transitions $\Delta_c$, matched by both the AMP and LAMP algorithms. Dotted white line marks the phase transition of canonical PCA.

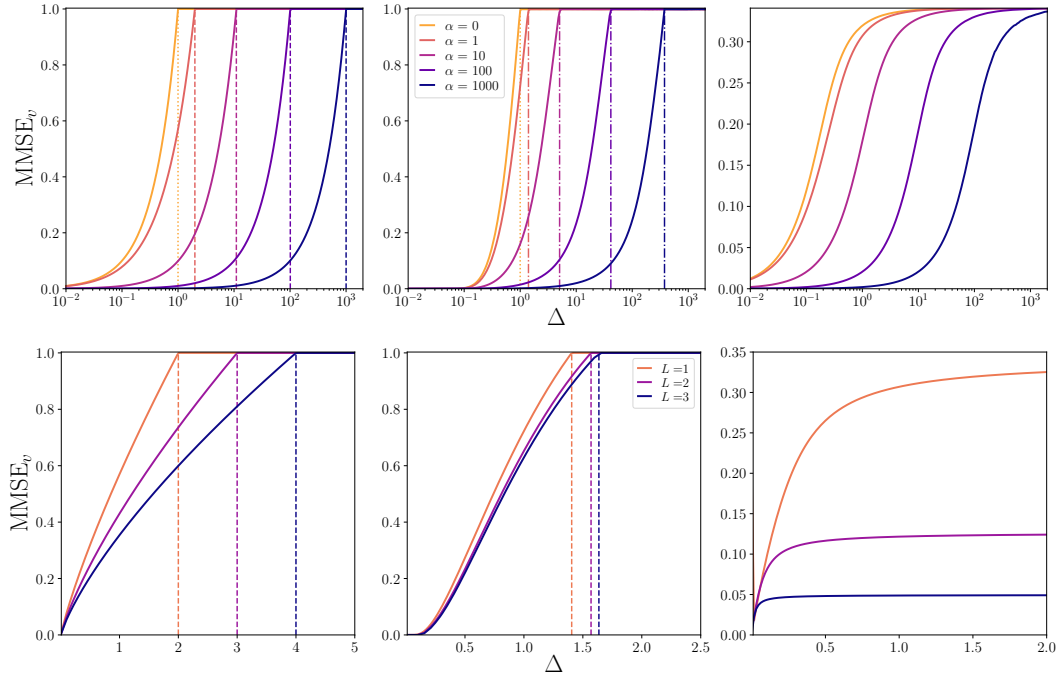

Figure 2: Spiked Wigner model: $\mathrm{MMSE}_v$ as a function of noise $\Delta$ - (**upper**) for a wide range of compression ratios $\alpha = 0, 1, 10, 100, 1000$, for $L = 1$ linear (left), sign (center), and relu (right) activations. Unique stable fixed point of (12) is found for all these cases - (**lower**) for different depths $L = 1, 2, 3$ with constant compressive ratio $\alpha_1 = \alpha_2 = \alpha_3 = 1$, for linear (left), sign (center), and relu (right) activations. The second moment of the variable $v$ for $L = 1, 2, 3$ are $\rho_v^{(L)} = 1$ for linear and sign, while for ReLU $\rho_v^{(L)} = 1/2^L$. Similarly a unique stable fixed point is found in these cases.

Thus we have not identified any first order phase transition (in the physics terminology). This is illustrated in Fig. 2 for larger values of $\alpha$ (**upper**) and for different depths $L$ (**lower**), where we solved the eq. (12) iteratively from uncorrelated initial condition, and from initial condition corresponding to the ground truth signal, and found that both lead to the same fixed point. In particular, as a unique fixed point is found, the Bayes optimal errors are continuous and we did not observe any algorithmic gap. Details of the expressions equivalent to eq. (12-14) for $L \geq 1$ are detailed in SM sec. 2.3.

## 3 Approximate message passing with generative priors

A straightforward algorithmic evaluation of the Bayes-optimal estimator is exponentially costly. This section is devoted to the analysis of an approximate message passing (AMP) algorithm that for the analyzed cases is able to reach the optimal performance (in the thermodynamic limit). For the purpose of presentation, we focus again on the spiked Wigner model (see SM for the spiked Wishart model). For separable priors, the AMP for the spiked Wigner model is well known [14–16]. It can, however, be extended to non-separable priors [6, 34, 35]. We show in SM sec. 4 how AMP can be generalized to handle the generative model (4). Iterating this derivation leads naturally to its multi-layer version ML-AMP for $L \geq 1$. In particular AMP for $L = 1$ reads:

---

**Input:** $Y \in \mathbb{R}^{p \times p}$ and $W \in \mathbb{R}^{p \times k}$:
*Initialize to zero:* $(\mathbf{g}, \hat{\mathbf{v}}, \mathbf{B}_v, A_v)^{t=0}$.
*Initialize with:* $\hat{\mathbf{v}}^{t=1} = \mathcal{N}(0, \sigma^2)$, $\hat{\mathbf{z}}^{t=1} = \mathcal{N}(0, \sigma^2)$, and $\hat{\mathbf{c}}_v^{t=1} = \mathbb{1}_p$, $\hat{\mathbf{c}}_z^{t=1} = \mathbb{1}_k$, $t = 1$.
**repeat**
  *Spiked layer:*
  $\mathbf{B}_v^t = \frac{1}{\Delta} \frac{Y}{\sqrt{p}} \hat{\mathbf{v}}^t - \frac{1}{\Delta} \frac{\left(\mathbb{1}_p^\intercal \hat{\mathbf{c}}_v^t\right)}{p} \hat{\mathbf{v}}^{t-1}$   and   $A_v^t = \frac{1}{\Delta p} \|\hat{\mathbf{v}}^t\|_2^2 \mathrm{I}_p$.
  *Generative layer:*
  $V^t = \frac{1}{k} \left(\mathbb{1}_k^\intercal \hat{\mathbf{c}}_z^t\right) \mathrm{I}_p$,   $\boldsymbol{\omega}^t = \frac{1}{\sqrt{k}} W \hat{\mathbf{z}}^t - V^t \mathbf{g}^{t-1}$   and   $\mathbf{g}^t = f_{\mathrm{out}}\left(\mathbf{B}_v^t, A_v^t, \boldsymbol{\omega}^t, V^t\right)$,
  $\Lambda^t = \frac{1}{k} \|\mathbf{g}^t\|_2^2 \mathrm{I}_k$   and   $\boldsymbol{\gamma}^t = \frac{1}{\sqrt{k}} W^\intercal \mathbf{g}^t + \Lambda^t \hat{\mathbf{z}}^t$.
  *Update of the estimated marginals:*
  $\hat{\mathbf{v}}^{t+1} = f_v(\mathbf{B}_v^t, A_v^t, \boldsymbol{\omega}^t, V^t)$      and      $\hat{\mathbf{c}}_v^{t+1} = \partial_B f_v(\mathbf{B}_v^t, A_v^t, \boldsymbol{\omega}^t, V^t)$,
  $\hat{\mathbf{z}}^{t+1} = f_z(\boldsymbol{\gamma}^t, \Lambda^t)$      and      $\hat{\mathbf{c}}_z^{t+1} = \partial_\gamma f_z(\boldsymbol{\gamma}^t, \Lambda^t)$,
  $t = t + 1$.
**until** Convergence.
**Output:** $\hat{\mathbf{v}}, \hat{\mathbf{z}}$.

**Algorithm 1:** AMP algorithm for the spiked Wigner model with single-layer generative prior.

---

where $\mathrm{I}_s$ and $\mathbb{1}_s$ denote respectively the identity matrix and vector of ones of size $s$. The update functions $f_{\mathrm{out}}$ and $f_v$ are the means of $V^{-1}(x - \omega)$ and $v$ with respect to $Q_{\mathrm{out}}$, eq. (10), while the update function $f_z$ is the mean of $z$ with respect to $Q_z$, eq. (10).

The algorithm for the spiked Wishart model is very similar and both derivations are given in SM sec. 4. We define the overlap of the AMP estimator with the ground truth spike as $(\hat{\mathbf{v}}^t)^\intercal \mathbf{v}^\star / p \longrightarrow q_v^t$ as $p \to \infty$. Perhaps the most important virtue of AMP-type algorithms is that their asymptotic performance can be tracked exactly via a set of scalar equations called *state evolution*. This fact has been proven for a range of models including the spiked matrix models with separable priors in [36], and with non-separable priors in [35]. To help the reader understand the state evolution equations we provide a heuristic derivation in the SM, section 4.4. For $L = 1$, the state evolution states that the overlap $q_v^t$ evolves under iterations of the AMP algorithm as:

$$q_v^{t+1} = 2 \partial_{q_v} \Psi_{\mathrm{out}}\left(\frac{q_v^t}{\Delta}, q_z^t\right), \quad q_z^{t+1} = 2 \partial_{\hat{q}_z} \Psi_z\left(\hat{q}_z^t\right), \quad \hat{q}_z^t = 2\alpha \partial_{q_z} \Psi_{\mathrm{out}}\left(\frac{q_v^t}{\Delta}, q_z^t\right), \quad (15)$$

with initialization $q_v^{t=0} = \varepsilon$, $q_z^{t=0} = \varepsilon$ and a small $\varepsilon > 0$. We notice immediately that (15) are the same equations as the fixed point equations related to the Bayes-optimal estimation (12) with specific time-indices and initialization, but crucially the same fixed points. This observation generalizes naturally to $L > 1$. Thus the analysis of fixed points in sec. 2.2 applies also to the behaviour of AMP. In particular in all the scenarios for which we solved the corresponding equations numerically we found the stable fixed point of (12) to be unique or equivalently the Bayes optimal errors as a function of the noise to be continuous. Hence under the assumption that the data was created using the model from eq. (1) and the spike from eq. (3) with i.i.d weight matrices $W^{(l)}$ and i.i.d. Gaussian entries, it means the AMP algorithm is able to reach asymptotically the optimal performance in all these cases. This is further illustrated in Fig. 3 where we explicitly compare runs of AMP on finite size instances with the results of the asymptotic state evolution, thus also giving an idea of the amplitude of the finite size effects. Note that we provide a demonstration notebook in [37] that compares AMP, LAMP and PCA numerical performances. Finally as has been done in previous works, e.g. [5, 8–10] for compressed sensing and denoising, translating our results to practical situations in designing an AMP algorithm that takes care of correlated GAN or VAE weights is still under investigation.

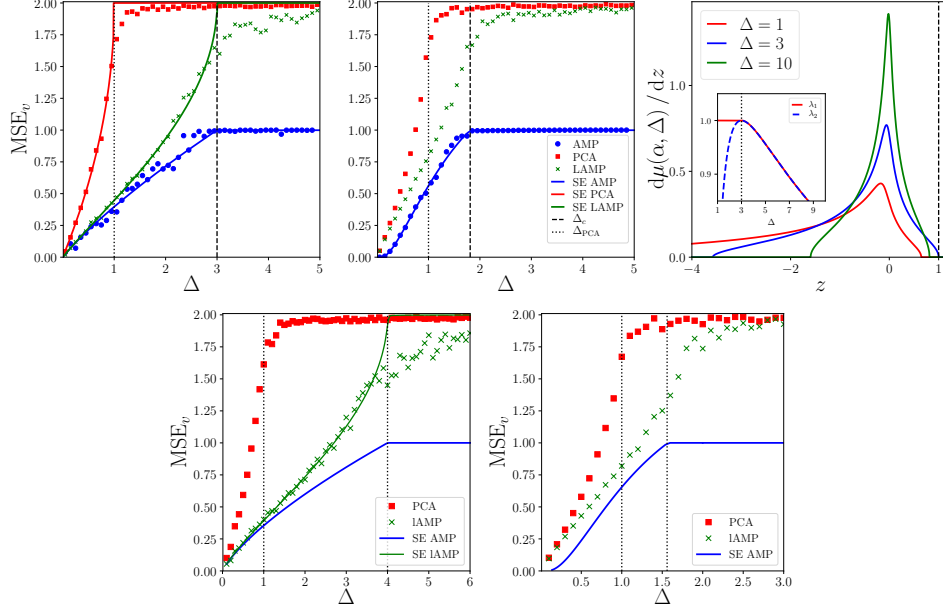

Figure 3: Comparison between PCA, LAMP and AMP - (**upper**) for (left) the linear, (center) and sign activations, for $L = 1$ and compression ratio $\alpha = 2$. Lines correspond to the theoretical asymptotic performance of PCA (red line), LAMP (green line) and AMP (blue line). Dots correspond to simulations of PCA (red squares), LAMP (green crosses) for $k = 10^4$ and AMP (blue points) for $k = 5.10^3$, $\sigma^2 = 1$. (Right) Illustration of the spectral phase transition in the matrix $\Gamma_p^{vv}$ eq. (18) at $\alpha = 2$ with an informative leading eigenvector with eigenvalue equal to 1 out of the bulk for $\Delta \leq 1 + \alpha$. We show the bulk spectral density $\mu(\alpha, \Delta)$. The inset shows the two leading eigenvalues - (**lower**) for (left) three layers generative model with $(\alpha_1, \alpha_2, \alpha_3) = (1, 1, 1)$ using linear activations ($k = 10^4$) (right) two layers generative model with $(\alpha_1, \alpha_2) = (1, 1)$ using sign activations ($k = 2.10^4$). The vertical lines show the PCA and the optimal threshold respectively.

## 4 Spectral methods for generative priors

Spectral methods are the most common class of algorithms used for spiked matrix estimation. For instance, canonical PCA estimates the spike from the leading eigenvector of the matrix $Y$. A classical result from Baik, Ben Arous and Péché (BBP) [38] shows that this eigenvector is correlated with the signal if and only if the signal-to-noise ratio $\rho_v^2/\Delta > 1$. For sparse separable priors (with $\rho_v^2 = \Theta(1)$), $\Delta_{\mathrm{PCA}} = \rho_v^2$ is also the threshold for AMP and it is conjectured that no polynomial algorithm can improve upon it [21]. In the previous section we show that for the analyzed generative priors AMP has a better threshold than PCA. Here we design a spectral method, called LAMP, that matches the AMP threshold and is hence superior over the canonical PCA. In order to do so, we follow the powerful strategy pioneered in [39] and linearize the AMP around its non-informative fixed point. In the spiked Wigner model with a single-layer prior ($L = 1$) the linearized AMP leads to the following operator:

$$\Gamma_p^{vv} = \frac{1}{\Delta} \left( (a - b)\mathrm{I}_p + b\frac{WW^\intercal}{k} + c\frac{\mathbb{1}_p \mathbb{1}_k^\intercal}{k} \frac{W^\intercal}{\sqrt{k}} \right) \times \left( \frac{Y}{\sqrt{p}} - a\mathrm{I}_p \right), \tag{16}$$

where parameters are moments of distributions $P_z$ and $Q_{\mathrm{out}}^0$ according to

$$a \equiv \rho_v, \quad b \equiv \rho_z^{-1}\mathbb{E}_{Q_{\mathrm{out}}^0}[vx]^2, \quad c \equiv \frac{1}{2}\rho_z^{-3}\mathbb{E}_{P_z}\left[z^3\right]\mathbb{E}_{Q_{\mathrm{out}}^0}[vx^2]\mathbb{E}_{Q_{\mathrm{out}}^0}[vx]. \tag{17}$$

We denote the spectral algorithm that takes the leading eigenvectors of (16) as LAMP (for linearized-AMP). Its derivation is presented in SM sec. 5 together with the one for the spiked Wishart model. For the specific case of Gaussian $z$ and prior (4) with the sign activation function we obtain $(a, b, c) = (1, 2/\pi, 0)$. For linear activation we get $(a, b, c) = (1, 1, 0)$, leading to

$$\Gamma_p^{vv} = \frac{1}{\Delta} K_p \left[ \frac{Y}{\sqrt{p}} - \mathrm{I}_p \right] \text{ with } K_p = \frac{[WW^\intercal]}{k} = \Sigma \approx \frac{1}{n}\sum_{\alpha=1}^{n} \mathbf{v}^\alpha (\mathbf{v}^\alpha)^\intercal, \tag{18}$$

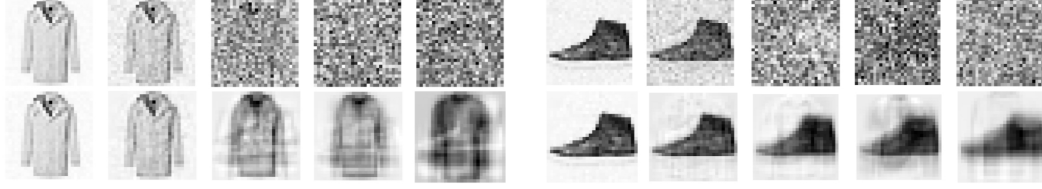

Figure 4: Illustration of canonical PCA (top line) and the LAMP (bottom line) spectral methods Alg. 2 on the spiked Wigner model. The covariance $K_p$ is estimated empirically, see (18), from the FashionMNIST database [40]. The estimation of the spike is shown for two images from FashionMNIST, with (from left to right), noise variance $\Delta = 0.01, 0.1, 1, 2, 10$.

where the last two equalities come from the fact that for the model (4) with linear activation and Gaussian separable $P_z$, $K_p$ is asymptotically equal to the covariance matrix between samples of spikes, $\Sigma$. The same observation holds for the sign activation function. In fact, the spectral method based on the matrix in eq. (18) can also be derived linearizing AMP with a Gaussian prior with covariance $\Sigma$. Interestingly, as the spectral method based on the matrix $K_p$ in eq. (18) can be empirically estimated directly from $n$ samples of spikes, $\mathbf{v}^\alpha$, $\alpha = 1, \dots, n$, without the knowledge of the generative model $(\varphi, W)$ itself, it suggests a simple practical implementation of LAMP Alg. 2 for any prior $P_v$.

**Input:** Observed matrix $Y \in \mathbb{R}^{p \times p}$, prior $P_v$ on $\mathbf{v} \in \mathbb{R}^p$

Take the leading eigenvector $\hat{\mathbf{v}} \in \mathbb{R}^p$ of $K_p \left[ \frac{Y}{\sqrt{p}} - \mathrm{I}_p \right]$ with $K_p = \mathbb{E}_{P_v} \left[ \mathbf{v}\mathbf{v}^\mathsf{T} \right]$ .

**Algorithm 2:** LAMP spectral algorithm

Analogously to the state evolution for AMP, the asymptotic performance of both PCA and LAMP can be evaluated in a closed-form for the spiked Wigner model with single-layer generative prior with linear activation (4). The corresponding expressions are derived in SM sec. 5 and plotted in Fig. 3 for the three considered algorithms that illustrates LAMP spectral method reaches the same threshold than ML-AMP for different depths $L$ and activations.

For illustration purposes, we display the behaviour of this spectral method on the spiked Wigner model with spikes coming from the Fashion-MNIST dataset in Fig. 4. A demonstration notebook is provided in [37], illustrating PCA and LAMP performances on Fashion-MNIST dataset.

Remarkably, the performance of the spectral method based on matrix (18) can be investigated independently of AMP using random matrix theory. An analysis of the random matrix (18) shows that a spectral phase transition for generative prior with linear activations appears at $\Delta_c = 1 + \alpha$ (as for AMP). This transition is analogous to the well-known BBP transition [38], but a non-GOE random matrix (18) needs to be analyzed. For the spiked Wigner models with linear generative prior we prove two theorems describing the behavior of the supremum of the bulk spectral density, the transition of the largest eigenvalue and the correlation of the corresponding eigenvector:

**Theorem 2** (Bulk of the spectral density, spiked Wigner, linear activation). *Let $\alpha, \Delta > 0$, then:*

($i$) *The spectral measure of $\Gamma_p^{vv}$ converges almost surely and in the weak sense to a compactly supported probability measure $\mu(\alpha, \Delta)$. We denote $\lambda_{\max}$ the supremum of the support of $\mu(\alpha, \Delta)$.*

($ii$) *For any $\alpha > 0$, as a function of $\Delta$, $\lambda_{\max}$ has a unique global maximum, reached exactly at the point $\Delta = \Delta_c(\alpha) = 1 + \alpha$. Moreover, $\lambda_{\max}(\alpha, \Delta_c(\alpha)) = 1$.*

**Theorem 3** (Transition of the largest eigenvalue and eigenvector, spiked Wigner, linear activation). *Let $\alpha > 0$. We denote $\lambda_1 \geq \lambda_2$ the first and second eigenvalues of $\Gamma_p^{vv}$. If $\Delta \geq \Delta_c(\alpha)$, then as $p \to \infty$ we have a.s. $\lambda_1 \to \lambda_{\max}$ and $\lambda_2 \to \lambda_{\max}$. If $\Delta \leq \Delta_c(\alpha)$, then as $p \to \infty$ we have a.s. $\lambda_1 \to 1$ and $\lambda_2 \to \lambda_{\max}$. Further, denoting $\tilde{\mathbf{v}}$ a normalized ($\|\tilde{\mathbf{v}}\|^2 = p$ ) eigenvector of $\Gamma_p^{vv}$ with eigenvalue $\lambda_1$, then $|\tilde{\mathbf{v}}^\mathsf{T}\mathbf{v}^\star|^2/p^2 \to \epsilon(\Delta)$ a.s., where $\epsilon(\Delta) = 0$ for all $\Delta \geq \Delta_c(\alpha)$, $\epsilon(\Delta) > 0$ for all $\Delta < \Delta_c(\alpha)$ and $\lim_{\Delta \to 0} \epsilon(\Delta) = 1$.*

Thm. 2 and Thm. 3 are illustrated in Fig. 3. The proof gives the value of $\epsilon(\Delta)$, which turns out to lead to the same MSE as in Fig. 3 in the linear case. We state the theorems counterparts for the $\mathbf{uv}^\mathsf{T}$ linear case in SM sec. 7. The proofs of the theorems and the precise arguments used to derive the eigenvalue density, the transition of $\lambda_1$ and the computation of $\epsilon(\Delta)$ are given in SM sec. 7, and a Mathematica demonstration notebook is also provided in [37]. We also describe in SM the difficulties to circumvent to generalize the analysis to a non-linear activation function with random matrix theory.

# 5 Acknowledgments

This work is supported by the ERC under the European Union's Horizon 2020 Research and Innovation Program 714608-SMiLe, as well as by the French Agence Nationale de la Recherche under grant ANR-17-CE23-0023-01 PAIL. We gratefully acknowledge the support of NVIDIA Corporation with the donation of the Titan Xp GPU used for this research. We thank Google Cloud for providing us access to their platform through the Research Credits Application program. We would also like to thank the Kavli Institute for Theoretical Physics (KITP) for welcoming us during part of this research, with the support of the National Science Foundation under Grant No. NSF PHY-1748958. We thank Ahmed El Alaoui for insightful discussions about the proof of the Bayes optimal performance, and Remi Monasson for his insightful lecture series that inspired partly this work. Additional funding is acknowledged by AM from 'Chaire de recherche sur les modèles et sciences des données', Fondation CFM pour la Recherche-ENS.

## Footnotes

† Université Paris-Saclay, CNRS, CEA, Institut de physique théorique, 91191, Gif-sur-Yvette, France.

[2]This result holds only for sparsity $\rho = \Theta(1)$. A line of works shows that when sparsity $k$ scales slower than linearly with $p$, algorithms more performant than PCA exist [22,24]

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
