[Supplementary Material · supplementary.pdf]

# The spiked matrix model with generative priors Supplementary materials

## Contents

# 1 Definitions and notations

In this section we recall the models introduced in the main body of the article, and introduce the notations used throughout the Supplementary Material.

## 1.1 Models

**Spiked Wigner model ($\mathbf{vv}^\intercal$):** Consider an unknown vector (the spike) $\mathbf{v}^\star \in \mathbb{R}^p$ drawn from a distribution $P_v$, we observe a matrix $Y \in \mathbb{R}^{p \times p}$ such that:

$$Y = \frac{1}{\sqrt{p}} \mathbf{v}^\star \mathbf{v}^{\star\intercal} + \sqrt{\Delta}\xi \,, \tag{1.1}$$

with symmetric noise $\xi \in \mathbb{R}^{p \times p}$ drawn from $\xi_{ij} \underset{\text{i.i.d.}}{\sim} \mathcal{N}(0,1)$ and $\Delta > 0$. The aim is to find back the hidden spike $\mathbf{v}^\star$ from the observation of $Y$.

**Spiked Wishart (or spiked covariance) model ($\mathbf{uv}^\intercal$):** Consider two unknown vectors $\mathbf{u}^\star \in \mathbb{R}^n$ and $\mathbf{v}^\star \in \mathbb{R}^p$ drawn from distributions $P_u$ and $P_v$, we observe $Y \in \mathbb{R}^{n \times p}$ such that

$$Y = \frac{1}{\sqrt{p}} \mathbf{u}^\star \mathbf{v}^{\star\intercal} + \sqrt{\Delta}\xi \,, \tag{1.2}$$

with noise $\xi \in \mathbb{R}^{n \times p}$ drawn $\xi_{i\mu} \underset{\text{i.i.d.}}{\sim} \mathcal{N}(0,1)$, $\Delta > 0$, and the goal is to find back the hidden spikes $\mathbf{u}^\star$ and $\mathbf{v}^\star$ from the observation of $Y$. We define the ratio between the spike dimensions $\beta = n/p$.

In either models, we are interested in the case where $\mathbf{v}^\star$ is given by a generative model. In the setting studied here the generative model is a fully-connected single-layer neural network (a.k.a. generalised linear model) with Gaussian random weights $W \in \mathbb{R}^{p \times k}$, $W_{il} \underset{\text{i.i.d.}}{\sim} \mathcal{N}(0,1)$ and latent variable $\mathbf{z}^\star \in \mathbb{R}^k$ drawn from a given factorised distribution $P_z$,

$$\mathbf{v}^\star = \varphi\left(\frac{1}{\sqrt{k}} W \mathbf{z}^\star\right) \quad \text{with} \quad z_l^\star \underset{\text{i.i.d.}}{\sim} P_z, \tag{1.3}$$

where $\varphi : \mathbb{R} \to \mathbb{R}$ is the activation function, a real-valued function acting component-wise on $\mathbb{R}^p$ that can be deterministic or stochastic. An equivalent formulation of eq. (1.3) is

$$\mathbf{v}^\star \sim P_{\text{out}}\left(\cdot \Big| \frac{1}{\sqrt{k}} W \mathbf{z}^\star\right). \tag{1.4}$$

For instance, a deterministic layer with activation $\varphi$ is written in this formulation as $P_{\text{out}}(v|x) = \delta(v - \varphi(x))$. We define the compression rate of the signal as $\alpha = p/k$.

Although we will mainly focus on the single-layer model, some of our results apply more broadly to any generative prior with a well-defined free energy density in the thermodynamic limit. In particular, we will mention the example of a fully-connected multi-layer generative prior, given by

$$\mathbf{v}^\star = \varphi^{(L)}\left(\frac{1}{\sqrt{k_L}} W^{(L)} \cdots \varphi^{(1)}\left(\frac{1}{\sqrt{k_1}} W^{(1)} \mathbf{z}\right)\right) \quad \text{with} \quad z_l^\star \underset{\text{i.i.d.}}{\sim} P_z \tag{1.5}$$

where now $\{\varphi^{(l)}\}_{1 \le l \le L}$ are a family of real-valued component-wise activation functions and $W_{\nu_l \nu_{l-1}}^{(l)} \underset{\text{i.i.d.}}{\sim} \mathcal{N}(0,1)$ are independently drawn random weights. The equivalent probabilistic formulation of the multi-layer case is

$$\mathbf{v} \sim P_{\text{out}}^{(L)}\left(\cdot \Big| \frac{1}{\sqrt{k_L}} W^{(L)} \mathbf{h}^{(L)}\right), \qquad\qquad \mathbf{v} \in \mathbb{R}^p$$

$$\mathbf{h}^{(L)} \sim P_{\text{out}}^{(L-1)}\left(\cdot \Big| \frac{1}{\sqrt{k_{L-1}}} W^{(L-1)} \mathbf{h}^{(L-1)}\right), \qquad\qquad \mathbf{h}^{(L)} \in \mathbb{R}^{k_L}$$

$$\vdots$$

$$\mathbf{h}^{(2)} \sim P_{\text{out}}^{(1)}\left(\cdot \Big| \frac{1}{\sqrt{k_1}} W^{(1)} \mathbf{z}\right), \qquad\qquad \mathbf{h}^{(2)} \in \mathbb{R}^{k_2}$$

$$\mathbf{z} \underset{\text{i.i.d.}}{\sim} P_z, \qquad\qquad \mathbf{z} \in \mathbb{R}^{k_1} \tag{1.6}$$

where we introduced the hidden variables $\mathbf{h}^{(l)} \in \mathbb{R}^{k_l}$ for $2 \le l \le L$ and the family of densities $\left\{ P_{\text{out}}^{(l)} \right\}_{1 \le l \le L}$. In this case, we define the compression rate as the ratio between the dimensions of the latent variable in the first layer $\mathbf{z} \in \mathbb{R}^{k_1}$ and the signal $\mathbf{v} \in \mathbb{R}^p$, $\alpha = p/k_1$. It is also useful to define the compression at each layer, $\alpha_l = k_l/k_1$. The thermodynamic limit for this generative model is defined by taking $p \to \infty$ while keeping all $\alpha, \alpha_l \sim O(1)$, $1 \le l \le L$. As one might expect, the single-layer generative prior is a particular case with $L = 1$.

## 1.2  Bayesian inference and posterior distribution

Since the information about the generative model $P_v$ of the spike is given, the optimal estimator for $\mathbf{v}^\star$ is the mean of its posterior distribution, $\hat{\mathbf{v}}^{\text{opt}} = \mathbb{E}_{P(\mathbf{v}^\star|Y)} \mathbf{v}$, which in general reads

$$P(\mathbf{v}^\star|Y) = \frac{1}{P(Y)} P_v(\mathbf{v}^\star) \prod_{1 \le i < j \le p} \frac{1}{\sqrt{2\pi\Delta}} e^{-\frac{1}{2\Delta}\left(Y_{ij} - \frac{v_i^\star v_j^\star}{\sqrt{p}}\right)^2}, \tag{1.7}$$

for the $\mathbf{v}\mathbf{v}^\intercal$ model and by

$$P(\mathbf{v}^\star|Y) = \frac{1}{P(Y)} P_v(\mathbf{v}^\star) \int_{\mathbb{R}^n} d\mathbf{u} \, P_u(\mathbf{u}) \prod_{1 \le i \le p, 1 \le \mu \le n} \frac{1}{\sqrt{2\pi\Delta}} e^{-\frac{1}{2\Delta}\left(Y_{\mu i} - \frac{u_\mu^\star v_i^\star}{\sqrt{p}},\right)^2} \tag{1.8}$$

for the $\mathbf{u}\mathbf{v}^\intercal$ model. In both cases the evidence $P(Y)$ is fixed as the normalisation of the posterior. In the specific case of a single-layer generative model from eq. (1.3), we can be more explicit and write the prior for $\mathbf{v}^\star$ explicitly

$$P_v(\mathbf{v}^\star) = \int_{\mathbb{R}^k} d\mathbf{z}^\star P_z(\mathbf{z}^\star) \prod_{i=1}^p P_{\text{out}}\left( v_i^\star \Big| \frac{1}{\sqrt{k}} \sum_{l=1}^k W_{il} z_l^\star \right). \tag{1.9}$$

The multi-layer case is written similarly by integrating over the intermediate hidden variables and their respective distributions. It is important to stress that we assume the structure of the generative model is known, i.e. $(P_z, P_{\text{out}}, W)$ (and $P_u$ in the $\mathbf{u}\mathbf{v}^\intercal$ case) are given and the only unknowns of the problem are the spike $\mathbf{v}^\star$ and the corresponding latent variable $\mathbf{z}^\star$. This setting, in which the Bayesian estimator is optimal, is commonly refereed as the *Bayes-optimal inference*.

In principle eqs. (1.7) and (1.8) are of little use, since sampling from these high-dimensional distributions is a hard problem. Luckily, physicists have been dealing with high-dimensional distributions - such as the Gibbs measure in statistical physics - for a long time. The *replica trick* and the *approximate message passing* (AMP) algorithm presented in the main body of the paper are two of the statistical physics inspired techniques we borrow to circumvent the hindrance of dimensionality.

**Summary of the Supplementary Material:**  A detailed account of the derivation of eq. (7) from the replica method is given in Section 2. Although the replica calculation is not mathematically rigorous, it gives a constructive method to compute the mutual information. The final expression can be made rigorous using an interpolation method, which we detail in Section 3. The sketch for the derivation of the AMP algorithm 1 and its associated spectral algorithm in eq. (16) are discussed respectively in Section 4 and 5. We detail the stability analysis of the state evolution equations leading to the transition point for generic activation function in Section 6, and finally we present a rigorous proof for the transition in the case of linear activation in Section 7.

## 1.3  Notation and conventions

**Index convention:**  In the whole paper, we use the convention that indices $\mu$, $i$ and $l$ correspond respectively to variables $\mathbf{u}$, $\mathbf{v}$ and $\mathbf{z}$ such that $\mu \in [1 : n]$, $i \in [1 : p]$ and $l \in [1 : k]$.

Unless otherwise stated, $\xi, \eta \in \mathbb{R}$ denote independent random variables variables distributed according to $\mathcal{N}(0, 1)$.

**Normalised second moments**

We define $\rho_v$ as the normalised second moments of the priors $P_v, P_u$ and $P_z$ respectively,

$$\rho_v = \lim_{p \to \infty} \mathbb{E}_{P_v}\left[\frac{\mathbf{v}^\intercal \mathbf{v}}{p}\right], \qquad \rho_u = \lim_{n \to \infty} \mathbb{E}_{P_u}\left[\frac{\mathbf{u}^\intercal \mathbf{u}}{n}\right], \qquad \rho_z = \lim_{z \to \infty} \mathbb{E}_{P_z}\left[\frac{\mathbf{z}^\intercal \mathbf{z}}{k}\right]. \tag{1.10}$$

In the case we consider $P_z(\mathbf{z}) = \prod\limits_{l=1}^{k} P_z(z_l)$, $\rho_z$ is simply the one-dimensional second moment of $P_z$

$$\rho_z = \mathbb{E}_{P_z} z^2. \tag{1.11}$$

In the case $P_v$ is the single-layer generative model in eq. (1.9) with $W_{il} \underset{\text{i.i.d.}}{\sim} \mathcal{N}(0,1)$ and $z_l \underset{\text{i.i.d.}}{\sim} P_z$, $\rho_v$ is self-averaging in the thermodynamic limit and is given by

$$\rho_v = \mathbb{E}_{Q_{\text{out}}^0} v^2, \tag{1.12}$$

where $Q_{\text{out}}^0$ is defined below in eq. (1.16).

**Denoising distributions**

The upshot of the replica calculation is that the high dimensional mutual information between the spike and the data $I(Y, \mathbf{v}^\star)$ is given by a simple one-dimensional expression, c.f. the right-hand side of the main part eq. (7). This expression can be interpreted as the mutual information of a one-dimensional denoising problem.

Below we introduce the one-dimensional probability densities appearing in the factorised mutual information, from which the free energy and the AMP update equations are derived from:

$$Q_u(u; B, A) \equiv \frac{1}{\mathcal{Z}_u(B, A)} P_u(u) e^{-\frac{1}{2}Au^2 + Bu}, \tag{1.13}$$

$$Q_z(z; \gamma, \Lambda) \equiv \frac{1}{\mathcal{Z}_z(\gamma, \Lambda)} P_z(z) e^{-\frac{1}{2}\Lambda z^2 + \gamma z}, \tag{1.14}$$

$$Q_{\text{out}}(v, x; B, A, \omega, V) \equiv \frac{1}{\mathcal{Z}_{\text{out}}(B, A, \omega, V)} e^{-\frac{1}{2}Av^2 + Bv} P_{\text{out}}(v|x) \frac{e^{-\frac{1}{2}V^{-1}(x-\omega)^2}}{\sqrt{2\pi V}}, \tag{1.15}$$

$$Q_{\text{out}}^0(v, x; \rho_z) \equiv Q_{\text{out}}(v, x; 0, 0, 0, \rho_z) = \frac{1}{\mathcal{Z}_{\text{out}}^0} P_{\text{out}}(v|x) \frac{e^{-\frac{1}{2\rho_z}x^2}}{\sqrt{2\pi\rho_z}}. \tag{1.16}$$

**Free entropy terms**

The mutual information density can be written in terms of the partition functions of the denoising distributions above as:

$$\Psi_u(x) \equiv \mathbb{E}_\xi\left[\mathcal{Z}_u\left(x^{1/2}\xi, x\right)\log\left(\mathcal{Z}_u\left(x^{1/2}\xi, x\right)\right)\right], \tag{1.17}$$

$$\Psi_z(x) \equiv \mathbb{E}_\xi\left[\mathcal{Z}_z\left(x^{1/2}\xi, x\right)\log\left(\mathcal{Z}_z\left(x^{1/2}\xi, x\right)\right)\right], \tag{1.18}$$

$$\Psi_{\text{out}}(x, y) \equiv \mathbb{E}_{\xi,\eta}\left[\mathcal{Z}_{\text{out}}\left(x^{1/2}\xi, x, y^{1/2}\eta, \rho_z - y\right)\log\left(\mathcal{Z}_{\text{out}}\left(x^{1/2}\xi, x, y^{1/2}\eta, \rho_z - y\right)\right)\right]. \tag{1.19}$$

**AMP update functions**

Similarly, the update functions appearing in AMP are also given in terms of the moments of the above denoising distributions:

$$f_u(B, A) \equiv \partial_B \log(\mathcal{Z}_u) = \mathbb{E}_{Q_u}[u], \quad \partial_B f_u(B, A) \equiv \mathbb{E}_{Q_u}[u^2] - (f_u)^2 \tag{1.20}$$

$$f_z(\gamma, \Lambda) \equiv \partial_\gamma \log(\mathcal{Z}_z) = \mathbb{E}_{Q_z}[z], \quad \partial_\gamma f_z(\gamma, \Lambda) \equiv \mathbb{E}_{Q_z}[z^2] - (f_z)^2 \tag{1.21}$$

$$f_v(B, A, \omega, v) \equiv \partial_B \log(\mathcal{Z}_{\text{out}}) = \mathbb{E}_{Q_{\text{out}}}[v], \quad \partial_B f_v(B, A, \omega, v) \equiv \mathbb{E}_{Q_{\text{out}}}[v^2] - (f_v)^2 \tag{1.22}$$

$$f_{\text{out}}(B, A, \omega, v) \equiv \partial_\omega \log(\mathcal{Z}_{\text{out}}) = V^{-1}\mathbb{E}_{Q_{\text{out}}}[x - \omega], \quad \partial_\omega f_{\text{out}}(B, A, \omega, v) \equiv \frac{\partial f_{\text{out}}}{\partial \omega} \tag{1.23}$$

## 2 Mutual information from the replica trick

In this section we give a derivation for the mutual information formula in main part eq. (7) from the *replica trick*. The derivation is detailed for the symmetric $\mathbf{v}\mathbf{v}^{\mathsf{T}}$ model, since the derivation for the asymmetric $\mathbf{u}\mathbf{v}^{\mathsf{T}}$ model follows exactly the same steps. In both cases, it closely follows the calculation of the replica free energy of the spiked matrix model with factorized prior in [1].

Before diving into the derivation, we note that the formula in main part eq. (7) actually holds for any channel of the form

$$P(Y|\omega) = \prod_{1 \leq i < j \leq p} e^{g(Y_{ij}, \omega_{ij})}, \tag{2.1}$$

where $\omega \in \mathbb{R}^{p \times p}$ is a matrix with components $\omega_{ij} \equiv \frac{v_i v_j}{\sqrt{p}}$ and $g : \mathbb{R}^2 \to \mathbb{R}$ is any two-dimensional real function such that $P(Y|\omega)$ is properly normalised. The gaussian noise in eq. (1) is a particular case given by $g(Y, \omega) = -\frac{1}{2\Delta}(Y - \omega)^2 - \frac{1}{2} \log 2\pi\Delta$.

The first step in the derivation is to note that the mutual information $I(Y, \mathbf{v}^\star)$ between the observed data $Y$ and the spike $\mathbf{v}^\star$ can be writen as

$$I(Y, \mathbf{v}^\star) = \frac{1}{4\Delta} \mathbb{E}_{P_v} \left[ \mathbf{v}^{\star\mathsf{T}} \mathbf{v}^\star \right]^2 - \mathbb{E}_Y \log \mathcal{Z}(Y), \tag{2.2}$$

where

$$\mathcal{Z}(Y) = \int_{\mathbb{R}^p} \mathrm{d}\mathbf{v} \, P_v(\mathbf{v}) \prod_{1 \leq i < j \leq p} e^{g(Y_{ij}, \omega_{ij}) - g(Y_{ij}, 0)}. \tag{2.3}$$

Note that since the data is generated from a planted spike $\mathbf{v}^\star$, we have $Y = Y(\mathbf{v}^\star)$, and therefore the partition function $\mathcal{Z}$ depends on $\mathbf{v}^\star$ implicitly through $Y$.

### 2.1 Derivation of the replica free energy for the $\mathbf{v}\mathbf{v}^{\mathsf{T}}$ model

The partition function $\mathcal{Z}$ is a $p$-dimensional integral, and computing the average over $Y$ (a $p \times p$ integral) of $\log \mathcal{Z}$ seems hopeless. The replica trick is a way to surmount this hindrance. It consists of writing

$$\mathbb{E}_Y \log Z = \lim_{r \to 0^+} \frac{1}{r} \left( \mathbb{E}_Y \mathcal{Z}^r - 1 \right). \tag{2.4}$$

Note that $\mathcal{Z}^r$ is the partition function of $r$ non-interacting copies (named in the physics literature and hereafter *replicas*) of the initial system. The average over the *replicated partition function* $\mathcal{Z}^r$ can be conveniently written as

$$\mathbb{E}_Y \mathcal{Z}^r = \int \prod_{1 \leq i < j \leq p} \mathrm{d}Y_{ij} \, e^{g(Y_{ij}, 0)} \int_{\mathbb{R}^{p \times (r+1)}} \prod_{a=0}^{r} \mathrm{d}\mathbf{v}^a P_v \left( \mathbf{v}^a \right) \prod_{a=0}^{r} \prod_{1 \leq i < j \leq p} e^{g\left(Y_{ij}, \omega_{ij}^a\right) - g(Y_{ij}, 0)}, \tag{2.5}$$

where in the second line we have defined

$$\mathbf{v}^a = \begin{cases} \mathbf{v}^\star & \text{for } a = 0 \\ \mathbf{v}^a & \text{for } 1 \leq a \leq r. \end{cases} \tag{2.6}$$

**Averaging over $Y$**

The key observation to simplify the integrals in eq. (2.5) is to note that $\omega_{ij}$ is of order $1/\sqrt{p}$, and therefore in the large-$p$ limit of interest, we can keep only terms of order $1/p$,

$$\exp\left( \sum_{a=0}^{r} \left[ g(Y_{ij}, \omega_{ij}^a) - g(Y_{ij}, 0) \right] \right) = 1 + \sum_{a=0}^{r} (\partial_\omega g)_{\omega=0} \, \omega_{ij}^a + \frac{1}{2} \sum_{a=0}^{r} (\partial_\omega^2 g)_{\omega=0} \left( \omega_{ij}^a \right)^2$$

$$+ \frac{1}{2} \sum_{a,b=0}^{r} (\partial_\omega g)_{\omega=0}^2 \, \omega_{ij}^a \omega_{ij}^b + \mathcal{O}\left( p^{-3/2} \right) \tag{2.7}$$

From the normalisation condition of $P(Y|\omega)$, we can derive the following relations

$$\int \prod_{1 \leq i < j \leq p} dY_{ij} \, e^{g(Y_{ij},0)} = 1,$$

$$\int \prod_{1 \leq i < j \leq p} dY_{ij} \, e^{g(Y_{ij},0)} \, (\partial_\omega g)_{\omega=0} = 0,$$

$$\int \prod_{1 \leq i < j \leq p} dY_{ij} \, e^{g(Y_{ij},0)} \left[ \partial_\omega^2 g + (\partial_\omega g)^2 \right]_{\omega=0} = 0. \tag{2.8}$$

Further defining

$$\Delta^{-1} = \int \prod_{1 \leq i < j \leq p} dY_{ij} \, e^{g(Y_{ij},0)} \, (\partial_\omega g)_{\omega=0}^2, \tag{2.9}$$

allows us to evaluate the integral over $Y$ term by term in the expansion in eq. (2.7),

$$\mathbb{E}_Y \mathcal{Z}^r = \int_{\mathbb{R}^{p \times (r+1)}} \prod_{a=0}^{r} d\mathbf{v}^a \, P_v\left(\mathbf{v}^a\right) \prod_{1 \leq i < j \leq p} \left[ 1 + \frac{1}{2\Delta} \sum_{0 \leq a < b \leq r} \omega_{ij}^a \omega_{ij}^b + \mathcal{O}\left(p^{-3/2}\right) \right]$$

$$= \int_{\mathbb{R}^{p \times (r+1)}} \prod_{a=0}^{r} d\mathbf{v}^a \, P_v\left(\mathbf{v}^a\right) \prod_{1 \leq i < j \leq p} e^{\frac{1}{2\Delta} \sum_{0 \leq a < b \leq r} \omega_{ij}^a \omega_{ij}^b} + \mathcal{O}\left(p^{-3/2}\right). \tag{2.10}$$

The upshot of this expansion is that on the large-$p$ limit $\Delta$ is the only relevant parameter we need from the channel. Therefore, from the perspective of the mutual information density, a channel with parameter $\Delta$ is completely equivalent to a Gaussian channel with variance $\Delta$. This property is known as *channel universality* [1].

**Rewritting as a saddle-point problem**

Note that we can rewrite

$$\sum_{1 \leq i < j \leq p} \omega_{ij}^a \omega_{ij}^b = \frac{1}{p} \sum_{1 \leq i < j \leq n} v_i^a v_j^a v_i^b v_j^b = \frac{p}{2} \left(q_v^{ab}\right)^2, \tag{2.11}$$

where we defined the overlap between two replicas as $q_v^{ab} = p^{-1} \sum_{i=1}^{p} v_i^a v_i^b$. This allows us to write the average over the replicated partition function as a function of a set of order parameters $q_v^{ab}$, and therefore to factorise all the index $i$ dependence of the exponential,

$$\mathbb{E}_Y \mathcal{Z}^r = \int_{\mathbb{R}^{p \times (r+1)}} \prod_{a=0}^{r} d\mathbf{v}^a \, P_v\left(\mathbf{v}^a\right) e^{\frac{p}{4\Delta} \sum_{0 \leq a < b \leq r} \left(q_v^{ab}\right)^2}. \tag{2.12}$$

Since the expression above only depends on $q_v^{ab}$ now, we exchange the integral over the spike for an integral over this order parameter by introducing

$$1 \propto \int_{\mathbb{R}^{(r+1) \times (r+1)}} \prod_{0 \leq a < b \leq r} dq_v^{ab} \prod_{0 \leq a < b \leq r} \delta \left( \sum_{i=1}^{p} q_v^{ab} - p q_v^{ab} \right)$$

$$\propto \int_{\mathbb{R}^{(r+1) \times (r+1)}} \prod_{0 \leq a < b \leq r} dq_v^{ab} \int_{(i\mathbb{R})^{(r+1) \times (r+1)}} \prod_{0 \leq a < b \leq r} \hat{q}_v^{ab} e^{-p \sum_{0 \leq a < b \leq r} \hat{q}_v^{ab} q_v^{ab} + \sum_{0 \leq a < b \leq r} \hat{q}_v^{ab} \sum_{i=1}^{p} v_i^a v_i^b}$$

$$\tag{2.13}$$

Note that we neglected some constants and made a rotation to the complex axis over the Fourier integral. These will not be important for the argument that follows.

Inserting this identity in eq. (2.12) yields

$$\mathbb{E}_Y \mathcal{Z}^r \propto \int_{\mathbb{R}^{(r+1)\times(r+1)}} \prod_{0 \le a < b \le r} \mathrm{d}q_v^{ab} \int_{(i\mathbb{R})^{(r+1)\times(r+1)}} \prod_{0 \le a < b \le r} \hat{q}_v^{ab} \, e^{p\Phi^{(r)}\left(q^{ab}, \hat{q}^{ab}\right)},$$

$$\Phi^{(r)}(q_v^{ab}, \hat{q}_v^{ab}) = \frac{1}{4\Delta} \sum_{0 \le a < b \le r} \left(q_v^{ab}\right)^2 - \sum_{0 \le a < b \le r} \hat{q}_v^{ab} q_v^{ab} + \Psi_v^{(r)}(\hat{q}_v^{ab}), \qquad (2.14)$$

where $\Psi_v^{(r)}(\hat{q}_v^{ab})$ contains all the information about the prior $P_v$:

$$\Psi_v^{(r)}(\hat{q}_v^{ab}) = \frac{1}{p} \log \int_{\mathbb{R}^{p \times (r+1)}} \prod_{a=0}^r \mathrm{d}\mathbf{v}^a \, P_v\left(\mathbf{v}^a\right) \prod_{i=1}^p e^{\sum_{0 \le a < b \le p} v_i^a \hat{q}_v^{ab} v_i^b}. \qquad (2.15)$$

Note that when the prior factorises, $P_v(\mathbf{v}) = \prod_{i=1}^p P_v(v_i)$, $\Psi_v^{(r)}$ is given by a simple one-dimensional integral. However in the case of a generative model for $\mathbf{v}$, $P_v$ is kept general.

We are interested in the mutual information density in the thermodynamic limit. According to eq. (2.35), this is given by

$$\lim_{p \to \infty} i_p(Y, \mathbf{v}^\star) = \lim_{p \to \infty} \frac{1}{p} I(Y, \mathbf{v}^\star) = \frac{1}{4\Delta} \lim_{p \to \infty} \mathbb{E}_{P_v} \left[ \frac{\mathbf{v}^{\star\mathsf{T}} \mathbf{v}^\star}{p} \right] - \lim_{p \to \infty} \frac{1}{p} \mathbb{E}_Y \log \mathcal{Z} \qquad (2.16)$$

$$= \frac{\rho_v^2}{4\Delta} - \lim_{r \to 0^+} \frac{1}{r} \left( \lim_{p \to \infty} \frac{1}{p} \mathbb{E}_Y \mathcal{Z}^r \right). \qquad (2.17)$$

where we assumed that $\rho_v$, the re-scaled second moment of $P_v$, remains finite and that we can commute the $r \to 0^+$ and the $p \to \infty$ limit. Since $\mathbb{E}_Y \mathcal{Z}^r$ is given in terms of an integral weighted by $e^{p\Psi^{(r)}}$, in the limit $p \to \infty$ the integral will be dominated by the configurations of $(q_v^{ab}, \hat{q}^{ab})$ that extremise the potential $\Psi^{(r)}$. This extremality condition, known as the Laplace method, yields the following *saddle-point equations*,

$$\hat{q}_v^{ab} = \frac{1}{2\Delta} q_v^{ab}, \qquad\qquad q_v^{ab} = \lim_{p \to \infty} \partial_{\hat{q}_v} \Psi_v^{(r)}(\hat{q}_v^{ab}). \qquad (2.18)$$

where we also assume that $P_v$ is such that $\Psi_v^{(r)}$ remains well defined in the limit $p \to \infty$.

**Replica symmetric solution**

Enforcing the first saddle-point equation allow us to write

$$\lim_{p \to \infty} \frac{1}{p} \mathbb{E}_Y \mathcal{Z}^r = \underset{q_v^{ab}}{\mathbf{extr}} \left[ -\frac{1}{2\Delta} \sum_{0 \le a < b \le r} \left(q_v^{ab}\right)^2 + \lim_{p \to \infty} \Psi_v^{(r)} \left( \frac{q_v^{ab}}{\Delta} \right) \right] \qquad (2.19)$$

Solving this extremisation problem for general matrices is cumbersome. We therefore restrict ourselves to solutions that are *replica symmetric*

$$q_v^{ab} = q_v \qquad \text{for} \qquad 0 \le a \le r. \qquad (2.20)$$

The replica symmetry assumption might seen restrictive, but it is justified in the Bayes-optimal case under consideration - see [2]. Replica symmetry allow us to factor the $r$ dependence explicitly for each term,

$$\sum_{0 \le a < b \le r} \left(q_v^{ab}\right)^2 = \frac{r(r+1)}{2} q_v^2, \qquad \sum_{0 \le a < b \le r} v_i^a q_v^{ab} v_i^b = q_v v^\star \sum_{a=1}^r v_i^a + q_v \sum_{a,b=1}^r v_i^a v_i^b \qquad (2.21)$$

the last sum that couples $a, b$ can be decoupled using

$$e^{\frac{q_v}{2} \sum_{a,b=1}^r v_i^a v_i^b} = \mathbb{E}_\xi \left[ e^{-\sqrt{q_v}\xi \sum_{a=1}^r (v_i^a)^2} \right] \qquad (2.22)$$

where $\xi \sim \mathcal{N}(0,1)$. This transformation factorise $\Psi_v^{(r)}$ in replica space,

$$\Psi_v^{(r)}(q_v) = \frac{1}{p}\log \int_{\mathbb{R}^p} \mathbf{dv}^\star \, P_v\left(\mathbf{v}^\star\right) \int_{\mathbb{R}} \frac{\mathrm{d}\xi}{\sqrt{2\pi}} e^{-\frac{1}{2}\xi^2} \left[\int_{\mathbb{R}^p} \mathbf{dv}\, P_v(\mathbf{v}) \prod_{i=1}^p e^{-\frac{q_v}{2\Delta} v_i^2 + \left(\frac{q_v}{\Delta} v_i^\star + \sqrt{\frac{q}{\Delta}}\xi\right) v_i}\right]^r$$

$$\underset{r\to 0^+}{=} \frac{r}{p}\mathbb{E}_{\xi, P_v(\mathbf{v}^\star)} \log \int_{\mathbb{R}^p} \mathbf{dv}\, P_v(\mathbf{v}) \prod_{i=1}^p e^{-\frac{q_v}{2\Delta} v_i^2 + \left(\frac{q_v}{\Delta} v_i^\star + \sqrt{\frac{q}{\Delta}}\xi\right) v_i} + \mathcal{O}\left(r^2\right). \tag{2.23}$$

allowing us to take the $r \to 0^+$ limit explicitly, and giving the following partial result

$$\lim_{p\to\infty} i_p(Y, \mathbf{v}^\star) = \frac{\rho_v^2}{4\Delta} + \underset{q_v}{\mathbf{extr}} \left[\frac{1}{4\Delta} q_v^2 - \lim_{p\to\infty} \Psi_v\left(\frac{q_v}{\Delta}\right)\right], \tag{2.24}$$

where

$$\Psi_v\left(\frac{q_v}{\Delta}\right) = \lim_{r\to 0^+} \Psi_v^{(r)} = \frac{1}{p}\mathbb{E}_{\xi, P_v(\mathbf{v}^\star)} \log \mathbb{E}_{P_v(\mathbf{v})} \left[\prod_{i=1}^p e^{-\frac{q_v}{2\Delta} v_i^2 + \left(\frac{q_v}{\Delta} v_i^\star + \sqrt{\frac{q_v}{\Delta}}\xi\right) v_i}\right]. \tag{2.25}$$

**Interpretations of $\Psi_v$ as a mutual information**

The prior term $\Psi_v$ in the free energy has an interesting interpretation as the mutual information of an effective denoising problem over $\mathbf{v}$. To see this, we complete the square in the exponential of eq. (2.25),

$$\Psi_v\left(x\right) = \frac{1}{p}\mathbb{E}_{\xi, P_v(\mathbf{v}^\star)} \log \int_{\mathbb{R}^p} \mathbf{dv}\, P_v(\mathbf{v}) \prod_{i=1}^p e^{-\frac{x}{2}\left[v_i - \left(v_i^\star + x^{-1/2}\xi\right)\right]^2 + \frac{x}{2}\left(v_i^\star + x^{-1/2}\xi\right)^2},$$

$$= \frac{x}{2p}\mathbb{E}_{\xi, P_v(\mathbf{v}^\star)} \sum_{i=1}^p \left(v_i^\star + x^{-1/2}\xi\right)^2 + \frac{1}{p}\mathbb{E}_{\xi, P_v(\mathbf{v}^\star)} \log \int_{\mathbb{R}^p} \mathbf{dv}\, P_v(\mathbf{v}) \prod_{i=1}^p e^{-\frac{x}{2}\left[v_i - \left(v_i^\star + x^{-1/2}\xi\right)\right]^2},$$

$$= \frac{x}{2}\mathbb{E}_{P_v} \left[\frac{\mathbf{v}^\intercal \mathbf{v}}{p}\right] + \frac{1}{2} + \frac{1}{p}\mathbb{E}_{\xi, P_v(\mathbf{v}^\star)} \log \int_{\mathbb{R}^p} \mathbf{dv}\, P_v(\mathbf{v}) \prod_{i=1}^p e^{-\frac{x}{2}\left[v_i - \left(v_i^\star + x^{-1/2}\xi\right)\right]^2}. \tag{2.26}$$

The last integral is a convolution between the prior $P_v$ and a un-normalised Gaussian. Up to an aditive constant it admits a natural representation as the mutual information of a denoising problem,

$$\frac{1}{p}\mathbb{E}_{\xi, P_v(\mathbf{v}^\star)} \log \int_{\mathbb{R}^p} \mathbf{dv}\, P_v(\mathbf{v}) \prod_{i=1}^p e^{-\frac{x}{2}\left[v_i - \left(v_i^\star + x^{-1/2}\xi\right)\right]^2} = -\frac{1}{p}I(\mathbf{v}^\star; \mathbf{v}^\star + x^{-1/2}\xi) - \frac{1}{2}. \tag{2.27}$$

Putting together with eq. (2.26) and taking the limit,

$$\lim_{p\to\infty} \Psi_v\left(\frac{q_v}{\Delta}\right) = \frac{q_v \rho_v}{2\Delta} - \lim_{p\to\infty} \frac{1}{p}I\left(\mathbf{v}^\star; \mathbf{v}^\star + \sqrt{\frac{\Delta}{q_v}}\xi\right). \tag{2.28}$$

Together with eq. (2.24), this representation lead to eq. (6) in the main article.

Interestingly, the signal to noise ratio in the effective denoising problem is proportional to $\Delta$ and inversely proportional to the overlap $q_v$. This is quite intuitive: when $\Delta \gg 1$ (or the overlap with the ground truth is small), denoising is hard. On the other hand, when $\Delta = 0$ the mutual information reaches its upper bound, given by the entropy of $P_v$.

## 2.2 Free energy for the $uv^\intercal$ model

The exact same steps outlined above can be followed for the spiked Wishart model with spikes $\mathbf{u}^\star \in \mathbb{R}^n$ and $\mathbf{v}^\star \in \mathbb{R}^p$ drawn from non-factorisable priors $P_u$ and $P_v$ respectively. In this case, the free energy density associated with the following partition function

$$\mathcal{Z}^{uv}(Y) = \int_{\mathbb{R}^p} \mathbf{dv}\, P_v\left(\mathbf{v}\right) \int_{\mathbb{R}^n} \mathbf{du}\, P_u\left(\mathbf{u}\right) \prod_{\mu=1}^n \prod_{i=1}^p e^{g\left(Y_{\mu i}, \frac{u_\mu v_i}{\sqrt{p}}\right) - g(Y_{\mu i}, 0)} \tag{2.29}$$

is given by

$$\lim_{p\to\infty}\frac{1}{p}\mathbb{E}_Y\log\mathcal{Z}^{uv}=\underset{q_u,q_v}{\mathbf{extr}}\left[\frac{\beta}{2\Delta}q_uq_v-\lim_{p\to\infty}\Psi_v\left(\beta\frac{q_u}{\Delta}\right)-\beta\lim_{n\to\infty}\Psi_u\left(\frac{q_v}{\Delta}\right)\right]\qquad(2.30)$$

with $\beta=n/p$ fixed. The functions $\Psi_v,\Psi_u$ are given by

$$\Psi_u\left(\beta\frac{q_v}{\Delta}\right)=\frac{1}{n}\mathbb{E}_{\xi,P_u(\mathbf{u}^\star)}\log\int_{\mathbb{R}^n}d\mathbf{u}\,P_u(\mathbf{u})\prod_{\mu=1}^{n}e^{-\beta\frac{q_v}{2\Delta}u_\mu^2+\left(\beta\frac{q_v}{\Delta}u_\mu^\star+\sqrt{\beta\frac{q_v}{\Delta}}\xi\right)u_\mu}$$

$$\Psi_v\left(\frac{q_u}{\Delta}\right)=\frac{1}{p}\mathbb{E}_{\xi,P_v(\mathbf{v}^\star)}\log\int_{\mathbb{R}^p}d\mathbf{v}\,P_v(\mathbf{v})\prod_{i=1}^{p}e^{-\frac{q_u}{2\Delta}v_i^2+\left(\frac{q_u}{\Delta}v_i^\star+\sqrt{\frac{q_u}{\Delta}}\xi\right)v_\mu}\qquad(2.31)$$

## 2.3 Application to generative priors

**Generalised linear model prior**

The expression we derived for the mutual information density in the $\mathbf{v}\mathbf{v}^\intercal$ model is valid for any prior $P_v$ as long as $\Psi_v$ is well defined in the thermodynamic limit. For the specific case when

$$P_v(\mathbf{v})=\int_{\mathbb{R}^k}\left(\prod_{l=1}^{k}dz_l\,P_z(z_l)\right)\prod_{i=1}^{p}P_{\text{out}}\left(v_i\Big|\frac{1}{\sqrt{k}}\sum_{l=1}^{k}W_{il}z_l\right),\qquad(2.32)$$

with $W_{il}\underset{\text{i.i.d.}}{\sim}\mathcal{N}(0,1)$, $\Psi_v$ is, up to a global $1/\alpha$ scaling, the Bayes-optimal free energy of a generalised linear model with channel given by

$$\tilde{P}_{\text{out}}\left(v|x;\xi,q_v\right)=P_{\text{out}}(v|x)e^{-\frac{q_v}{2\Delta}v^2+\sqrt{\frac{q_v}{\Delta}}\xi v},\qquad(2.33)$$

and factorised prior $P_z$. The expression for this free energy is well known - see for example [3] for a derivation and [3] for a proof - and reads

$$\lim_{p\to\infty}\Psi_v=\frac{1}{\alpha}\underset{q_z,\hat{q}_z}{\mathbf{extr}}\left[-\frac{1}{2}q_z\hat{q}_z+\alpha\Psi_{\text{out}}\left(\frac{q_v}{\Delta},q_z\right)+\Psi_z\left(\hat{q}_z\right)\right]\qquad(2.34)$$

where the functions $\Psi_{\text{out}}$ and $\Psi_z$ are defined in eq. (1.3). Inserting this expression in our general formula for the mutual information density eq. (2.24) give us

$$\lim_{p\to\infty}i_p=\frac{\rho_v}{4\Delta}+\underset{q_v,q_z,\hat{q}_z}{\mathbf{extr}}\left[\frac{1}{4\Delta}q_v^2+\frac{1}{2\alpha}\hat{q}_zq_z-\Psi_{\text{out}}\left(\frac{q_v}{\Delta},q_z\right)-\frac{1}{\alpha}\Psi_z(\hat{q}_z)\right]\qquad(2.35)$$

which is precisely the result from eq. (7). The extremisation problem in eq. (2.35) is solved by looking for the directions $(q_v,\hat{q}_z,q_z)$ of zero gradient of the potential $\Psi_v$. These saddle-point equations are known in this context as *state evolution equations*, and they can be conveniently written in terms of the auxiliary function we defined in Section 1.3, equations (1.16-1.23) as

$$q_v=2\partial_{q_v}\Psi_{\text{out}}\left(\frac{q_v}{\Delta},q_z\right)=\mathbb{E}_{\xi,\eta}\left[\mathcal{Z}_{\text{out}}\left(\sqrt{\frac{q_v}{\Delta}}\xi,\frac{q_v}{\Delta},\sqrt{q_z}\eta,\rho_z-q_z\right)f_v\left(\sqrt{\frac{q_v}{\Delta}}\xi,\frac{q_v}{\Delta},\sqrt{q_z}\eta,\rho_z-q_z\right)^2\right]$$

$$\hat{q}_z=2\alpha\partial_{q_z}\Psi_{\text{out}}\left(\frac{q_v}{\Delta},q_z\right)=\mathbb{E}_{\xi,\eta}\left[\mathcal{Z}_{\text{out}}\left(\sqrt{\frac{q_v}{\Delta}}\xi,\frac{q_v}{\Delta},\sqrt{q_z}\eta,\rho_z-q_z\right)f_{\text{out}}\left(\sqrt{\frac{q_v}{\Delta}}\xi,\frac{q_v}{\Delta},\sqrt{q_z}\eta,\rho_z-q_z\right)^2\right]$$

$$q_z=2\partial_{\hat{q}_z}\Psi_z\left(\hat{q}_z\right)=\mathbb{E}_\xi\left[\mathcal{Z}_z\left(\sqrt{\hat{q}_z}\xi,\hat{q}_z\right)f_z\left(\sqrt{\hat{q}_z}\xi,\hat{q}_z\right)^2\right]\qquad(2.36)$$

**Multi-layer prior**

The multi-layer prior can be conveniently written as

$$P_v(\mathbf{v})=\int\prod_{l=1}^{L}\prod_{\nu_l=1}^{k_l}dh_{\nu_l}^{(l)}P_{\text{out}}^{(l-1)}\left(h_{\nu_l}^{(l)}\Big|\frac{1}{\sqrt{k_{l-1}}}\sum_{\nu_{l-1}=1}^{k_{l-1}}W_{\nu_l\nu_{l-1}}^{(l-1)}h_{\nu_{l-1}}\right)\prod_{i=1}^{p}P_{\text{out}}^{(L)}\left(v_i\Big|\frac{1}{\sqrt{k_L}}\sum_{\nu_L=1}^{k_L}W_{i\nu_L}h_L\right),$$

$$(2.37)$$

where we define $\mathbf{h}^{(1)} \equiv \mathbf{z} \in \mathbb{R}^{k_1}$ and $P_{\text{out}}^{(0)} \equiv P_z$. As in the single-layer case, the Bayes-optimal free energy of $P_v$ has been computed in [4], and in our notation it is written as

$$\lim_{p \to \infty} \Psi_v = \frac{1}{\alpha} \underset{\{\hat{q}_l, q_l\}_{1 \le l \le L}}{\text{extr}} \left[ -\frac{1}{2} \sum_{l=1}^{L} \alpha_l \hat{q}_l q_l + \alpha \Psi_{\text{out}} \left( \frac{q_v}{\Delta}, q_L \right) + \sum_{l=2}^{L} \alpha_l \Psi_{\text{out}} \left( \hat{q}_l, q_{l-1} \right) + \Psi_z \left( \hat{q}_1 \right) \right],$$

(2.38)

where in this case $\alpha = p/k_1$ and we defined $\alpha_l = k_l/k_1$ for $1 \le l \le L$ (note in particular that $\alpha_1 = 1$). The $(\hat{q}_l, q_l)$ are the overlaps of the hidden variables $\mathbf{h}^{(l)}$ at each layer, and to be consistent with the shorthand notation introduced we have $(\hat{q}_1, q_1) = (\hat{q}_z, q_z)$. Inserting this expression in our general formula for the mutual information density eq. (2.24):

$$\lim_{p \to \infty} i_p = \frac{\rho_v}{4\Delta} +$$

$$\underset{q_v, \{\hat{q}_l, q_l\}_l}{\text{extr}} \left[ \frac{1}{4\Delta} q_v^2 + \frac{1}{2\alpha} \sum_{l=1}^{L} \alpha_l \hat{q}_l q_l - \frac{1}{\alpha} \sum_{l=2}^{L} \alpha_l \Psi_{\text{out}} \left( \hat{q}_l, q_{l-1} \right) - \Psi_{\text{out}} \left( \frac{q_v}{\Delta}, q_L \right) - \frac{1}{\alpha} \Psi_z \left( \hat{q}_1 \right) \right].$$

(2.39)

Taking the extremization over $q_v$ and $(\hat{q}_l, q_l)_{1 \le l \le L}$ in eq. (2.39), we obtain the following system of coupled fixed point equations:

$$
\begin{cases}
q_v = \Lambda_x \left( \frac{q_v}{\Delta}, q_L \right) \\
q_L = \Lambda_x \left( \hat{q}_L, q_{L-1} \right) \\
\quad \vdots \\
q_l = \Lambda_x \left( \hat{q}_l, q_{l-1} \right) \\
\quad \vdots \\
q_z = \Lambda_z \left( \hat{q}_z \right)
\end{cases}
\qquad
\begin{cases}
\hat{q}_L = \tilde{\alpha}_L \Lambda_{\text{out}} \left( \frac{q_v}{\Delta}, q_L \right) \\
\hat{q}_{L-1} = \tilde{\alpha}_{L-1} \Lambda_{\text{out}} \left( \hat{q}_L, q_{L-1} \right) \\
\quad \vdots \\
\hat{q}_l = \tilde{\alpha}_l \Lambda_{\text{out}} \left( \hat{q}_{l+1}, q_l \right) \\
\quad \vdots \\
\hat{q}_z = \tilde{\alpha}_1 \Lambda_{\text{out}} \left( \hat{q}_2, q_z \right)
\end{cases}
$$

(2.40)

where we have defined the update functions $\Lambda_x(x, y) \equiv \partial_x \Psi_{\text{out}}(x, y)$ and $\Lambda_{\text{out}}(x, y) \equiv \partial_y \Psi_{\text{out}}(x, y)$ and the layer-wise aspect ratios $\tilde{\alpha}_l = k_{l+1}/k_l = \alpha_{l+1}/\alpha_l$.

An important first question that can be answered from eqs. (2.40) is when does the Bayes-optimal estimator performs better than a random guess from the prior distribution $P_v$. For instance, we intuitively expect that when the prior is not biased towards a particular direction in $\mathbb{R}^p$ and for very high noise $\Delta \gg 1$ better-than-random estimation is not possible. In terms of fixed points of eqs. (2.40), this situation corresponds to the existence of the *non-informative* fixed point $q_v^\star = 0$ (i.e. maximum $\text{MSE}_v = \rho_v$, or zero overlap with the spike). Evaluating the right-hand side of eqs. (2.40) at $q_v = 0$, we can see that $q_v^\star = 0$ is a fixed point if

$$\mathbb{E}_{P_z}[z] = 0 \quad \text{and} \quad \mathbb{E}_{Q_{\text{out}}^{(l),0}}[v] = 0,$$

(2.41)

where $Q_{\text{out}}^{(l),0}(v, x) \equiv Q_{\text{out}}^{(l)}(v, x; 0, 0, 0, \rho_l)$ from eq. (10). Note that for multi-layer network with deterministic channels and $\varphi^{(l)} \equiv \varphi$ for all $l$, the second condition is equivalent to $\varphi$ being an odd function.

When the condition (2.41) holds, $(q_v, q_L, \hat{q}_L, \ldots, \hat{q}_z, q_z) = (0, 0, 0, \ldots, 0, 0)$ is a fixed point of eq. (2.40). The numerical stability of this fixed point determines a phase transition point $\Delta_c$, defined as the noise below which the fixed point $\mathbf{0} \in \mathbb{R}^{L+1}$ becomes unstable. The transition will then correspond to the value of $\Delta$ for which the largest eigenvalue of the Jacobian of the eqs. (2.40) at 0

becomes greater than one. This Jacobian is given explicitly by

$$
\begin{array}{c}
\begin{array}{ccccccccccccc}
q_v & \hat{q}_L & q_L & \hat{q}_{L-1} & q_{L-1} & \cdots & \hat{q}_{l+1} & q_{l+1} & \hat{q}_l & q_l & \cdots & \hat{q}_z & q_z
\end{array}\\
\left(
\begin{array}{ccccccccccccc}
\frac{1}{\Delta}m_{vv}^{(L)} & 0 & \frac{1}{\rho_L^2}m_{vx}^{(L)} & 0 & 0 & \cdots & 0 & 0 & 0 & 0 & \cdots & 0 & 0 \\
\frac{\tilde{\alpha}_L}{\Delta}m_{vx}^{(L)} & 0 & \frac{\tilde{\alpha}_L}{\rho_L^2}m_{xx}^{(L)} & 0 & 0 & \cdots & 0 & 0 & 0 & 0 & \cdots & 0 & 0 \\
0 & m_{vv}^{(L-1)} & 0 & 0 & \frac{1}{\rho_{L-1}^2}m_{vx}^{(L-1)} & \cdots & 0 & 0 & 0 & 0 & \cdots & 0 & 0 \\
0 & \tilde{\alpha}_{L-1}m_{vx}^{(L-1)} & 0 & 0 & \frac{\tilde{\alpha}_{L-1}}{\rho_{L-1}^2}m_{xx}^{(L-1)} & \cdots & 0 & 0 & 0 & 0 & \cdots & 0 & 0 \\
0 & 0 & 0 & m_{vv}^{(L-2)} & 0 & \cdots & 0 & 0 & 0 & 0 & \cdots & 0 & 0 \\
0 & 0 & 0 & \tilde{\alpha}_{L-2}m_{vx}^{(L-2)} & 0 & \cdots & 0 & 0 & 0 & 0 & \cdots & 0 & 0 \\
 & & \vdots & & & \vdots & & \vdots & & \vdots & & \vdots & \\
0 & 0 & 0 & 0 & 0 & \cdots & m_{vv}^{(l)} & 0 & 0 & \frac{1}{\rho_l^2}m_{vx}^{(l)} & \cdots & 0 & 0 \\
0 & 0 & 0 & 0 & 0 & \cdots & \tilde{\alpha}_l m_{vx}^{(l)} & 0 & 0 & \frac{\tilde{\alpha}_l}{\rho_l^2}m_{xx}^{(l)} & \cdots & 0 & 0 \\
 & & \vdots & & & \vdots & & \vdots & & \vdots & & & \\
0 & 0 & 0 & 0 & 0 & \cdots & 0 & 0 & 0 & 0 & \cdots & m_{zz} & 0
\end{array}
\right)
\begin{array}{c}
q_v \\ \hat{q}_L \\ q_L \\ \hat{q}_{L-1} \\ q_{L-1} \\ \hat{q}_{L-2} \\ \vdots \\ q_{l+1} \\ \hat{q}_l \\ \vdots \\ q_z
\end{array}
\end{array}
$$

$$(2.42)$$

where we have defined the following shorthand for the second moments of $Q_{\text{out}}^{(l),0}(v,x)$:

$$
m_{vv}^{(l)} = \left(\mathbb{E}_{Q_{\text{out}}^{(l),0}}v^2\right)^2, \quad m_{vx}^{(l)} = \left(\mathbb{E}_{Q_{\text{out}}^{(l),0}}vx\right)^2, \quad m_{xx}^{(l)} = \left(\mathbb{E}_{Q_{\text{out}}^{(l),0}}x^2 - \rho_l\right)^2, \quad m_{zz} = \left(\mathbb{E}_{P_z}z^2\right)^2
$$

$$(2.43)$$

This result is given in full generality, and it is instructive to compute $\Delta_c$ in specific cases. Consider $P_z(z) = \mathcal{N}_z(0,1)$ and layer-wise constant activation $P_{\text{out}}^{(l)}(v|x) = \delta(v - \varphi(x))$. For the previous odd activation functions discussed, we find that

**Linear activation:** For $\varphi(x) = x$ the leading eigenvalue of the Jacobian becomes one at

$$
\Delta_c = 1 + \sum_{l=1}^{L} \frac{\alpha}{\alpha_l}.
$$

$$(2.44)$$

Note in particular that for $L = 1$ and in the limit $\alpha = 0$ we recover the phase transition $\Delta_c = 1$ known from the case with separable prior [1]. For $\alpha > 0$, we have $\Delta_c > 1$ meaning the spike can be estimated more efficiently when its structure is accounted for. In particular, the deeper the generative network for the spike, the easier estimation becomes.

**Sign activation:** For $\varphi(x) = \text{sgn}(x)$ the leading eigenvalue of the Jacobian becomes one at

$$
\Delta_c = 1 + \sum_{l=1}^{L} \left(\frac{4}{\pi^2}\right)^l \frac{\alpha}{\alpha_l}.
$$

$$(2.45)$$

For $L = 1$ and $\alpha = 0$, $P_v = \text{Bern}(1/2)$, and the transition $\Delta_c = 1$ agrees with the one found for a separable prior distribution [1]. As in the linear case, for $\alpha > 0$, we can estimate the spike for larger values of noise than in the separable case, and depth also improves estimation.

# 3 Proof of the mutual information for the $\mathbf{v}\mathbf{v}^\intercal$ case

In this section, we present a proof of the theorem 1 in the main part, for the mutual information of Wigner model eq. (1.1) with structured prior

$$Y = \frac{1}{\sqrt{p}}\mathbf{v}^\star\mathbf{v}^{\star\intercal} + \sqrt{\Delta}\xi\,, \tag{3.1}$$

where the spike $\mathbf{v}^\star \in \mathbb{R}^p$ is drawn from $P_v$.

## 3.1 Notations, free energies, and Gibbs average

The mutual information being invariant to reparametrization, we shall work instead inside this section with the following notations:

$$Y = \sqrt{\frac{\lambda}{p}}\mathbf{v}^\star\mathbf{v}^{\star\intercal} + \xi\,, \tag{3.2}$$

where $\lambda$ is the signal to noise ratio. Up to the reparametrization, it corresponds to our model with $\lambda = \Delta^{-1}$. We thus aim to compute $\frac{I(Y;\mathbf{v})}{p}$ for this model.

While the information theoretic notation is convenient in stating the theorem, it is more convinient to use statistical physics notation and "free energies" for the proof, that relies heavily on concepts from mathematical physics. Let us first translate one into the other. The mutual information between the observation $Y$ and the unknown $\mathbf{v}$ is defined using the entropy as $I(Y;\mathbf{v}) = H(Y) - H(Y|\mathbf{v})$. Using Bayes theorem one obtains $H(Y) = \mathbb{E}_Y\{\log \mathbb{E}_{P_v} P_Y(Y|\mathbf{v})\}$ and a straightforward computation shows that the mutual information per variable is then expressed as

$$\frac{I(Y;\mathbf{v})}{p} = f_p + \lambda\frac{\mathbb{E}[\mathbf{v}^\intercal\mathbf{v}]}{4p}\,, \tag{3.3}$$

where, using again statistical physics terms, $f_p = -E_Y\left[\log \mathcal{Z}_p(Y)\right]/p$ is the so called free energy density and $\mathcal{Z}_p(Y)$ the partition function defined by

$$\mathcal{Z}_p(Y) \equiv \int_{\mathbb{R}^p} \mathrm{d}\mathbf{v}\, P_v(\mathbf{v}) \exp\left(\sum_{i<j}\left(-\lambda\frac{v_i^2 v_j^2}{2p} + \sqrt{\lambda}\frac{v_i v_j Y_{ij}}{\sqrt{p}}\right)\right)\,. \tag{3.4}$$

Correspondingly, we define the Hamiltonian:

$$-H(\mathbf{v}) \equiv \sum_{i<j}\sqrt{\frac{\lambda}{p}}Y_{ij}v_i v_j - \frac{\lambda}{2p}v_i^2 v_j^2 = \sum_{i<j}\sqrt{\frac{\lambda}{p}}\xi_{ij}v_i v_j + \frac{\lambda}{p}v_i v_j v_i^\star v_j^\star - \frac{\lambda}{2p}v_i^2 v_j^2.$$

so that the partition function (3.4) is associated with the Gibbs-Boltzmann measure $e^{-H}/\mathcal{Z}_p(Y)$.

Consider now the term $I\left(\mathbf{v};\mathbf{v} + \mathbf{z}/\sqrt{q_v\lambda}\right)$ that enters the expression to be proven eq. (6). This is the mutual information for another denoising problem, in which we assume one observes a noisy version of the vector $\mathbf{v}^\star$, denoted $\tilde{\mathbf{y}}$ such that

$$\tilde{\mathbf{y}} = \frac{1}{\sigma}\mathbf{v}^* + \mathbf{z}, \tag{3.5}$$

where $\mathbf{z} \sim \mathcal{N}(\mathbf{0}_p, \mathrm{I}_p)$ and $\sigma = 1/\sqrt{q_v\lambda}$, where we shall assume that the limit exists. Again, it is easier to work with free energies. We thus write the corresponding posterior distribution as

$$P(\mathbf{v}|\tilde{\mathbf{y}}) = \frac{1}{\mathcal{Z}_0(\tilde{\mathbf{y}},\sigma)}P_v(\mathbf{v})\exp\left(-\frac{\|\mathbf{v}\|_2^2}{2\sigma^2} + \frac{\mathbf{v}^\intercal\tilde{\mathbf{y}}}{\sigma}\right)\,, \tag{3.6}$$

where $\mathcal{Z}_0(\tilde{\mathbf{y}})$ is the normalization factor. For this denoising problem, the averaged free energy per variables reads

$$f_p^0(\sigma) \equiv -\frac{1}{p}\mathbb{E}_{\tilde{\mathbf{y}}}[\log \mathcal{Z}_0(\tilde{\mathbf{y}},\sigma)], \tag{3.7}$$

and a short computation shows that

$$I\left(\mathbf{v}; \mathbf{v} + \frac{1}{\sqrt{q_v \lambda}} \mathbf{z}\right) = f_p^0 \left(\frac{1}{\sqrt{\lambda q_v}}\right) + \frac{\rho_v \lambda q_v}{2}$$

Putting all the pieces together, this means that we need to prove the following statement on the free energy $f_p$: the free energy $f_p = -\mathbb{E}_Y \left[\log \mathcal{Z}_p(Y)\right]/p$ is given, as $p \to \infty$ by

$$\lim_{p \to \infty} f_p = \min \phi_{\mathrm{RS}} \left(\frac{1}{\sqrt{q_v \lambda}}\right) \text{ with } \phi_{\mathrm{RS}}(r) \equiv \lim_{p \to \infty} f_p^0(r) + \frac{\lambda q_v^2}{4}. \qquad (3.8)$$

This statement is equivalent to theorem 1, and we shall present a proof for the case where the prior over $\mathbf{v}$ has a "good" limit: we shall assume that the limiting free energy exists and concentrates over the disorder, and that the distribution over each $v_i$ is bounded. To do so, it will be useful to consider Gibbs averages, and to work with $r$ copies of the same system. For any $g : (\mathbb{R}^p)^{r+1} \mapsto \mathbb{R}$, we define the Gibbs average as

$$\left\langle g(\mathbf{v}^{(1)}, \cdots, \mathbf{v}^{(r)}, \mathbf{v}^\star) \right\rangle \equiv \frac{\int g(\mathbf{v}^{(1)}, \cdots, \mathbf{v}^{(r)}, \mathbf{v}^\star) \prod_{l=1}^r e^{-H(\mathbf{v}^{(l)})} \mathrm{d}P_v(\mathbf{v}^{(l)})}{\left(\int e^{-H(\mathbf{v}^{(l)})} \mathrm{d}P_v(\mathbf{v}^{(l)})\right)^r}. \qquad (3.9)$$

This is the average of $g$ with respect to the posterior distribution of $r$ copies $\mathbf{v}^{(1)}, \cdots, \mathbf{v}^{(r)}$ of $\mathbf{v}^\star$. The variables $\{\mathbf{v}^l\}_{l=1\ldots r}$ are called *replicas*, and are interpreted as random variables independently drawn from the posterior. When $r = 1$ we simply write $g(\mathbf{v}, \mathbf{v}^\star)$ instead of $g(\mathbf{v}^{(1)}, \mathbf{v}^\star)$. Finally we shall denote the overlaps between two replicas as follows: for $l, l' = 1\ldots r$, we let

$$R_{l,l'} \equiv \mathbf{v}^{(l)} \cdot \mathbf{v}^{(l')} = \frac{1}{p} \sum_{i=1}^p v_i^{(l)} v_i^{(l')}. \qquad (3.10)$$

A simple but useful consequence of Bayes rule is that the $(r+1)$-tuples $(\mathbf{v}^{(1)}, \cdots, \mathbf{v}^{(r+1)})$ and $(\mathbf{v}^{(1)}, \ldots, \mathbf{v}^{(r)}, \mathbf{v}^*)$ are the same law under $\mathbb{E}\langle \cdot \rangle$ (see [5] or proposition 16 in [6]). This bears the name of *the Nishimori property* in the spin glass literature [2].

## 3.2 Guerra Interpolation for the upper bound

We start by using the Guerra interpolation to prove an exact formula for the free energy. Here the proof is a *verbatim* reproduction of the argument of [5] for non-factorized prior based on Guerra's interpolation [7].

Let $t \in [0, 1]$ and let $q_v$ be a non-negative variable. We now consider an interpolating Hamiltonian

$$-H_t(\mathbf{v}) \equiv \sum_{i<j} \sqrt{\frac{t\lambda}{p}} \xi_{ij} v_i v_j + \frac{t\lambda}{p} v_i v_j^\star v_j v_j^\star - \frac{t\lambda}{2p} v_i^2 v_j^2$$

$$+ \sum_{i=1}^p \sqrt{(1-t)\lambda q_v} z_i v_i + (1-t)\lambda q_v v_i v_i^\star - \frac{(1-t)\lambda q_v}{2} v_i^2.$$

The Gibbs states associated with this Hamiltonian $-H_t$ correspond to an estimation problem given an augmented set of observations

$$\begin{cases} Y_{ij} = \sqrt{\frac{t\lambda}{p}} v_i^\star v_j^\star + \xi_{ij}, & 1 \le i \le j \le p, \\ \tilde{y}_i = \sqrt{(1-t)\lambda q_v} v_i^\star + z_i, & 1 \le i \le p. \end{cases}$$

Reproducing the argument of [5], we prove using Guerra's interpolation [7] and the Nishimori property that

$$f_p \le \phi_{\mathrm{RS}}(\lambda) + \frac{K}{p}. \qquad (3.11)$$

for some constant $K$. We define

$$\varphi(t) \equiv -\frac{1}{p} \mathbb{E} \log \int e^{-H_t(\mathbf{v})} \mathrm{d}P_v(\mathbf{v}). \qquad (3.12)$$

A simple calculation based on Gaussian integration by parts (in technical terms, Stein's lemma) shows that

$$\varphi'(t) = \frac{\lambda}{4}\mathbb{E}\left\langle (R_{1,2} - q_v)^2 \right\rangle_t - \frac{\lambda}{4}q_v^2 - \frac{\lambda}{4p^2}\sum_{i=1}^{p}\mathbb{E}\left\langle v_i^{(1)\,2}v_i^{(2)\,2} \right\rangle_t$$
$$- \frac{\lambda}{2}\mathbb{E}\left\langle (R_{1,*} - q_v)^2 \right\rangle_t + \frac{\lambda}{2}q_v^2 + \frac{\lambda}{2p^2}\sum_{i=1}^{p}\mathbb{E}\left\langle v_i^{\,2}v_i^{*\,2} \right\rangle_t,$$

We now use the Nishimori property, and the expressions involving the pairs $(\mathbf{v}, \mathbf{v}^\star)$ and $(\mathbf{v}^{(1)}, \mathbf{v}^{(2)})$ become equal. We thus obtain

$$\varphi'(t) = -\frac{\lambda}{4}\mathbb{E}\left\langle (R_{1,*} - q_v)^2 \right\rangle_t + \frac{\lambda}{4}q_v^2 + \frac{\lambda}{4p^2}\sum_{i=1}^{p}\mathbb{E}\left\langle v_i^{\,2}v_i^{*\,2} \right\rangle_t. \tag{3.13}$$

Observe that the last term is $\mathcal{O}\left(1/p\right)$ since the variables $v_i$ are bounded. Moreover, the first term is always non-negative so we obtain

$$\varphi'(t) \leq \frac{\lambda}{4}q_v^2 + \frac{K}{p}. \tag{3.14}$$

Since $\varphi(1) = f_p$ and $\varphi(0) = f_p^0(1/\sqrt{\lambda q_v})$, integrating over $t$, we obtain for all $q_v \geq 0$, $f_p \leq \phi_{\mathrm{RS}}(\lambda, q_v) + \frac{K}{p}$, and this yields the upper bound:

**Proposition 3.1** (Upper bound on the Free energy). *: There exists $K > 0$ such that for all $q_v \in \mathbb{R}$ we have*

$$f_p \leq f_p^0(1/\sqrt{\lambda q_v}) + \frac{\lambda q_v^2}{4} + \frac{K}{p}. \tag{3.15}$$

### 3.3 A bound of the Franz-Parisi Potential

To attack the lower bound, we shall adapt the argument of [8], that uses the Franz-Parisi potential [9], and this will require additional concentration properties on the prior model. For $\mathbf{v}^\star \in \mathbb{R}^p$ fixed, $m \in \mathbb{R}$ and $\epsilon > 0$ we follow [8] and define

$$\Phi_\epsilon^p(m, \mathbf{v}^\star) \equiv -\frac{1}{p}\mathbb{E}\log\int_{\mathbb{R}^p} \mathbb{1}\{R_{1,*} \in [m, m+\epsilon]\}e^{-H(\mathbf{v})}\mathrm{d}P_v(\mathbf{v}). \tag{3.16}$$

This is simply the free energy with configurations forced to be at a distance $m$ (to precision $\epsilon$) from the ground truth. Note that since the measure is limited to a subset of configurations, it is clear that $\mathbb{E}_{\mathbf{v}^\star}\Phi_\epsilon^p(m, \mathbf{v}^\star) \geq f_p$.

We are now going to prove an interpolating bound for the Franz-Parisi Potential. Let $t \in [0, 1]$ and consider a slightly different interpolating Hamiltonian

$$-H_t(\mathbf{v}) \equiv \sum_{i<j}\sqrt{\frac{t\lambda}{p}}\xi_{ij}v_iv_j + \frac{t\lambda}{p}v_iv_i^\star v_jv_j^\star - \frac{t\lambda}{2p}v_i^2v_j^2$$
$$+ \sum_{i=1}^{p}\sqrt{(1-t)\lambda q_v}z_iv_i + (1-t)\lambda mv_iv_i^\star - \frac{(1-t)\lambda q_v}{2}v_i^2,$$

Notice the subtle change: in front of the term $(1-t)v_iv_i^\star$ we replace the $q_v$ from the former section by $m$. We define now

$$\varphi_{\epsilon,m}(t) \equiv -\frac{1}{p}\mathbb{E}\log\int_{\mathbb{R}^p}e^{-H_t(\mathbf{v})}\mathbb{1}\{R_{1,*} \in [m, m+\epsilon]\}\mathrm{d}P_v(\mathbf{v}). \tag{3.17}$$

Denoting now the Gibbs average with the additional constraint $\mathbb{1}\{R_{1,*} \in [m, m+\epsilon]\}$ as $\langle\rangle_t^{m,\epsilon}$, we find when we repeat the former computation:

$$\varphi'_{\epsilon,m}(t) = \frac{\lambda}{4}\mathbb{E}\left\langle (R_{1,2} - q_v)^2 \right\rangle_t^{m,\epsilon} - \frac{\lambda}{4}q_v^2 + \frac{\lambda}{2}m^2 - \frac{\lambda}{2}\mathbb{E}\left\langle (R_{1,*} - m)^2 \right\rangle_t^{m,\epsilon} + o\,(1)$$

The trick is now to notice that, by construction, the $\mathbb{E}\left\langle (R_{1,*} - m)^2 \right\rangle_t^{m,\epsilon} \leq \epsilon^2$ given the overlap restriction, and therefore

$$\varphi'_{\epsilon,m}(t) \geq \frac{\lambda}{4}\mathbb{E}\left\langle (R_{1,2} - q_v)^2 \right\rangle_t^{m,\epsilon} - \frac{\lambda}{4}q_v^2 + \frac{\lambda}{2}m^2 - \frac{\lambda\epsilon^2}{2} + o(1),$$

and

$$\varphi'_{\epsilon,m}(t) \geq -\frac{\lambda}{4}q_v^2 + \frac{\lambda}{2}m^2 - \frac{\lambda\epsilon^2}{2} + o(1).$$

We now denote

$$f_p^0(\sigma, \mathbf{v}^\star) \equiv -\frac{1}{N}\mathbb{E}_z[\log \mathcal{Z}_0(\tilde{\mathbf{y}}, \sigma)], \tag{3.18}$$

with the previous $f_p^0$ being the expectation $f_p^0(\sigma) \equiv \mathbb{E}_{\mathbf{v}^\star}[f_p^0(\sigma, \mathbf{v}^\star)]$. Then, since $\varphi_{\epsilon,m}(1) = \Phi_\epsilon^p(m, \mathbf{v}^\star)$ and $\varphi_{\epsilon,m}(0) \leq f_p^0(1/\sqrt{\lambda q_v})$ (again, this is an obvious consequence of the restriction in the sum) integrating over $t$, we obtain a bound for the Parisi-Franz potential for any $q_v$ and $m$. Using, in particular, the value $m = q_v$, this yields yields the following result:

**Proposition 3.2** (Lower bound on the Franz-Parisi potential). *: There exists $K > 0$ such that for any $m = q_v$ and $\epsilon > 0$ we have*

$$\Phi_\epsilon^p(m = q_v, \mathbf{v}^\star) \geq f_p^0\left(1/\sqrt{\lambda q_v}, \mathbf{v}^\star\right) + \frac{\lambda q_v^2}{4} - \frac{\lambda}{2}\epsilon^2 + \frac{K}{p}. \tag{3.19}$$

### 3.4 From the Potential to a Lower bound on the free energy

It remains to connect the Franz-Parisi potential to the actual free energy. This is done by proving a Laplace-like result between the free energy and the Franz-Parisi free energy, again following the separable case in [8]:

**Proposition 3.3.** *There exists $K > 0$ such that for all $\epsilon > 0$, we have*

$$f_p \geq \mathbb{E}_{\mathbf{v}^\star}\left[\min_{l \in \mathcal{Z}, |l| \leq K/\epsilon} \Phi_\epsilon^p(l\epsilon, \mathbf{v}^\star)\right] - \frac{\log(K/\epsilon)}{\sqrt{p}}. \tag{3.20}$$

Combining this proposition with the bound on the Franz-Parisi potential, we see that

$$f_p \geq \mathbb{E}_{\mathbf{v}^\star}\left[\min_{\substack{q_v = l\epsilon \\ |l| \leq K/\epsilon}} f_p^0\left(1/\sqrt{\lambda q_v}, \mathbf{v}^\star\right) + \frac{\lambda q_v^2}{4}\right] - \frac{\lambda}{2}\epsilon^2 - \frac{\log(K/\epsilon)}{\sqrt{p}}. \tag{3.21}$$

At this point, we need to push the expectation with respect to the spike *inside* the minimum. This is the *only assumption* that we are going to require over the generative model: *that its free energy concentrates over the distribution of spikes*. This finally leads to following result:

**Proposition 3.4** (Laplace principle). *Assume that the free energy $f_p^0(\mathbf{v}^\star)$ concentrates such that* $\mathbb{E}\left[\left|f_p^0(\frac{1}{\sqrt{\lambda q_v}}, \mathbf{v}^\star) - \mathbb{E}\left[f_p^0(\frac{1}{\sqrt{\lambda q_v}}, \mathbf{v}^\star)\right]\right|\right] < C/\sqrt{p}$ *for some constant $C$ for all $q_v$ in $[0, \rho_v)$, then:*

$$f_p \geq \min_{q_v}\left[f_p^0\left(1/\sqrt{\lambda q_v}\right) + \frac{\lambda q_v^2}{4}\right] + o\left(\frac{\log p}{\sqrt{p}}\right). \tag{3.22}$$

which gives us the needed converse bound. To conclude this section, let us prove these propositions.
***Proof of Proposition 3.3.*** Let $\epsilon > 0$. Since the prior $P_v$ has bounded support, we can grid the set of the overlap values $R_{1,*}$ by $2K/\epsilon$ many intervals of size $\epsilon$ for some $K > 0$. This allows the following discretisation, where $l$ runs over the finite range $\{-K/\epsilon, \cdots, K/\epsilon\}$:

$$-f_p = \frac{1}{p}\mathbb{E}\log\sum_l \int_{\mathbb{R}^p} \mathbb{1}\{R_{1,*} \in [l\epsilon, (l+1)\epsilon)\}e^{-H(\mathbf{v})}\mathrm{d}P_v(\mathbf{v})$$

$$\leq \frac{1}{p}\mathbb{E}\log\frac{2K}{\epsilon}\max_l \int_{\mathbb{R}^p} \mathbb{1}\{R_{1,*} \in [l\epsilon, (l+1)\epsilon)\}e^{-H(\mathbf{v})}\mathrm{d}P_v(\mathbf{v})$$

$$= \frac{1}{p}\mathbb{E}\max_l \log\int_{\mathbb{R}^p} \mathbb{1}\{R_{1,*} \in [l\epsilon, (l+1)\epsilon)\}e^{-H(\mathbf{v})}\mathrm{d}P_v(\mathbf{v}) + \frac{\log(2K/\epsilon)}{p}. \tag{3.23}$$

Note that in the above, the expectation $\mathbb{E}$ is taken with respect to both the noise matrix $\xi$ and the spike $\mathbf{v}^\star$. We shall now use concentration of measure to push the expectation over $\xi$ to the other side of the maximum in order to recover the Franz-Parisi potential as defined in the previous section.

Let

$$Z_l \equiv \int_{\mathbb{R}^p} \mathbb{1}\{R_{1,*} \in [l\epsilon, (l+1)\epsilon)\} e^{-H(\mathbf{v})} dP_v(\mathbf{v}). \tag{3.24}$$

One can show that each term $X_l = \frac{1}{p} \log Z_l$ individually concentrates around its expectation with respect to the random variable $\xi$. This follows from the following lemma

**Lemma 3.1.** *[from [8]] There exists a constant $K > 0$ such that for all $\gamma \geq 0$ and all $l$,*

$$\mathbb{E}_\xi e^{\gamma(X_l - \mathbb{E}_\xi[X_l])} \leq \frac{K\gamma}{\sqrt{p}} e^{K\gamma^2/p}. \tag{3.25}$$

that is a direct consequence of the Tsirelson-Ibragimov-Sudakov inequality [10], see [8], Lemma 7.

Given that all $X_l$ concentrates, the expectation of the maximum concentrates as well:

$$\mathbb{E}_\xi \max_l (X_l - \mathbb{E}_\xi[X_l]) \leq \frac{1}{\gamma} \log \mathbb{E}_\xi \exp\left(\gamma \max_l (X_l - \mathbb{E}_\xi[X_l])\right)$$

$$= \frac{1}{\gamma} \log \mathbb{E}_\xi \max_l e^{\gamma(X_l - \mathbb{E}'[X_l])}$$

$$\leq \frac{1}{\gamma} \log \mathbb{E}_\xi \sum_l e^{\gamma(X_l - \mathbb{E}_\xi[X_l])}$$

$$\leq \frac{1}{\gamma} \log \left(\frac{2K}{\epsilon} \frac{\gamma K}{\sqrt{p}} e^{\gamma^2 K/p}\right)$$

$$= \frac{\log(2K/\epsilon)}{\gamma} + \frac{1}{\gamma} \log \frac{\gamma K}{\sqrt{p}} + \frac{\gamma K}{p}.$$

We set $\gamma = \sqrt{p}$ and obtain

$$\mathbb{E}_\xi \max_l (X_l - \mathbb{E}_\xi[X_l]) \leq \frac{\log(K/\epsilon)}{\sqrt{p}}. \tag{3.26}$$

Therefore, inserting the above estimates into (3.23), we obtain

$$-f_p \leq \mathbb{E}_{\mathbf{v}^\star} \max_l \mathbb{E}_\xi X_l + \frac{\log(K/\epsilon)}{\sqrt{p}} + \frac{\log(K/\epsilon)}{p} \leq \mathbb{E}_{\mathbf{v}^\star} \max_l \Phi_\epsilon(l\epsilon, \mathbf{v}^\star) + 2\frac{\log(K/\epsilon)}{\sqrt{p}}$$

so that finally

$$f_p \geq \mathbb{E}_{\mathbf{v}^\star} \min_l \Phi_\epsilon(l\epsilon, \mathbf{v}^\star) - \frac{\log(K/\epsilon)}{\sqrt{p}},$$

for some constant $K$. ∎

***Proof of Proposition 3.4.*** We now wish to push the expectation with respect to $\mathbf{v}^\star$ inside the minimum. We start by using again $q_v = l\epsilon$ and defining the following random (in $\mathbf{v}^\star$ variable):

$$\tilde{X}_l = -\left(f_p^0\left(1/\sqrt{\lambda l\epsilon}, \mathbf{v}^\star\right) + \frac{\lambda q_v^2}{4}\right) \tag{3.27}$$

and start from Proposition 3.3:

$$-f_p \leq \mathbb{E}_{\mathbf{v}^\star}\left[\max_{\substack{q_v = l\epsilon \\ |l| \leq K/\epsilon}} \tilde{X}_l\right] + \frac{\lambda}{2}\epsilon^2 + \frac{\log(K/\epsilon)}{\sqrt{p}}. \tag{3.28}$$

We now wish to push the max inside. We proceed as follow:

$$\mathbb{E}_{\mathbf{v}^\star}\left[|\max_l \left(\tilde{X}_l - \mathbb{E}[\tilde{X}_l]\right)|\right] \leq \mathbb{E}_{\mathbf{v}^\star}\left[\sum_l |\left(\tilde{X}_l - \mathbb{E}[\tilde{X}_l]\right)|\right] \tag{3.29}$$

$$= \sum_l \mathbb{E}_{\mathbf{v}^\star}\left[|\left(\tilde{X}_l - \mathbb{E}[\tilde{X}_l]\right)|\right] \tag{3.30}$$

$$\leq \sum_l \frac{C}{\sqrt{p}} = \frac{K}{\epsilon\sqrt{p}} \tag{3.31}$$

Inserting this in eq.(3.28) we find that

$$- f_p \leq \max_{\substack{q_v = l\epsilon \\ |l| \leq K/\epsilon}} \left[ \mathbb{E}_{\mathbf{v}^\star} \tilde{X}_l \right] + \frac{\lambda}{2} \epsilon^2 + \frac{K'}{\epsilon \sqrt{p}} + \frac{\log(K/\epsilon)}{\sqrt{p}}, \tag{3.32}$$

and therefore,

$$f_p \geq \min_{\substack{q_v = l\epsilon \\ |l| \leq K/\epsilon}} \left[ - \mathbb{E}_{\mathbf{v}^\star} \tilde{X}_l \right] - \frac{\lambda}{2} \epsilon^2 - \frac{K'}{\epsilon \sqrt{p}} - \frac{\log(K/\epsilon)}{\sqrt{p}}, \tag{3.33}$$

so that choosing finally $\epsilon = p^{-1/4}$ we reach

$$f_p \geq \min_{\substack{q_v = l\epsilon \\ |l| \leq K/\epsilon}} \left[ f_p^0 \left( 1/\sqrt{\lambda q_v} \right) + \frac{\lambda q_v^2}{4} \right] + o\left( \frac{\log p}{\sqrt{p}} \right). \tag{3.34}$$

∎

## 3.5 Main theorem

We can now combine the upper and lower bound to reach the statement of the main theorem, presented in the main as theorem 1:

**Theorem 3.1.** *[Mutual information and MMSE for the spiked Wigner model with structured spike] Assume the spikes $\mathbf{v}^\star$ come from a sequence (of growing dimension) of generic structured prior $P_v$ on $\mathbb{R}^p$, such that*

1. *The free energy $f_p^0(\lambda q_v) = -\frac{1}{N} \mathbb{E}_{\tilde{\mathbf{y}}}[\log \mathcal{Z}_0(\tilde{\mathbf{y}}, 1/\sqrt{\lambda q_v})]$ has a limit $f_0(\lambda q_v)$ for all $q_v \in [0, \rho_v]$ as $p \to \infty$.*

2. *The free energy $f_p^0(\mathbf{v}^\star)$ concentrates such that $\mathbb{E}\left[ |f_p^0(1/\sqrt{\lambda q_v}, \mathbf{v}^\star) - \mathbb{E}\left[ f_p^0(1/\sqrt{\lambda q_v}, \mathbf{v}^\star) \right] | \right] < C/\sqrt{p}$ for some constant $C$ for all $q_v \in [0, \rho_v]$ as $p \to \infty$:*

*then*

$$\lim_{p \to \infty} i_p \equiv \lim_{p \to \infty} \frac{I(Y; \mathbf{v}^\star)}{p} = \inf_{\rho_v \geq q_v \geq 0} i_{\mathrm{RS}}(\Delta, q_v), \tag{3.35}$$

*with*

$$i_{\mathrm{RS}}(\Delta, q_v) = \frac{(\rho_v - q_v)^2}{4\Delta} + \lim_{p \to \infty} \frac{I\left( \mathbf{v}; \mathbf{v} + \sqrt{\frac{\Delta}{q_v}} z \right)}{p} \tag{3.36}$$

*with $z$ being a Gaussian vector with zero mean, unit diagonal variance and $\rho_v = \lim_{p \to \infty} \mathbb{E}_{P_v}[\mathbf{v}^\mathsf{T} \mathbf{v}]/p$.*

## 3.6 Mean-squared errors

It remains to deduce the optimal mean squared errors from the mutual information. These are actually simple application of known results which we reproduce here briefly for completeness. It is instructive to distinguish between the reconstruction of the spike and the reconstruction of the rank-one matrix.

Let us first focus on the denoising problem, where one aim to reconstruct the rank-one matrix $X^\star = \mathbf{v}^\star \mathbf{v}^{\star\mathsf{T}}$. In this case the mean squared error between an estimate $\hat{X}(Y)$ and the hidden one $X^\star$ reads

$$\mathrm{Matrix} - \mathrm{mse}(\hat{X}, Y) = \frac{1}{p^2} \| \mathbf{v}^\star \mathbf{v}^{\star\mathsf{T}} - \hat{X}(Y) \|_2^2 \tag{3.37}$$

It is well-known [11] that the mean squared error is minimized by using the conditional expectation of the signal given the observation, that is the posterior mean. The minimal mean square error is thus given by

$$\mathrm{Matrix} - \mathrm{MMSE}(Y) = \frac{1}{p^2} \| \mathbf{v}^\star \mathbf{v}^{\star\mathsf{T}} - \mathbb{E}[\mathbf{v} \mathbf{v}^\mathsf{T} | Y] \|_F^2 \tag{3.38}$$

We can now state the result:

**Theorem 3.2.** *[Matrix MMSE, from [12, 13, 14]] The matrix-MMSE is asymptotically given by*

$$\lim_{p \to \infty} \mathrm{Matrix} - \mathrm{MMSE}(Y) = \rho_v^2 - (q_v^\star)^2 \tag{3.39}$$

*where $q_v^\star$ is the optimizer of the function $i_{\mathrm{RS}}(\Delta, q_v)$.*

*Proof.* This is a simple application of the I-MMSE theorem [15], that has been used in this context multiple-times (see e.g [12, 13, 14]). Indeed, the I-MMSE theorem states that, denoting $\lambda = \Delta^{-1}$:

$$\frac{d}{d\lambda}\frac{I}{p} = \frac{1}{4}\text{Matrix} - \text{MMSE}(Y) \tag{3.40}$$

We thus need to compute the derivative of the mutual information:

$$\frac{d}{d\lambda}i_{\text{RS}}(q_v^\star, \Delta = 1/\lambda) = \partial_\lambda i_{\text{RS}}(q_v^\star, \Delta = 1/\lambda) + \partial_{q_v} i_{\text{RS}}(q_v, \Delta = 1/\lambda)|_{q_v^\star}\partial_\lambda(q_v^\star) \tag{3.41}$$

$$= \partial_\lambda i_{\text{RS}}(q_v^\star, \Delta = 1/\lambda) \tag{3.42}$$

where we used $\partial_{q_v} i_{\text{RS}}(q, \Delta = 1/\lambda)|_{q_v^\star} = 0$. Denoting then $\mathcal{I}(\lambda_v, q_v) = \lim_{p \to \infty} \frac{I\left(\mathbf{v};\mathbf{v}+\sqrt{\frac{1}{\lambda q_v}}\mathbf{z}\right)}{p}$ we find

$$\partial_\lambda i_{\text{RS}}(q_v^\star, \Delta = 1/\lambda) = \frac{(\rho_v - q_v)^2}{4} + \partial_\lambda \mathcal{I}(\lambda_v, q_v)|_{q_v^\star} \tag{3.43}$$

We now use the fact that the derivate of the replica mutual information is zero at $q^\star$. This implies

$$\frac{\lambda}{2}(\rho_v - q_v^\star) = \partial_{q_v} \mathcal{I}(\lambda_v, q_v)|_{q_v^\star} = \frac{\lambda}{q_v^\star}\partial_\lambda \mathcal{I}(\lambda_v, q_v)|_{q_v^\star} \tag{3.44}$$

so that

$$\partial_\lambda i_{\text{RS}}(q_v^\star, \Delta = 1/\lambda) = \frac{(\rho_v - q_v)^2}{4} + \frac{1}{2}(\rho_v - q_v^\star)q_v^\star = \frac{1}{4}\left(\rho_v^2 - (q_v^\star)^2\right) \tag{3.45}$$

which proves the claim. ∎

We now consider the problem of reconstruction the spike itself. In this case the mean square error reads

$$\text{Vector} - \text{mse}(\hat{X}, Y) = \frac{1}{p}\|\mathbf{v} - \hat{\mathbf{v}}(Y)\|_2^2 \tag{3.46}$$

$$\text{Vector} - \text{MMSE}(Y) = \frac{1}{p}\|\mathbf{v} - \mathbb{E}[\mathbf{v}|Y]\|_2^2 \tag{3.47}$$

$$\tag{3.48}$$

Taking the square and averaging, we thus find that the asymptotic vector MMSE reads

$$\text{Vector} - \text{MMSE}(Y) = \rho_v + \frac{\|\mathbb{E}[\mathbf{v}|Y]\|_2^2}{p} - 2\frac{\mathbb{E}[\mathbf{v}^{\mathsf{T}}\mathbf{v}^\star|Y]}{p} = \rho_v - \frac{\mathbb{E}[\mathbf{v}^{\mathsf{T}}\mathbf{v}^\star|Y]}{p} \tag{3.49}$$

where we have use the Nishimori identity. In order to show that the MMSE is given by $\rho_v - q_v^\star$, we thus needs to show that $q_v^\star$ is indeed equal to $\frac{\mathbb{E}[\mathbf{v}^{\mathsf{T}}\mathbf{v}^\star|Y]}{p}$.

Fortunately, this is easy done by using Theorem 7 in [16], which apply in our case since it only depends on the free energy and the Franz-Parisi bound, that we have reproduced in the coupled cases in the present section. This proposition states the convergence in probability of the overlaps:

**Theorem 3.3** (Convergence in probability of the overlap, from [16]). *Informally, for the Wigner-Spikel model:*

$$\lim_{p \to \infty} \mathbb{E}\langle \mathbb{1}(|R_{1,*}| - q_v^\star| \geq \epsilon)\rangle \to 0 \tag{3.50}$$

*Note that the absolute value is necessary here, because if the prior is symmetric, is it impossible to distinguish between $\mathbf{v}^\star$ and $-\mathbf{v}^\star$. If the prior is not symmetric, then the absolute value can be removed.*

# 4 Heuristic derivation of AMP from the two simples AMP algorithms

In this section we present the derivation of the AMP algorithm described in sec. 3 of the main part. The idea is to simplify the Belief Propagation (BP) equations by expanding them in the large $n, p, k$ limits. Together with a Gaussian ansatz for the distribution of BP messages, this yields a set of $\mathcal{O}\left(k^2\right)$ simplified equations known as *relaxed BP* (rBP) equations. The last step to get the AMP algorithm is to remove the target dependency of the messages that further reduces the number of iterative equations to $\mathcal{O}\left(k\right)$.

Our derivation is closely related to the derivation of AMP for a series of statistical inference problems with factorised priors, see for example [1] and references therein. In the interest of the reader, instead of repeating these steps in detail here we describe how two AMP algorithms derived for independent inference problems can be composed into a single AMP for a structured inference problem. In particular, this is illustrated for the case of interest in this manuscript, namely a spiked-matrix estimation with single-layer generative model prior. In this case, the underlying inference problems are the *rank-one matrix factorization* (MF) [1] and the *generalized linear model* (GLM)[3]. We focus the derivation on the more general Wishart model ($\mathbf{u}\mathbf{v}^\intercal$) as the result for the Wigner model ($\mathbf{v}\mathbf{v}^\intercal$) flows directly from it.

**Factor graph:** In order to compose AMP algorithms, the idea is to replace the separable prior $P_v$ of the variable $\mathbf{v}$ in the low-rank MF model by a non-separable prior coming from a GLM model with channel $P_{\text{out}}$ (see definition in eq. (1.3)), while keeping separable distributions $P_u$ and $P_z$ for the variables $\mathbf{u} \in \mathbb{R}^n$ and $\mathbf{z} \in \mathbb{R}^k$.[1] Hence to obtain the factor graph of the $\mathbf{u}\mathbf{v}^\intercal$ model, we connect the factor graphs of the MF (in green) and GLM (in red) models together by means of $P_{\text{out}}$ (in black) (See Fig. 1).

$$P_u(u_\mu) \quad u_\mu \qquad P\left(Y_{\mu i}\Big|\tfrac{1}{\sqrt{p}}u_\mu v_i\right) \qquad v_i \qquad P_{\text{out}}\left(v_i\Big|\tfrac{1}{\sqrt{k}}\mathbf{W}_i^\intercal\mathbf{z}\right) \quad z_l \quad P_z(z_l)$$

Figure 1: Factor graph corresponding to a low-rank matrix factorization layer (green) with a prior coming from a GLM (red). We stress that in the classical low-rank layer, red part does not exist and black nodes $P_{\text{out}}(v_i|.)$ are replaced by separable prior $P_v(v_i)$.

## 4.1 Heuristic Derivation

We recall the AMP equations for the two modules and we will explain how to plug them together.

**AMP equations for the MF layer (variables v and u):** Consider the low-rank matrix factorization model eq. (1.2) with separable priors $P_u$ and $P_v$ for the variables $\mathbf{u}$ and $\mathbf{v}$. The corresponding non-

Bayes-optimal AMP equations, given in [1], read:

$$
\begin{cases}
\hat{\mathbf{u}}^{t+1} = f_u(\mathbf{B}_u^t, A_u^t)\,, \\[4pt]
\hat{\mathbf{c}}_u^{t+1} = \partial_B f_u(\mathbf{B}_u^t, A_u^t)\,, \\[4pt]
\hat{\mathbf{v}}^{t+1} = f_v(\mathbf{B}_v^t, A_v^t)\,, \\[4pt]
\hat{\mathbf{c}}_v^{t+1} = \partial_B f_v(\mathbf{B}_v^t, A_v^t)\,,
\end{cases}
\quad\text{and}\quad
\begin{cases}
\mathbf{B}_v^t = \frac{1}{\sqrt{p}} S^\mathsf{T} \hat{\mathbf{u}}^t - \frac{1}{p}(S^2)^\mathsf{T} \hat{\mathbf{c}}_u^t I_p \hat{\mathbf{v}}^{t-1}\,, \\[4pt]
A_v^t = \left[ \frac{1}{p}(S^2)^\mathsf{T}(\hat{\mathbf{u}}^t)^2 - \frac{1}{p} R^\mathsf{T}\left(\hat{\mathbf{c}}_u^t + (\hat{\mathbf{u}}^t)^2\right)\right] I_p\,, \\[4pt]
\mathbf{B}_u^t = \frac{1}{\sqrt{p}} S \hat{\mathbf{v}}^t - \frac{1}{p} S^2 \hat{\mathbf{c}}_v^t I_n \hat{\mathbf{u}}^{t-1}\,, \\[4pt]
A_u^t = \left[ \frac{1}{p} S^2 (\hat{\mathbf{v}}^t)^2 - \frac{1}{p} R\left(\hat{\mathbf{c}}_v^t + (\hat{\mathbf{v}}^t)^2\right)\right] I_n\,,
\end{cases}
\tag{4.1}
$$

with matrices $S$ and $R$ defined as

$$
S_{\mu i} = \frac{Y_{\mu i}}{\Delta} \quad\text{and}\quad R_{\mu i} = -\frac{1}{\Delta} + S_{\mu i}^2\,,
\tag{4.2}
$$

and the operation $(\cdot)^2$ is taken component-wise. The update function $f_u$ is the mean of $Q_u$, defined in sec. 1.3, and $f_v$ is the mean of the distribution $Q_v(v; B, A) \equiv \dfrac{1}{\mathcal{Z}_v(B, A)} P_v(v) e^{-\frac{1}{2} A v^2 + B v}$.

**AMP equations for the GLM layer (variable z):**  On the other hand, the non-Bayes-optimal AMP equations for the GLM model in eq. (1.3), given in [3], read

$$
\begin{cases}
\hat{\mathbf{z}}^{t+1} = f_z(\boldsymbol{\gamma}^t, \Lambda^t) \\[4pt]
\hat{\mathbf{c}}_z^{t+1} = \partial_\gamma f_z(\boldsymbol{\gamma}^t, \Lambda^t) \\[4pt]
\mathbf{g}^t = f_{\text{out}}\left(\mathbf{v}^\star, \boldsymbol{\omega}^t, V^t\right)
\end{cases}
\quad\text{and}\quad
\begin{cases}
\Lambda^t = -\frac{1}{k}(W^2)^\mathsf{T} \partial_\omega \mathbf{g}^t I_k \quad\text{and}\quad \boldsymbol{\gamma}^t = \frac{1}{\sqrt{k}} W^\mathsf{T} \mathbf{g}^t + \Lambda^t \hat{\mathbf{z}}^t \\[4pt]
V^t = \frac{1}{k}(W^2) \hat{\mathbf{c}}_z^t I_p \quad\text{and}\quad \boldsymbol{\omega}^t = \frac{1}{\sqrt{k}} W \hat{\mathbf{z}}^t - V^t \mathbf{g}^{t-1}
\end{cases}
\tag{4.3}
$$

where $f_z$ is the mean of $Q_z$ defined in SM. 1.3 and $f_{\text{out}}$ is the mean of $V^{-1}(x - \omega)$ with respect to $Q_{\text{out}}\left(x; v^\star, \omega, V\right) = \frac{P_{\text{out}}(v^\star | x)}{\mathcal{Z}_{\text{out}}(v^\star, \omega, V)} e^{-\frac{1}{2} V^{-1}(x-\omega)^2}$.

**Plug and play:**  In principle composing the AMP equations for the inference problems above is complicated, and require analyzing the BP equations on the composed factor graph in Fig. 1. However, the upshot of this cumbersome computation is rather simple: the AMP equations for the composed model are equivalent to coupling the MF eqs.(4.1) and the GLM eqs.(4.3) by replacing $Q_v(v; B, A)$ and $Q_{\text{out}}(x; \omega, V)$ with the following joint distribution:

$$
Q_{\text{out}}(v, x; B, A, \omega, V) \equiv \frac{1}{\mathcal{Z}_{\text{out}}(B, A, \omega, V)} e^{-\frac{1}{2} A v^2 + B v} P_{\text{out}}(v | x) e^{-\frac{1}{2} V^{-1}(x-\omega)^2}\,.
\tag{4.4}
$$

The associated update functions $f_v$, $f_{\text{out}}$ are thus replaced by the mean of $v$ and $V^{-1}(x - \omega)$ with respect to this new joint distribution $Q_{\text{out}}$. Replacing this distribution in both AMP algorithms eq. (4.1)-(4.3), we obtain the AMP algorithm of the structured model, summarized in the next section.

## 4.2 Summary of the AMP algorithms - $vv^\mathsf{T}$ and $uv^\mathsf{T}$

Replacing the separable distributions $Q_u$ and $Q_{\text{out}}$ by the joint distribution eq. (4.4) and corresponding update functions as described above, we obtain the following AMP algorithm for the Wishart model:

### 4.2.1 Wishart model ($uv^\mathsf{T}$):

**Input:** vector $Y \in \mathbb{R}^{n \times p}$ and matrix $W \in \mathbb{R}^{p \times k}$:
*Initialize to zero:* $(\mathbf{g}, \hat{\mathbf{u}}, \hat{\mathbf{v}}, \mathbf{B}_u, A_u, \mathbf{B}_v, A_v)^{t=0}$
*Initialize with:* $\hat{\mathbf{u}}^{t=1} = \mathcal{N}(0, \sigma^2)$, $\hat{\mathbf{v}}^{t=1} = \mathcal{N}(0, \sigma^2)$, $\hat{\mathbf{z}}^{t=1} = \mathcal{N}(0, \sigma^2)$,
$\quad\quad\quad\quad \hat{\mathbf{c}}_u^{t=1} = \mathbb{1}_n, \hat{\mathbf{c}}_v^{t=1} = \mathbb{1}_p, \hat{\mathbf{c}}_z^{t=1} = \mathbb{1}_k$. $t = 1$
**repeat**
*Spiked layer:*
$$
\mathbf{B}_u^t = \frac{1}{\sqrt{p}} S \hat{\mathbf{v}}^t - \frac{1}{p} S^2 \hat{\mathbf{c}}_v^t I_n \hat{\mathbf{u}}^{t-1} \quad\text{and}\quad A_u^t = \left[\frac{1}{p} S^2 (\hat{\mathbf{v}}^t)^2 - \frac{1}{p} R\left(\hat{\mathbf{c}}_v^t + (\hat{\mathbf{v}}^t)^2\right)\right] I_n
$$

$$\mathbf{B}_v^t = \tfrac{1}{\sqrt{p}} S^\intercal \hat{\mathbf{u}}^t - \tfrac{1}{p}(S^2)^\intercal \hat{\mathbf{c}}_u^t I_p \hat{\mathbf{v}}^{t-1} \quad \text{and} \quad A_v^t = \left[ \tfrac{1}{p}(S^2)^\intercal (\hat{\mathbf{u}}^t)^2 - \tfrac{1}{p} R^\intercal \left( \hat{\mathbf{c}}_u^t + (\hat{\mathbf{u}}^t)^2 \right) \right] I_p$$

*Generative layer:*
$$V^t = \tfrac{1}{k}(W^2)\hat{\mathbf{c}}_z^t I_p \quad \text{and} \quad \boldsymbol{\omega}^t = \tfrac{1}{\sqrt{k}} W \hat{\mathbf{z}}^t - V^t \mathbf{g}^{t-1} \quad \text{and} \quad \mathbf{g}^t = f_{\text{out}}\left( \mathbf{B}_v^t, A_v^t, \boldsymbol{\omega}^t, V^t \right)$$

$$\Lambda^t = -\tfrac{1}{k}(W^2)^\intercal \partial_\omega \mathbf{g}^t I_k \quad \text{and} \quad \boldsymbol{\gamma}^t = \tfrac{1}{\sqrt{k}} W^\intercal \mathbf{g}^t + \Lambda^t \hat{\mathbf{z}}^t$$

*Update of the estimated marginals:*
$$\hat{\mathbf{u}}^{t+1} = f_u(\mathbf{B}_u^t, A_u^t) \quad \text{and} \quad \hat{\mathbf{c}}_u^{t+1} = \partial_B f_u(\mathbf{B}_u^t, A_u^t)$$

$$\hat{\mathbf{v}}^{t+1} = f_v(\mathbf{B}_v^t, A_v^t, \boldsymbol{\omega}^t, V^t) \quad \text{and} \quad \hat{\mathbf{c}}_v^{t+1} = \partial_B f_v(\mathbf{B}_v^t, A_v^t, \boldsymbol{\omega}^t, V^t)$$

$$\hat{\mathbf{z}}^{t+1} = f_z(\boldsymbol{\gamma}^t, \Lambda^t) \quad \text{and} \quad \hat{\mathbf{c}}_z^{t+1} = \partial_\gamma f_z(\boldsymbol{\gamma}^t, \Lambda^t)$$

$t = t + 1$
**until** Convergence
**Output:** $\hat{\mathbf{u}}, \hat{\mathbf{v}}, \hat{\mathbf{z}}$

### 4.2.2  Wigner model ($vv^\intercal$):

The AMP algorithm for the Wigner model can be easily obtained from the one above by simply taking taking $\mathbf{u}^t = \mathbf{v}^t$, and removing redundant equations:

**Input:** vector $Y \in \mathbb{R}^{p \times p}$ and matrix $W \in \mathbb{R}^{p \times k}$:
*Initialize to zero:* $(\mathbf{g}, \hat{\mathbf{v}}, \mathbf{B}_v, A_v)^{t=0}$
*Initialize with:* $\hat{\mathbf{v}}^{t=1} = \mathcal{N}(0, \sigma^2)$, $\hat{\mathbf{z}}^{t=1} = \mathcal{N}(0, \sigma^2)$,
$\qquad\qquad \hat{\mathbf{c}}_v^{t=1} = \mathbb{1}_p, \hat{\mathbf{c}}_z^{t=1} = \mathbb{1}_k$. $t = 1$
**repeat**
  *Spiked layer:*
$$\mathbf{B}_v^t = \tfrac{1}{\sqrt{p}} S \hat{\mathbf{v}}^t - \tfrac{1}{p} S^2 \hat{\mathbf{c}}_v^t I_p \hat{\mathbf{v}}^{t-1} \quad \text{and} \quad A_v^t = \left[ \tfrac{1}{p} S^2 (\hat{\mathbf{v}}^t)^2 - \tfrac{1}{p} R \left( \hat{\mathbf{c}}_v^t + (\hat{\mathbf{v}}^t)^2 \right) \right] I_p$$

  *Generative layer:*
$$V^t = \tfrac{1}{k}(W^2)\hat{\mathbf{c}}_z^t I_p \quad \text{and} \quad \boldsymbol{\omega}^t = \tfrac{1}{\sqrt{k}} W \hat{\mathbf{z}}^t - V^t \mathbf{g}^{t-1} \quad \text{and} \quad \mathbf{g}^t = f_{\text{out}}\left( \mathbf{B}_v^t, A_v^t, \boldsymbol{\omega}^t, V^t \right)$$

$$\Lambda^t = -\tfrac{1}{k}(W^2)^\intercal \partial_\omega \mathbf{g}^t I_k \quad \text{and} \quad \boldsymbol{\gamma}^t = \tfrac{1}{\sqrt{k}} W^\intercal \mathbf{g}^t + \Lambda^t \hat{\mathbf{z}}^t$$

  *Update of the estimated marginals:*
$$\hat{\mathbf{v}}^{t+1} = f_v(\mathbf{B}_v^t, A_v^t, \boldsymbol{\omega}^t, V^t) \quad \text{and} \quad \hat{\mathbf{c}}_v^{t+1} = \partial_B f_v(\mathbf{B}_v^t, A_v^t, \boldsymbol{\omega}^t, V^t)$$

$$\hat{\mathbf{z}}^{t+1} = f_z(\boldsymbol{\gamma}^t, \Lambda^t) \quad \text{and} \quad \hat{\mathbf{c}}_z^{t+1} = \partial_\gamma f_z(\boldsymbol{\gamma}^t, \Lambda^t)$$

$t = t + 1$
**until** Convergence
**Output:** $\hat{\mathbf{u}}, \hat{\mathbf{v}}, \hat{\mathbf{z}}$

## 4.3  Simplified algorithms in the Bayes-optimal setting

In the Bayes-optimal setting, it can be shown using Nishimori property (See sec. 3 of [1]) that:

$$\langle R \rangle = 0 \iff \langle S^2 \rangle = \frac{1}{\Delta}, \qquad\qquad \langle \partial_\omega \mathbf{g}^t \rangle = -\langle (\mathbf{g}^t)^2 \rangle \qquad (4.5)$$

where $\langle \cdot \rangle$ denotes the average with respect to the posterior distribution in eq. (1.8).

Note that the AMP algorithm derived above is also valid for arbitrary weight matrix $W \in \mathbb{R}^{p \times k}$. In the case of interest where $W_{il} \underset{\text{i.i.d.}}{\sim} \mathcal{N}(0,1)$, we can further simplify $\mathbb{E}\left[ W_{il}^2 \right] = 1$. Together, these simplifications give:

### 4.3.1  Wishart model ($uv^\intercal$) - Bayes-optimal
  **Input:** vector $Y \in \mathbb{R}^{n \times p}$ and matrix $W \in \mathbb{R}^{p \times k}$:

*Initialize to zero:* $(\mathbf{g}, \hat{\mathbf{u}}, \hat{\mathbf{v}}, \mathbf{B}_v, A_v, \mathbf{B}_u, A_u)^{t=0}$
*Initialize with:* $\hat{\mathbf{u}}^{t=1} = \mathcal{N}(0, \sigma^2), \hat{\mathbf{v}}^{t=1} = \mathcal{N}(0, \sigma^2), \hat{\mathbf{z}}^{t=1} = \mathcal{N}(0, \sigma^2),$
$\hat{\mathbf{c}}_u^{t=1} = \mathbb{1}_n, \hat{\mathbf{c}}_v^{t=1} = \mathbb{1}_p, \hat{\mathbf{c}}_z^{t=1} = \mathbb{1}_k. \ t = 1$
**repeat**
   *Spiked layer:*
$$\mathbf{B}_u^t = \frac{1}{\Delta}\frac{Y}{\sqrt{p}}\hat{\mathbf{v}}^t - \frac{1}{\Delta}\frac{\mathbb{1}_p^\intercal \hat{\mathbf{c}}_v^t}{p}\mathrm{I}_n \hat{\mathbf{u}}^{t-1} \quad \text{and} \quad A_u^t = \frac{1}{\Delta}\frac{\|\hat{\mathbf{v}}^t\|_2^2}{p}\mathrm{I}_n$$

$$\mathbf{B}_v^t = \frac{1}{\Delta}\frac{Y^\intercal}{\sqrt{p}}\hat{\mathbf{u}}^t - \frac{1}{\Delta}\frac{\mathbb{1}_n^\intercal \hat{\mathbf{c}}_u^t}{p}\mathrm{I}_p \hat{\mathbf{v}}^{t-1} \quad \text{and} \quad A_v^t = \frac{1}{\Delta}\frac{\|\hat{\mathbf{u}}^t\|_2^2}{p}\mathrm{I}_p$$

   *Generative layer:*
$$V^t = \frac{1}{k}\left(\mathbb{1}_k^\intercal \hat{\mathbf{c}}_z^t\right)\mathrm{I}_p \quad \text{and} \quad \boldsymbol{\omega}^t = \frac{1}{\sqrt{k}}W\hat{\mathbf{z}}^t - V^t\mathbf{g}^{t-1} \quad \text{and} \quad \mathbf{g}^t = f_{\text{out}}\left(\mathbf{B}_v^t, A_v^t, \boldsymbol{\omega}^t, V^t\right)$$

$$\Lambda^t = \frac{1}{k}\|\mathbf{g}^t\|_2^2 \mathrm{I}_k \quad \text{and} \quad \boldsymbol{\gamma}^t = \frac{1}{\sqrt{k}}W^\intercal \mathbf{g}^t + \Lambda^t \hat{\mathbf{z}}^t$$

   *Update of the estimated marginals:*
$$\hat{\mathbf{u}}^{t+1} = f_u(\mathbf{B}_u^t, A_u^t) \quad \text{and} \quad \hat{\mathbf{c}}_u^{t+1} = \partial_B f_u(\mathbf{B}_u^t, A_u^t)$$

$$\hat{\mathbf{v}}^{t+1} = f_v(\mathbf{B}_v^t, A_v^t, \boldsymbol{\omega}^t, V^t) \quad \text{and} \quad \hat{\mathbf{c}}_v^{t+1} = \partial_B f_v(\mathbf{B}_v^t, A_v^t, \boldsymbol{\omega}^t, V^t)$$

$$\hat{\mathbf{z}}^{t+1} = f_z(\boldsymbol{\gamma}^t, \Lambda^t) \quad \text{and} \quad \hat{\mathbf{c}}_z^{t+1} = \partial_\gamma f_z(\boldsymbol{\gamma}^t, \Lambda^t)$$

   $t = t + 1$
**until** Convergence
**Output:** $\hat{\mathbf{u}}, \hat{\mathbf{v}}, \hat{\mathbf{z}}$

### 4.3.2 Wigner model ($\mathbf{vv}^\intercal$) - Bayes-optimal

**Input:** vector $Y \in \mathbb{R}^{p \times p}$ and matrix $W \in \mathbb{R}^{p \times k}$:
*Initialize to zero:* $(\mathbf{g}, \hat{\mathbf{v}}, \mathbf{B}_v, A_v)^{t=0}$
*Initialize with:* $\hat{\mathbf{v}}^{t=1} = \mathcal{N}(0, \sigma^2), \hat{\mathbf{z}}^{t=1} = \mathcal{N}(0, \sigma^2),$
$\hat{\mathbf{c}}_v^{t=1} = \mathbb{1}_p, \hat{\mathbf{c}}_z^{t=1} = \mathbb{1}_k. \ t = 1$
**repeat**
   *Spiked layer:*
$$\mathbf{B}_v^t = \frac{1}{\Delta}\frac{Y}{\sqrt{p}}\hat{\mathbf{v}}^t - \frac{1}{\Delta}\frac{\mathbb{1}_p^\intercal \hat{\mathbf{c}}_v^t}{p}\hat{\mathbf{v}}^{t-1} \quad \text{and} \quad A_v^t = \frac{1}{\Delta}\frac{\|\hat{\mathbf{v}}^t\|_2^2}{p}\mathrm{I}_p$$

   *Generative layer:*
$$V^t = \frac{1}{k}\left(\mathbb{1}_k^\intercal \hat{\mathbf{c}}_z^t\right)\mathrm{I}_p \quad \text{and} \quad \boldsymbol{\omega}^t = \frac{1}{\sqrt{k}}W\hat{\mathbf{z}}^t - V^t\mathbf{g}^{t-1} \quad \text{and} \quad \mathbf{g}^t = f_{\text{out}}\left(\mathbf{B}_v^t, A_v^t, \boldsymbol{\omega}^t, V^t\right)$$

$$\Lambda^t = \frac{1}{k}\|\hat{\mathbf{g}}^t\|_2^2 \mathrm{I}_k \quad \text{and} \quad \boldsymbol{\gamma}^t = \frac{1}{\sqrt{k}}W^\intercal \mathbf{g}^t + \Lambda^t \hat{\mathbf{z}}^t$$

   *Update of the estimated marginals:*
$$\hat{\mathbf{v}}^{t+1} = f_v(\mathbf{B}_v^t, A_v^t, \boldsymbol{\omega}^t, V^t) \quad \text{and} \quad \hat{\mathbf{c}}_v^{t+1} = \partial_B f_v(\mathbf{B}_v^t, A_v^t, \boldsymbol{\omega}^t, V^t)$$

$$\hat{\mathbf{z}}^{t+1} = f_z(\boldsymbol{\gamma}^t, \Lambda^t) \quad \text{and} \quad \hat{\mathbf{c}}_z^{t+1} = \partial_\gamma f_z(\boldsymbol{\gamma}^t, \Lambda^t)$$

   $t = t + 1$
**until** Convergence
**Output:** $\hat{\mathbf{v}}, \hat{\mathbf{z}}$

## 4.4 Derivation of the state evolution equations

The AMP algorithms above are valid for any large but finite sizes $k, n, p$. A central object of interest are the *state evolution equations* (SE) that predict the algorithm's behaviour in the infinite size limit $k \to \infty$. We show in this section the derivation of these equations, directly from the algorithm to explicitly show that it provides the same set of equations as the saddle point equations obtained from the replica free entropy eq. (2.36). As before, we focus on the derivation of the more general Wishart model $\mathbf{uv}^\intercal$, and quote the result for the symmetric $\mathbf{vv}^\intercal$. We first derive the SE equations without loss

of generality in the non-Bayes-optimal case, and we will state them in their simplified formulation in the Bayes-optimal case.

The idea is to compute the average distributions of the messages involved in the AMP algorithm updates in sec. (4.2), namely $\mathbf{B}_u, A_u, \mathbf{B}_v, A_v, \boldsymbol{\omega}, V, \boldsymbol{\gamma}$ and $V$. The usual derivation starts with rBP equations that we did not present here, see [1]. However this equations are roughly equivalent to AMP messages if we remove the Onsager terms containing messages with delayed time indices $(\cdot)^{t-1}$.

**Definition of the overlap parameters:**   We first define the order parameters, called overlaps in the physics literature, that will measure the correlation of the Bayesian estimator with the ground truth signals

$$m_u^t \equiv \mathbb{E}_{\mathbf{u}^\star} \lim_{n\to\infty} \frac{(\hat{\mathbf{u}}^t)^\mathsf{T}\mathbf{u}^\star}{n}, \quad q_u^t \equiv \mathbb{E}_{\mathbf{u}^\star} \lim_{n\to\infty} \frac{(\hat{\mathbf{u}}^t)^\mathsf{T}\hat{\mathbf{u}}^t}{n}, \quad \Sigma_u^t \equiv \mathbb{E}_{\mathbf{u}^\star} \lim_{n\to\infty} \frac{\mathbb{1}_n^\mathsf{T}\hat{\mathbf{c}}^{u,t}}{n},$$

$$m_v^t \equiv \mathbb{E}_{\mathbf{v}^\star} \lim_{p\to\infty} \frac{(\hat{\mathbf{v}}^t)^\mathsf{T}\mathbf{v}^\star}{p}, \quad q_v^t \equiv \mathbb{E}_{\mathbf{v}^\star} \lim_{p\to\infty} \frac{(\hat{\mathbf{v}}^t)^\mathsf{T}\hat{\mathbf{v}}^t}{p}, \quad \Sigma_v^t \equiv \mathbb{E}_{\mathbf{v}^\star} \lim_{p\to\infty} \frac{\mathbb{1}_p^\mathsf{T}\hat{\mathbf{c}}^{v,t}}{p}, \quad (4.6)$$

$$m_z^t \equiv \mathbb{E}_{\mathbf{z}^\star} \lim_{k\to\infty} \frac{(\hat{\mathbf{z}}^t)^\mathsf{T}\mathbf{z}^\star}{k}, \quad q_z^t \equiv \mathbb{E}_{\mathbf{z}^\star} \lim_{k\to\infty} \frac{(\hat{\mathbf{z}}^t)^\mathsf{T}\hat{\mathbf{z}}^t}{k}, \quad \Sigma_z^t \equiv \mathbb{E}_{\mathbf{z}^\star} \lim_{k\to\infty} \frac{\mathbb{1}_k^\mathsf{T}\hat{\mathbf{c}}^{z,t}}{k}.$$

As the algorithm performance, such as the mean squared error or the generalization error, can be computed directly from these overlap parameters, our goal is to derive the their average distribution in the infinite size limit.

**Messages distributions:**   As stressed above, we compute the average distribution of the messages, taking the average over variables $W, \xi$, the planted solutions $\mathbf{v}^\star, \mathbf{u}^\star, \mathbf{z}^\star$ and taking the limit $k \to \infty$. Note that we use the BP independence assumption over the messages and keep only dominant terms in the $1/p$ expansion.

- $B_u, A_u$:   Starting with (4.2.1), we obtain

$$\mathbb{E}\left[\mathbf{B}_u^t\right] = \frac{1}{\sqrt{p}\Delta}\mathbb{E}\left[Y\hat{\mathbf{v}}^t\right] = \frac{1}{\sqrt{p}\Delta}\mathbb{E}\left[\left(\frac{\mathbf{u}^\star(\mathbf{v}^\star)^\mathsf{T}}{\sqrt{p}} + \sqrt{\Delta}\xi\right)\hat{\mathbf{v}}^t\right] \xrightarrow[p\to\infty]{} \frac{m_v^t}{\Delta}\mathbf{u}^\star, \quad (4.7)$$

$$\mathbb{E}\left[\mathbf{B}_u^t(\mathbf{B}_u^t)^\mathsf{T}\right] = \frac{1}{p\Delta^2}\mathbb{E}\left[Y\hat{\mathbf{v}}^t(\hat{\mathbf{v}}^t)^\mathsf{T}Y^\mathsf{T}\right] = \frac{1}{\Delta}\frac{1}{p}\mathbb{E}\left[\xi\hat{\mathbf{v}}^t(\hat{\mathbf{v}}^t)^\mathsf{T}\xi^\mathsf{T}\right] + o\left(1/p\right) \xrightarrow[p\to\infty]{} \frac{q_v^t}{\Delta}\mathrm{I}_n, \quad (4.8)$$

$$\mathbb{E}\left[A_u^t\right] = \mathbb{E}\left[\frac{1}{p}S^2(\hat{\mathbf{v}}^t)^2 - \frac{1}{p}R\left(\hat{\mathbf{c}}_v^t + (\hat{\mathbf{v}}^t)^2\right)\right]\mathrm{I}_n \xrightarrow[p\to\infty]{} \frac{q_v^t}{\Delta}\mathrm{I}_n - \bar{R}\Sigma_v^t\mathrm{I}_n. \quad (4.9)$$

where we defined, see [1],

$$\bar{R} = \mathbb{E}_{P(Y|\omega)}\left[\partial_\omega^2 g + (\partial_\omega g)^2\right]_{\omega=0} = \int \prod_{1\leq i\leq p, 1\leq\mu\leq n} dY_{\mu i}\, e^{g^\star(Y_{\mu i},0)}\left[\partial_\omega^2 g + (\partial_\omega g)^2\right]_{Y,\omega=0}$$

with $P(Y|\omega)$, $g$ defined in sec. 2.1 and $g^\star$ the ground truth channel function. Note that in the Bayes-optimal case, $g^\star = g$ that yields $\bar{R} = 0$ as mentioned in eq. (2.8).

- $B_v, A_v$:   Similarly,

$$\mathbb{E}\left[\mathbf{B}_v^t\right] = \frac{1}{\sqrt{p}\Delta}\mathbb{E}\left[Y^\mathsf{T}\hat{\mathbf{u}}^t\right] = \frac{1}{\sqrt{p}\Delta}\mathbb{E}\left[\left(\frac{\mathbf{u}^\star(\mathbf{v}^\star)^\mathsf{T}}{\sqrt{p}} + \sqrt{\Delta}\xi\right)^\mathsf{T}\hat{\mathbf{u}}^t\right] \xrightarrow[p\to\infty]{} \beta\frac{m_u^t}{\Delta}\mathbf{v}^\star, \quad (4.10)$$

$$\mathbb{E}\left[\mathbf{B}_v^t(\mathbf{B}_v^t)^\mathsf{T}\right] = \frac{1}{p\Delta^2}\mathbb{E}\left[Y^\mathsf{T}\hat{\mathbf{u}}^t(\hat{\mathbf{u}}^t)^\mathsf{T}Y\right] \xrightarrow[p\to\infty]{} \beta\frac{q_u^t}{\Delta}\mathrm{I}_p, \quad (4.11)$$

$$\mathbb{E}\left[A_v^t\right] = \mathbb{E}\left[\frac{1}{p}(S^\mathsf{T})^2(\hat{\mathbf{u}}^t)^2 - \frac{1}{p}R\left(\hat{\mathbf{c}}_u^t + (\hat{\mathbf{u}}^t)^2\right)\right]\mathrm{I}_p \xrightarrow[p\to\infty]{} \beta\left(\frac{q_u^t}{\Delta} - \bar{R}\Sigma_u^t\right)\mathrm{I}_p. \quad (4.12)$$

- $\omega, V$:

$$\mathbb{E}\left[\boldsymbol{\omega}^t\right] = \mathbb{E}\left[\frac{1}{\sqrt{k}} W\hat{\mathbf{z}}^t\right] = \mathbf{0}_p\,, \tag{4.13}$$

$$\mathbb{E}\left[\boldsymbol{\omega}^t(\boldsymbol{\omega}^t)^\intercal\right] = \mathbb{E}\left[\frac{1}{k} W\hat{\mathbf{z}}^t(\hat{\mathbf{z}}^t)^\intercal W^\intercal\right] \xrightarrow[n\to\infty]{} q_z^t \mathrm{I}_p\,, \tag{4.14}$$

$$\mathbb{E}\left[V\right] = \mathbb{E}\left[\frac{1}{k}(W^2)\hat{\mathbf{c}}_z^t \mathrm{I}_p\right] \xrightarrow[k\to\infty]{} \Sigma_z^t \mathrm{I}_p\,. \tag{4.15}$$

**Conclusion:**    Finally we conclude that to leading order:

$$\mathbf{B}_u \sim \frac{m_v^t}{\Delta}\mathbf{u}^\star + \sqrt{\frac{q_v^t}{\Delta}}\boldsymbol{\xi}_u\,, \qquad\qquad A_u^t \sim \frac{q_v^t}{\Delta}\mathrm{I}_n - \bar{R}\Sigma_v^t \mathrm{I}_n\,, \tag{4.16}$$

$$\mathbf{B}_v \sim \beta\frac{m_u^t}{\Delta}\mathbf{v}^\star + \sqrt{\beta\frac{q_u^t}{\Delta}}\boldsymbol{\xi}_v\,, \qquad\qquad A_v^t \sim \beta\left(\frac{q_u^t}{\Delta} - \bar{R}\Sigma_u^t\right)\mathrm{I}_p\,, \tag{4.17}$$

$$\boldsymbol{\omega} \sim \sqrt{q_z^t}\boldsymbol{\eta}\,, \qquad\qquad V \sim \Sigma_z^t \mathrm{I}_p\,, \tag{4.18}$$

with $\boldsymbol{\xi}_u \sim \mathcal{N}\left(\mathbf{0}_n, \mathrm{I}_n\right), \boldsymbol{\xi}_v \sim \mathcal{N}\left(\mathbf{0}_n, \mathrm{I}_n\right), \boldsymbol{\eta} \sim \mathcal{N}\left(\mathbf{0}_p, \mathrm{I}_p\right)$.

### 4.4.1   State evolution - Non Bayes-optimal case

With the averaged limiting distributions of all the messages, we can now compute the state evolution of the overlaps. Using the definition of the overlaps eq. (4.7) and distributions in eq. (4.18), we obtain:

**Variable u:**

$$q_u^{t+1} \equiv \mathbb{E}_{\mathbf{u}^\star} \lim_{n\to\infty} \frac{1}{n}(\hat{\mathbf{u}}^{t+1})^\intercal \hat{\mathbf{u}}^{t+1} = \mathbb{E}_{\mathbf{u}^\star} \lim_{n\to\infty} \frac{1}{n} f_u(\mathbf{B}_u^t, A_u^t)^\intercal f_u(\mathbf{B}_u^t, A_u^t) \tag{4.19}$$

$$= \mathbb{E}_{u^\star, \xi}\left[f_u\left(\frac{m_v^t}{\Delta}u^\star + \sqrt{\frac{q_v^t}{\Delta}}\xi, \frac{q_v^t}{\Delta} - \bar{R}\Sigma_v^t\right)^2\right]$$

$$m_u^{t+1} \equiv \mathbb{E}_{\mathbf{u}^\star} \lim_{n\to\infty} \frac{1}{n}(\hat{\mathbf{u}}^{t+1})^\intercal \mathbf{u}^\star = \mathbb{E}_{\mathbf{u}^\star} \lim_{n\to\infty} \frac{1}{n} f_u(\mathbf{B}_u^t, A_u^t)^\intercal \mathbf{u}^\star \tag{4.20}$$

$$= \mathbb{E}_{u^\star, \xi}\left[f_u\left(\frac{m_v^t}{\Delta}u^\star + \sqrt{\frac{q_v^t}{\Delta}}\xi, \frac{q_v^t}{\Delta} - \bar{R}\Sigma_v^t\right)u^\star\right]$$

$$\Sigma_u^{t+1} \equiv \mathbb{E}_{\mathbf{u}^\star} \lim_{n\to\infty} \frac{1}{n}\mathbb{1}_n^\intercal \hat{\mathbf{c}}^{u,t+1} = \mathbb{E}_{\mathbf{u}^\star} \lim_{n\to\infty} \frac{1}{n}\partial_B f_u(\mathbf{B}_u^t, A_u^t)^\intercal \mathbb{1}_n \tag{4.21}$$

$$= \mathbb{E}_{u^\star, \xi}\left[\partial_B f_u\left(\frac{m_v^t}{\Delta}u^\star + \sqrt{\frac{q_v^t}{\Delta}}\xi, \frac{q_v^t}{\Delta} - \bar{R}\Sigma_v^t\right)^2\right]$$

**Variable v:**

$$q_v^{t+1} = \mathbb{E}_{\mathbf{v}^\star} \lim_{p\to\infty} \frac{1}{p} (\hat{\mathbf{v}}^{t+1})^\intercal \hat{\mathbf{v}}^{t+1} = \mathbb{E}_{\mathbf{v}^\star} \lim_{p\to\infty} \frac{1}{p} f_v(\mathbf{B}_v^t, A_v^t, \boldsymbol{\omega}^t, V^t)^\intercal f_v(\mathbf{B}_v^t, A_v^t, \boldsymbol{\omega}^t, V^t) \quad (4.22)$$

$$= \mathbb{E}_{v^\star, \xi, \eta} \left[ f_v \left( \frac{\beta m_u^t}{\Delta} v^\star + \sqrt{\frac{\beta q_u^t}{\Delta}} \xi, \beta \left( \frac{q_u^t}{\Delta} - \bar{R}\Sigma_u^t \right), \sqrt{q_z^t}\eta, \Sigma_z^t \right)^2 \right]$$

$$m_v^{t+1} = \mathbb{E}_{\mathbf{v}^\star} \lim_{p\to\infty} \frac{1}{p} (\hat{\mathbf{v}}^{t+1})^\intercal \hat{\mathbf{v}}^{t+1} = \mathbb{E}_{\mathbf{v}^\star} \lim_{p\to\infty} \frac{1}{p} f_v(\mathbf{B}_v^t, A_v^t, \boldsymbol{\omega}^t, V^t)^\intercal \mathbf{v}^\star \quad (4.23)$$

$$= \mathbb{E}_{v^\star, \xi, \eta} \left[ f_v \left( \frac{\beta m_u^t}{\Delta} v^\star + \sqrt{\frac{\beta q_u^t}{\Delta}} \xi, \beta \left( \frac{q_u^t}{\Delta} - \bar{R}\Sigma_u^t \right), \sqrt{q_z^t}\eta, \Sigma_z^t \right) v^\star \right]$$

$$\Sigma_v^{t+1} = \mathbb{E}_{\mathbf{v}^\star} \lim_{p\to\infty} \frac{1}{p} \mathbb{1}_p^\intercal \hat{\mathbf{c}}^{z,t+1} = \mathbb{E}_{\mathbf{v}^\star} \lim_{p\to\infty} \frac{1}{p} \partial_\gamma f_v(\mathbf{B}_v^t, A_v^t, \boldsymbol{\omega}^t, V^t)^\intercal \mathbb{1}_p \quad (4.24)$$

$$= \mathbb{E}_{v^\star, \xi, \eta} \left[ \partial_\gamma f_v \left( \frac{\beta m_u^t}{\Delta} v^\star + \sqrt{\frac{\beta q_u^t}{\Delta}} \xi, \beta \left( \frac{q_u^t}{\Delta} - \bar{R}\Sigma_u^t \right), \sqrt{q_z^t}\eta, \Sigma_z^t \right)^2 \right]$$

**Variable ẑ:** We define intermediate *hat* overlap parameters[2] that will be useful in the following. The hat overlaps don't have as much physical meaning as the standard overlaps that quantify the reconstruction performances. Though we might notice anyway that all the overlap parameters are built similarly as function of the update functions $f_u, f_v, f_z$ and $f_{\text{out}}$ (See eq. (1.23)

$$\hat{q}_z^t \equiv \alpha \mathbb{E}_{v^\star, \xi, \eta} \left[ f_{\text{out}} \left( \frac{\beta m_u^t}{\Delta} v^\star + \sqrt{\frac{\beta q_u^t}{\Delta}} \xi, \beta \left( \frac{q_u^t}{\Delta} - \bar{R}\Sigma_u^t \right), \sqrt{q_z^t}\eta, \Sigma_z^t \right)^2 \right] \quad (4.25)$$

$$\hat{m}_z^t \equiv \alpha \mathbb{E}_{v^\star, \xi, \eta} \left[ \partial_x f_{\text{out}} \left( \frac{\beta m_u^t}{\Delta} v^\star + \sqrt{\frac{\beta q_u^t}{\Delta}} \xi, \beta \left( \frac{q_u^t}{\Delta} - \bar{R}\Sigma_u^t \right), \sqrt{q_z^t}\eta, \Sigma_z^t \right) v^\star \right] \quad (4.26)$$

$$\hat{\Sigma}_z^t \equiv \alpha \mathbb{E}_{v^\star, \xi, \eta} \left[ -\partial_\omega f_{\text{out}} \left( \frac{\beta m_u^t}{\Delta} v^\star + \sqrt{\frac{\beta q_u^t}{\Delta}} \xi, \beta \left( \frac{q_u^t}{\Delta} - \bar{R}\Sigma_u^t \right), \sqrt{q_z^t}\eta, \Sigma_z^t \right) \right] \quad (4.27)$$

**Variable z:** Averages are explicitly expressed as a function of the *hat* overlaps introduced just above:

$$\mathbb{E} \left[ \boldsymbol{\gamma}^t \right] \sim \hat{m}_z^t \mathbf{z}^\star \quad (4.28)$$

$$\mathbb{E} \left[ \boldsymbol{\gamma}^t (\boldsymbol{\gamma}^t)^\intercal \right] \sim \hat{q}_z^t \mathrm{I}_k \quad (4.29)$$

$$\mathbb{E} \left[ \Lambda^t \right] \sim \hat{\Sigma}_z^t \mathrm{I}_k \quad (4.30)$$

And we conclude that at the leading order:

$$\boldsymbol{\gamma}^t \sim \hat{m}_z^t \mathbf{z}^\star + \sqrt{\hat{q}_z^t} \boldsymbol{\xi}, \qquad\qquad \Lambda^t \sim \hat{\Sigma}_z^t \mathrm{I}_k . \quad (4.31)$$

with $\boldsymbol{\xi} \sim \mathcal{N}(\mathbf{0}_k, \mathrm{I}_k)$.

From these later equations, we obtain

$$q_z^{t+1} = \mathbb{E}_{\mathbf{z}^\star} \lim_{k \to \infty} \frac{1}{k} (\hat{\mathbf{z}}^{t+1})^\mathsf{T} \hat{\mathbf{z}}^{t+1} = \mathbb{E}_{\mathbf{z}^\star} \lim_{k \to \infty} \frac{1}{k} f_z(\boldsymbol{\gamma}^t, \Lambda^t)^\mathsf{T} f_z(\boldsymbol{\gamma}^t, \Lambda^t) \tag{4.32}$$

$$= \mathbb{E}_{z^\star, \xi} \left[ f_z \left( \hat{m}_z^t z^\star + \sqrt{\hat{q}_z^t} \xi, \hat{\Sigma}_z^t \right)^2 \right]$$

$$m_z^{t+1} = \mathbb{E}_{\mathbf{z}^\star} \lim_{k \to \infty} \frac{1}{k} (\hat{\mathbf{z}}^{t+1})^\mathsf{T} \mathbf{z}^\star = \mathbb{E}_{\mathbf{z}^\star} \lim_{k \to \infty} \frac{1}{k} f_z(\boldsymbol{\gamma}^t, \Lambda^t)^\mathsf{T} \mathbf{z}^\star \tag{4.33}$$

$$= \mathbb{E}_{z^\star, \xi} \left[ f_z \left( \hat{m}_z^t z^\star + \sqrt{\hat{q}_z^t} \xi, \hat{\Sigma}_z^t \right) z^\star \right]$$

$$\Sigma_z^{t+1} = \mathbb{E}_{\mathbf{z}^\star} \lim_{k \to \infty} \frac{1}{k} \mathbb{1}_k^\mathsf{T} \hat{\mathbf{c}}^{z,t+1} = \mathbb{E}_{\mathbf{z}^\star} \lim_{k \to \infty} \frac{1}{k} \mathbb{1}_k^\mathsf{T} \partial_\gamma f_z(\boldsymbol{\gamma}^t, \Lambda^t) \tag{4.34}$$

$$= \mathbb{E}_{z^\star, \xi} \left[ \partial_\gamma f_z \left( \hat{m}_z^t z^\star + \sqrt{\hat{q}_z^t} \xi, \hat{\Sigma}_z^t \right) \right]$$

Equations (4.19- 4.27, 4.32-4.34) constitute the closed set of AMP *state evolution equations* in the non-Bayes-optimal case.

### 4.4.2 State evolution - Bayes-optimal case

In the Bayes-optimal case, the Nishimori property (See sec. 1.1) implies $m_u = q_u$, $m_z = q_z$, $m_v = q_v$ and $\hat{m}_z = \hat{q}_z$, $\bar{R} = 0$ and we also note that $\Sigma_z^t = \rho_z - q_z^t$, $\hat{\Sigma}_z^t = \hat{q}_z^t$. The set of twelve state evolution equations reduce to only four, and they can be rewritten using a change of variable.

**Wishart model**

$$q_u^{t+1} = \mathbb{E}_\xi \left[ \mathcal{Z}_u \left( \sqrt{\frac{q_v^t}{\Delta}} \xi, \frac{q_v^t}{\Delta} \right) f_u \left( \sqrt{\frac{q_v^t}{\Delta}} \xi, \frac{q_v^t}{\Delta} \right)^2 \right] \tag{4.35}$$

$$= 2 \partial_{q_v} \Psi_u \left( q_v^t \right) ,$$

$$q_z^{t+1} = \mathbb{E}_\xi \left[ \mathcal{Z}_z \left( \sqrt{\hat{q}_z^t} \xi, \hat{q}_z^t \right) f_z \left( \sqrt{\hat{q}_z^t} \xi, \hat{q}_z^t \right)^2 \right] \tag{4.36}$$

$$= 2 \partial_{\hat{q}_z} \Psi_z \left( \hat{q}_z^t \right) ,$$

$$\hat{q}_z^t = \alpha \mathbb{E}_{\xi, \eta} \left[ \mathcal{Z}_{\text{out}} \left( \sqrt{\frac{\beta q_u^t}{\Delta}} \xi, \beta \frac{q_u^t}{\Delta}, \sqrt{q_z^t} \eta, \rho_z - q_z^t \right) f_{\text{out}} \left( \sqrt{\frac{\beta q_u^t}{\Delta}} \xi, \beta \frac{q_u^t}{\Delta}, \sqrt{q_z^t} \eta, \rho_z - q_z^t \right)^2 \right]$$

$$\tag{4.37}$$

$$= 2 \alpha \partial_{q_z} \Psi_{\text{out}} \left( \frac{\beta q_u^t}{\Delta}, q_z^t \right) ,$$

$$q_v^{t+1} = \mathbb{E}_{\xi, \eta} \left[ \mathcal{Z}_{\text{out}} \left( \sqrt{\frac{\beta q_u^t}{\Delta}} \xi, \beta \frac{q_u^t}{\Delta}, \sqrt{q_z^t} \eta, \rho_z - q_z^t \right) f_v \left( \sqrt{\frac{\beta q_u^t}{\Delta}} \xi, \beta \frac{q_u^t}{\Delta}, \sqrt{q_z^t} \eta, \rho_z - q_z^t \right)^2 \right]$$

$$\tag{4.38}$$

$$= 2 \partial_{q_u} \Psi_{\text{out}} \left( \frac{\beta q_u^t}{\Delta}, q_z^t \right) .$$

**Wigner model**

The state evolution for the Wigner model ($\mathbf{v}\mathbf{v}^\intercal$) is a particular case of the state evolution of the Wishart model discussed above, obtained by simply restricting $q_u = q_v$ and $\beta = 1$. It finally reads

$$q_z^{t+1} = \mathbb{E}_\xi \left[ \mathcal{Z}_z \left( \sqrt{\hat{q}_z^t}\xi, \hat{q}_z^t \right) f_z \left( \sqrt{\hat{q}_z^t}\xi, \hat{q}_z^t \right)^2 \right] \tag{4.39}$$

$$= 2\partial_{\hat{q}_z} \Psi_z \left( \hat{q}_z^t \right) ,$$

$$\hat{q}_z^t = \alpha \mathbb{E}_{\xi,\eta} \left[ \mathcal{Z}_{\text{out}} \left( \sqrt{\frac{q_v^t}{\Delta}}\xi, \frac{q_v^t}{\Delta}, \sqrt{q_z^t}\eta, \rho_z - q_z^t \right) f_{\text{out}} \left( \sqrt{\frac{q_v^t}{\Delta}}\xi, \frac{q_v^t}{\Delta}, \sqrt{q_z^t}\eta, \rho_z - q_z^t \right)^2 \right] \tag{4.40}$$

$$= 2\alpha \partial_{q_z} \Psi_{\text{out}} \left( \frac{q_v^t}{\Delta}, q_z^t \right) ,$$

$$q_v^{t+1} = \mathbb{E}_{\xi,\eta} \left[ \mathcal{Z}_{\text{out}} \left( \sqrt{\frac{q_v^t}{\Delta}}\xi, \frac{q_v^t}{\Delta}, \sqrt{q_z^t}\eta, \rho_z - q_z^t \right) f_v \left( \sqrt{\frac{q_v^t}{\Delta}}\xi, \frac{q_v^t}{\Delta}, \sqrt{q_z^t}\eta, \rho_z - q_z^t \right)^2 \right] \tag{4.41}$$

$$= 2\partial_{q_v} \Psi_{\text{out}} \left( \frac{q_v^t}{\Delta}, q_z^t \right) ,$$

which are precisely the state evolution equations derived from the replica trick in sec. 2, eq. (2.36), except that the algorithm provides the correct time indices in which the iterations should be taken.

# 5 Heuristic derivation of LAMP

We present in this section the derivation of the linearized-AMP (LAMP) spectral algorithm. This method, pioneered in [17], relies on the existence of the non-informative fixed point of the SE equations eq. (2.40), $q_v = 0$ that translates to $\hat{\mathbf{v}} = 0$ in the AMP equations. Linearizing the Bayes-optimal AMP equations for the Wigner and Wishart models eq. (4.3.2)-(4.3.1) around this trivial fixed point will lead to the LAMP spectral method. First, we detail the calculation for the simpler Wigner model, and then generalize the spectral algorithm in the Wishart case. Finally, we derive the state evolution associated to spectral method in the case of linear activation function.

## 5.1 Wigner model: $\mathbf{vv}^\mathsf{T}$

We start deriving the existence conditions of the trivial non-informative fixed point in the Wigner model eq. (1.1), that refers to eq. (2.41) in the main part. These conditions can be alternatively derived from the SE eqs. (6.8)-(6.9) - see sec. 6.

**Existence of the uninformative fixed point:** Consider $\hat{\mathbf{v}} = \mathbf{0}$. We obtain easily from the algorithm (4.3.2), $(\mathbf{B}_v, A_v) = (\mathbf{0}, 0)$, leading to $\mathbf{g} = f_{\text{out}}(\mathbf{0}, 0, \boldsymbol{\omega}, V) = \mathbb{E}_{Q_{\text{out}}^0}[(\mathbf{x} - \boldsymbol{\omega})] = \mathbf{0}$, and $(\boldsymbol{\gamma}, \Lambda) = (\mathbf{0}, 0)$. Finally, inserting these values in the update functions $f_{\text{out}}$ and $f_v$, defined in eq. (1.23), we obtain sufficient conditions to get the trivial fixed point in the Wigner model:

$$(\hat{\mathbf{v}}, \hat{\mathbf{z}}) = (\mathbf{0}, \mathbf{0}) \quad \text{if } \mathcal{C} \equiv \left\{ \ \mathbb{E}_{Q_{\text{out}}^0}[v] = 0 \ \text{ and } \ \mathbb{E}_{P_v}[z] = 0 \right\} . \tag{5.1}$$

**Linearization:** To lighten notation, we denote with $|_\star$ quantities that are evaluated at $(\mathbf{B}_v, A_v, \boldsymbol{\omega}, V, \boldsymbol{\gamma}, \Lambda) = (\mathbf{0}, 0, \mathbf{0}, \rho_z \mathbf{I}_p, \mathbf{0}, 0)$, and we linearize the equations of the AMP algorithm 4.3.2 around the fixed point

$$(\hat{\mathbf{v}}, \hat{\mathbf{c}}_v) = (\mathbf{0}, \rho_v \mathbf{I}_p), \quad (\hat{\mathbf{z}}, \hat{\mathbf{c}}_z) = (\mathbf{0}, \rho_z \mathbf{I}_k), \tag{5.2}$$

$$(\mathbf{B}_v, A_v) = (\mathbf{0}, 0), \quad (\boldsymbol{\gamma}, \Lambda) = (\mathbf{0}, 0), \quad (\boldsymbol{\omega}, V, \mathbf{g}) = (\mathbf{0}, \rho_z \mathbf{I}_p, \mathbf{0}) . \tag{5.3}$$

In a scalar formulation, the linearization yields

$$\delta\hat{\mathbf{v}}_i^{t+1} = \partial_B f_v|_\star \delta\mathbf{B}_i^{v,t} + \partial_A f_v|_\star \delta A_i^{v,t} + \partial_\omega f_v|_\star \delta\boldsymbol{\omega}_i^t + \partial_V f_v|_\star \delta V_i^t , \tag{5.4}$$

$$\delta\hat{c}_i^{v,t+1} = \partial_{B,B}^2 f_v|_\star \delta\mathbf{B}_i^{v,t} + \partial_{A,B}^2 f_v|_\star \delta A_i^{v,t} + \partial_{\omega,B}^2 f_v|_\star \delta\boldsymbol{\omega}_i^t + \partial_{V,B}^2 f_v|_\star \delta V_i^t , \tag{5.5}$$

$$\delta\hat{\mathbf{z}}_l^{t+1} = \partial_\gamma f_z|_\star \delta\boldsymbol{\gamma}_l^t + \partial_\Lambda f_z|_\star \delta\Lambda_l^t , \tag{5.6}$$

$$\delta\hat{c}_i^{z,t+1} = \partial_{\gamma,\gamma}^2 f_z|_\star \delta\boldsymbol{\gamma}_l^t + \partial_{\Lambda,\gamma}^2 f_z|_\star \delta\Lambda_l^t , \tag{5.7}$$

$$\delta\mathbf{g}_i^t = \partial_B f_{\text{out}}|_\star \delta\mathbf{B}_i^{v,t} + \partial_A f_{\text{out}}|_\star \delta A_i^{v,t} + \partial_\omega f_{\text{out}}|_\star \delta\boldsymbol{\omega}_i^t + \partial_V f_{\text{out}}|_\star \delta V_i^t , \tag{5.8}$$

with

$$\delta\mathbf{B}_i^{v,t} = \frac{1}{\Delta} \sum_{j=1}^p \frac{Y_{ji}}{\sqrt{p}} \delta\hat{\mathbf{v}}_j^t - \frac{1}{\Delta} \left( \sum_{j=1}^p \frac{\hat{c}_j^{v,t}|_\star}{p} \right) \delta\hat{\mathbf{v}}_i^{t-1} - \frac{1}{\Delta} \left( \sum_{j=1}^p \frac{\delta\hat{c}_j^{v,t}}{p} \right) \hat{\mathbf{v}}_i^{t-1}|_\star , \tag{5.9}$$

$$\delta A^{v,t} = \frac{2}{\Delta} \sum_{j=1}^p \frac{\hat{\mathbf{v}}_j^t|_\star \delta\hat{\mathbf{v}}_j^t}{p} = 0 , \tag{5.10}$$

$$\delta\boldsymbol{\omega}_i^t = \frac{1}{\sqrt{k}} \sum_{l=1}^k W_{il} \delta\hat{\mathbf{z}}_l^t - \delta V_i^t g_i^{t-1}|_\star - V_i^t|_\star \delta\mathbf{g}_i^{t-1} , \tag{5.11}$$

$$\delta V^t = \frac{1}{k} \sum_{l=1}^k \delta\hat{c}_l^{z,t} , \tag{5.12}$$

$$\delta\Lambda^t = \frac{2}{k} \sum_{i=1}^p \mathbf{g}_i^t|_\star \delta\mathbf{g}_i^t = 0 , \tag{5.13}$$

$$\delta\boldsymbol{\gamma}_l^t = \frac{1}{\sqrt{k}} \sum_{i=1}^p W_{il} \delta\mathbf{g}_i^t + \delta\Lambda_l^t \hat{\mathbf{z}}_l^t|_\star + \Lambda_l^t|_\star \delta\hat{\mathbf{z}}_l^t . \tag{5.14}$$

These equations can be simplified and closed over three vectorial variables $\hat{\mathbf{v}} \in \mathbb{R}^p$, $\hat{\mathbf{z}} \in \mathbb{R}^k$ and $\boldsymbol{\omega} \in \mathbb{R}^p$, where we used the existence condition $\mathcal{C}$ that leads to $\partial_\omega f_{\text{out}}|_\star = \partial_V f_{\text{out}}|_\star = 0$. Finally, injecting eq. (5.9)-(5.14) in (5.4), (5.6), (5.11) we obtain

$$\delta \hat{\mathbf{v}}^{t+1} = \frac{1}{\Delta} \partial_B f_v|_\star \left( \frac{Y}{\sqrt{p}} \delta \hat{\mathbf{v}}^t - \partial_B f_v|_\star \mathrm{I}_p \delta \hat{\mathbf{v}}^{t-1} \right) + \partial_\omega f_v|_\star \mathrm{I}_p \delta \boldsymbol{\omega}^t + \frac{\partial_V f_v|_\star \partial_{\gamma,\gamma}^2 f_z|_\star}{\partial_\gamma f_z|_\star} \frac{\mathbb{1}_p \mathbb{1}_k^\intercal}{k} \delta \hat{\mathbf{z}}^t ,$$
(5.15)

$$\delta \hat{\mathbf{z}}^{t+1} = \frac{1}{\Delta} \partial_\gamma f_z|_\star \partial_B f_{\text{out}}|_\star \frac{W^\intercal}{\sqrt{k}} \left[ \frac{Y}{\sqrt{p}} \delta \hat{\mathbf{v}}^t - \partial_B f_v|_\star \mathrm{I}_p \delta \hat{\mathbf{v}}^{t-1} \right] ,$$
(5.16)

$$\delta \boldsymbol{\omega}^{t+1} = \frac{1}{\Delta} \left( \partial_\gamma f_z|_\star \partial_B f_{\text{out}}|_\star \frac{WW^\intercal}{k} \left[ \frac{Y}{\sqrt{p}} \delta \hat{\mathbf{v}}^t - \partial_B f_v|_\star \mathrm{I}_p \delta \hat{\mathbf{v}}^{t-1} \right] \right) -$$
(5.17)

$$\partial_\gamma f_z|_\star \partial_B f_{\text{out}}|_\star \left[ \frac{Y}{\sqrt{p}} \delta \hat{\mathbf{v}}^{t-1} - \partial_B f_v|_\star \mathrm{I}_p \delta \hat{\mathbf{v}}^{t-2} \right] .$$

**Conclusion:** This set of equations involves partial derivatives of $f_v$, $f_z$ and $f_{\text{out}}$ that can be simplified using the condition $\mathcal{C}$, and rewritten as moments of the distributions $P_z$ and $Q_{\text{out}}$:

$$\begin{cases} \partial_\gamma f_z|_\star & = \mathbb{E}_{P_z} \left[ z^2 \right] = \rho_z , \\ \partial_{\gamma,\gamma}^2 f_z|_\star & = -2 \partial_\Lambda f_z|_\star = \mathbb{E}_{P_z} \left[ z^3 \right] , \\ \partial_\omega f_{\text{out}}|_\star & = \partial_V f_{\text{out}}|_\star = 0 , \end{cases} \quad \text{and} \quad \begin{cases} \partial_B f_v|_\star & = \mathbb{E}_{Q_{\text{out}}^0} [v^2] = \rho_v , \\ \partial_\omega f_v|_\star & = \partial_B f_{\text{out}}|_\star = \rho_z^{-1} \mathbb{E}_{Q_{\text{out}}^0} [vx] , \\ \partial_V f_v|_\star & = \frac{1}{2} \rho_z^{-2} \mathbb{E}_{Q_{\text{out}}^0} [vx^2] . \end{cases}$$
(5.18)

Injecting eq. (5.17)-(5.16) in (5.15), we finally obtain a closed equation over $\hat{\mathbf{v}}$. Forgetting time indices, it leads the definition of the LAMP operator as

$$\Gamma_p^{vv} = \frac{1}{\Delta} \left( (a-b)\mathrm{I}_p + b\frac{WW^\intercal}{k} + c\frac{\mathbb{1}_p \mathbb{1}_k^\intercal}{k} \frac{W^\intercal}{\sqrt{k}} \right) \times \left( \frac{Y}{\sqrt{p}} - a\mathrm{I}_p \right) ,$$
(5.19)

with

$$a \equiv \mathbb{E}_{Q_{\text{out}}^0} [v^2] = \rho_v , \quad b \equiv \rho_z^{-1} \mathbb{E}_{Q_{\text{out}}^0} [vx]^2 , \quad c \equiv \frac{1}{2} \rho_z^{-3} \mathbb{E}_{P_z} \left[ z^3 \right] \mathbb{E}_{Q_{\text{out}}^0} [vx^2] \mathbb{E}_{Q_{\text{out}}^0} [vx] . \quad (5.20)$$

Note that in most of the cases we studied, the parameter $c$, taking into account the skewness of the variable $\mathbf{z}$, is zero, simplifying considerably the structured matrix as discussed in the main part. Taking the leading eigenvector of the operator $\Gamma_p^{vv}$ leads to the LAMP algorithm.

**Applications:** Consider a gaussian $P_z = \mathcal{N}_z (0, 1)$ or binary $P_z = \frac{1}{2} (\delta(z-1) + \delta(z+1))$ prior, for which $\rho_z = 1$. Taking a noiseless channel $P_{\text{out}}(v|x) = \delta(v - \varphi(x))$, condition $\mathcal{C}$ is verified, and we obtain simple and explicit coefficients

- Linear activation ($\varphi(x) = x$): $(a, b, c) = (1, 1, 0)$.
- Sign activation ($\varphi(x) = \text{sgn}(x)$): $(a, b, c) = (1, 2/\pi, 0)$.

## 5.2 Wishart model: $\mathbf{uv}^\intercal$

In this section, we generalize the previous derivation of the LAMP spectral algorithm for the Wishart model in eq. (1.2). The strategy is exactly the same: it follows from linearizing the AMP algorithm 4.3.1 in its Bayes-optimal version around the trivial fixed point. Except that in this case there are more equations to deal with.

**Existence of the uninformative fixed point:** Consider $(\hat{\mathbf{u}}, \hat{\mathbf{v}}) = (\mathbf{0}, \mathbf{0})$. Injecting this condition in the algorithm's equations, we simply obtain $(\mathbf{B}_u, A_u, \mathbf{B}_v, A_v) = (\mathbf{0}, 0, \mathbf{0}, 0)$. However, we now need $\mathbb{E}_{P_u} [u] = 0$ for this to be consistent with the update equation for $\hat{\mathbf{u}}^{t+1}$. Besides, this also implies $\mathbf{g} = f_{\text{out}}(\mathbf{0}, 0, \boldsymbol{\omega}, V) = \mathbb{E}_{Q_{\text{out}}^0} [(\mathbf{x} - \boldsymbol{\omega})] = \mathbf{0}$, and $(\gamma, \Lambda) = (\mathbf{0}, 0)$. Finally, putting all conditions

together in the update equations involving $f_v$, $f_u$ and $f_{\text{out}}$, defined in eq. (1.23), we arrive at the following sufficient conditions for the existence of the uninformative fixed point in the Wishart model:

$$(\hat{\mathbf{v}}, \hat{\mathbf{z}}) = (\mathbf{0}, \mathbf{0}) \quad \text{if } \mathcal{C} \equiv \left\{ \quad \mathbb{E}_{Q_{\text{out}}^0}[v] = 0, \quad \mathbb{E}_{P_v}[z] = 0 \quad \text{and} \quad \mathbb{E}_{P_u}[u] = 0 \right\}. \tag{5.21}$$

**Linearization:** As previously, to lighten notations we denote $|_\star$ quantities that are evaluated at $(\mathbf{B}_u, A_u, \mathbf{B}_v, A_v, \boldsymbol{\omega}, V, \boldsymbol{\gamma}, \Lambda) = (\mathbf{0}, 0, \mathbf{0}, 0, \mathbf{0}, \rho_z \mathrm{I}_p, \mathbf{0}, 0)$. We linearize AMP equations algorithm 4.3.1 around the fixed point

$$(\hat{\mathbf{u}}, \hat{\mathbf{c}}_u) = (\mathbf{0}, \rho_u \mathrm{I}_n), \quad (\hat{\mathbf{v}}, \hat{\mathbf{c}}_v) = (\mathbf{0}, \rho_v \mathrm{I}_p), \quad (\hat{\mathbf{z}}, \hat{\mathbf{c}}_z) = (\mathbf{0}, \rho_z \mathrm{I}_k), \tag{5.22}$$

$$(\mathbf{B}_u, A_u) = (\mathbf{0}, 0), \quad (\mathbf{B}_v, A_v) = (\mathbf{0}, 0), \quad (\boldsymbol{\gamma}, \Lambda) = (\mathbf{0}, 0), \quad (\boldsymbol{\omega}, V, \mathbf{g}) = (\mathbf{0}, \rho_z \mathrm{I}_p, \mathbf{0}). \tag{5.23}$$

In a scalar formulation, linearization yields four additional equations over the **u** variable:

$$\delta \hat{\mathbf{u}}_\mu^{t+1} = \partial_B f_u|_\star \delta \mathbf{B}_\mu^{u,t} + \partial_A f_u|_\star \delta A_\mu^{u,t}, \tag{5.24}$$

$$\delta \hat{c}_\mu^{u,t+1} = \partial_{B,B}^2 f_u|_\star \delta \mathbf{B}_\mu^{u,t} + \partial_{A,B}^2 f_u|_\star \delta A_\mu^{u,t}, \tag{5.25}$$

$$\delta \hat{\mathbf{v}}_i^{t+1} = \partial_B f_v|_\star \delta \mathbf{B}_i^{v,t} + \partial_A f_v|_\star \delta A_i^{v,t} + \partial_\omega f_v|_\star \delta \boldsymbol{\omega}_i^t + \partial_V f_v|_\star \delta V_i^t, \tag{5.26}$$

$$\delta \hat{c}_i^{v,t+1} = \partial_{B,B}^2 f_v|_\star \delta \mathbf{B}_i^{v,t} + \partial_{A,B}^2 f_v|_\star \delta A_i^{v,t} + \partial_{\omega,B}^2 f_v|_\star \delta \boldsymbol{\omega}_i^t + \partial_{V,B}^2 f_v|_\star \delta V_i^t, \tag{5.27}$$

$$\delta \hat{\mathbf{z}}_l^{t+1} = \partial_\gamma f_z|_\star \delta \boldsymbol{\gamma}_l^t + \partial_\Lambda f_z|_\star \delta \Lambda_l^t, \tag{5.28}$$

$$\delta \hat{c}_i^{z,t+1} = \partial_{\gamma,\gamma}^2 f_z|_\star \delta \boldsymbol{\gamma}_l^t + \partial_{\Lambda,\gamma}^2 f_z|_\star \delta \Lambda_l^t, \tag{5.29}$$

$$\delta \mathbf{g}_i^t = \partial_B f_{\text{out}}|_\star \delta \mathbf{B}_i^{v,t} + \partial_A f_{\text{out}}|_\star \delta A_i^{v,t} + \partial_\omega f_{\text{out}}|_\star \delta \boldsymbol{\omega}_i^t + \partial_V f_{\text{out}}|_\star \delta V_i^t, \tag{5.30}$$

and

$$\delta \mathbf{B}_\mu^{u,t} = \frac{1}{\Delta} \sum_{i=1}^p \frac{Y_{\mu i}}{\sqrt{p}} \delta \hat{\mathbf{v}}_i^t - \frac{1}{\Delta} \left( \sum_{i=1}^p \frac{\hat{c}_i^{v,t}|_\star}{p} \right) \delta \hat{\mathbf{u}}_\mu^{t-1} - \frac{1}{\Delta} \left( \sum_{i=1}^p \frac{\delta \hat{c}_i^{v,t}}{p} \right) \hat{\mathbf{u}}_\mu^{t-1}|_\star, \tag{5.31}$$

$$\delta A^{u,t} = \frac{2}{\Delta} \sum_{i=1}^p \frac{\hat{\mathbf{v}}_i^t|_\star \delta \hat{\mathbf{v}}_i^t}{p} = 0, \tag{5.32}$$

$$\delta \mathbf{B}_i^{v,t} = \frac{1}{\Delta} \sum_{\mu=1}^n \frac{Y_{\mu i}}{\sqrt{p}} \delta \hat{\mathbf{u}}_\mu^t - \frac{1}{\Delta} \left( \sum_{\mu=1}^n \frac{\hat{c}_\mu^{u,t}|_\star}{p} \right) \delta \hat{\mathbf{v}}_i^{t-1} - \frac{1}{\Delta} \left( \sum_{\mu=1}^n \frac{\delta \hat{c}_\mu^{u,t}}{p} \right) \hat{\mathbf{v}}_i^{t-1}|_\star, \tag{5.33}$$

$$\delta A^{v,t} = \frac{2}{\Delta} \sum_{\mu=1}^n \frac{\hat{\mathbf{u}}_\mu^t|_\star \delta \hat{\mathbf{u}}_\mu^t}{p} = 0, \tag{5.34}$$

$$\delta \boldsymbol{\omega}_i^t = \frac{1}{\sqrt{k}} \sum_{l=1}^k W_{il} \delta \hat{\mathbf{z}}_l^t - \delta V_i^t \mathbf{g}_i^{t-1}|_\star - V_i^t|_\star \delta \mathbf{g}_i^{t-1}, \tag{5.35}$$

$$\delta V^t = \frac{1}{k} \sum_{l=1}^k \delta \hat{c}_l^{z,t}, \tag{5.36}$$

$$\delta \Lambda^t = \frac{2}{k} \sum_{i=1}^p g_i^t|_\star \delta \mathbf{g}_i^t = 0, \tag{5.37}$$

$$\delta \boldsymbol{\gamma}_l^t = \frac{1}{\sqrt{k}} \sum_{i=1}^p W_{il} \delta \mathbf{g}_i^t + \delta \Lambda_l^t \hat{\mathbf{z}}_l^t|_\star + \Lambda_l^t|_\star \delta \hat{\mathbf{z}}_l^t. \tag{5.38}$$

These equations can be closed over four vectorial variables $\hat{\mathbf{u}} \in \mathbb{R}^n$, $\hat{\mathbf{v}} \in \mathbb{R}^p$, $\hat{\mathbf{z}} \in \mathbb{R}^k$ and $\boldsymbol{\omega} \in \mathbb{R}^p$, where we used the existence condition $\mathcal{C}$ leading again to $\partial_\omega f_{\text{out}}|_\star = \partial_V f_{\text{out}}|_\star = 0$. Finally, injecting

eq. (5.31)-(5.38) in (5.24), (5.26), (5.28), (5.35) we obtain:

$$\delta\hat{\mathbf{u}}^{t+1} = \frac{1}{\Delta}\partial_B f_u|_\star \left( \frac{Y}{\sqrt{p}}\delta\hat{\mathbf{v}}^t - \partial_B f_v|_\star \mathrm{I}_n \delta\hat{\mathbf{u}}^{t-1} \right) , \tag{5.39}$$

$$\delta\hat{\mathbf{v}}^{t+1} = \frac{1}{\Delta}\partial_B f_v|_\star \left( \frac{Y^\intercal}{\sqrt{p}}\delta\hat{\mathbf{u}}^t - \beta\partial_B f_u|_\star \mathrm{I}_p \delta\hat{\mathbf{v}}^{t-1} \right) + \partial_\omega f_v|_\star \mathrm{I}_p \delta\boldsymbol{\omega}^t + \frac{\partial_V f_v|_\star \partial^2_{\gamma,\gamma} f_z|_\star}{\partial_\gamma f_z|_\star}\frac{\mathbb{1}_p \mathbb{1}_k^\intercal}{k}\delta\hat{\mathbf{z}}^t , \tag{5.40}$$

$$\delta\hat{\mathbf{z}}^{t+1} = \frac{1}{\Delta}\partial_\gamma f_z|_\star \partial_B f_{\text{out}}|_\star \frac{W^\intercal}{\sqrt{k}} \left[ \frac{Y^\intercal}{\sqrt{p}}\delta\hat{\mathbf{u}}^t - \beta\partial_B f_u|_\star \mathrm{I}_p \delta\hat{\mathbf{v}}^{t-1} \right] , \tag{5.41}$$

$$\delta\boldsymbol{\omega}^{t+1} = \frac{1}{\Delta} \left( \partial_\gamma f_z|_\star \partial_B f_{\text{out}}|_\star \frac{W^\intercal}{\sqrt{k}} \left[ \frac{Y^\intercal}{\sqrt{p}}\delta\hat{\mathbf{u}}^t - \beta\partial_B f_u|_\star \mathrm{I}_p \delta\hat{\mathbf{v}}^{t-1} \right] \right) - \tag{5.42}$$
$$\partial_\gamma f_z|_\star \partial_B f_{\text{out}}|_\star \left[ \frac{Y^\intercal}{\sqrt{p}}\delta\hat{\mathbf{u}}^{t-1} - \beta\partial_B f_u|_\star \mathrm{I}_p \delta\hat{\mathbf{v}}^{t-2} \right] .$$

**Conclusion:** This set of equations involves partial derivatives of $f_u$, $f_v$, $f_{\text{out}}$ that can be simplified using the condition $\mathcal{C}$ and rewritten as moments of distributions $P_u$, $P_z$ and $Q_{\text{out}}$:

$$\begin{cases} \partial_\gamma f_z|_\star &= \mathbb{E}_{P_z}\left[z^2\right] = \rho_z , \\ \partial^2_{\gamma,\gamma} f_z|_\star &= -2\partial_\Lambda f_z|_\star = \mathbb{E}_{P_z}\left[z^3\right] , \\ \partial_\omega f_{\text{out}}|_\star &= \partial_V f_{\text{out}}|_\star = 0 , \\ \partial_B f_u|_\star &= \mathbb{E}_{P_u}[u^2] = \rho_u , \end{cases} \quad \text{and} \quad \begin{cases} \partial_B f_v|_\star &= \mathbb{E}_{Q^0_{\text{out}}}[v^2] = \rho_v , \\ \partial_\omega f_v|_\star &= \partial_B f_{\text{out}}|_\star = \rho_z^{-1}\mathbb{E}_{Q^0_{\text{out}}}[vx] , \\ \partial_V f_v|_\star &= \tfrac{1}{2}\rho_z^{-2}\mathbb{E}_{Q^0_{\text{out}}}[vx^2] . \end{cases} \tag{5.43}$$

Injecting eq. (5.42),(5.41)-(5.39) in (5.40), we finally obtain a self-consistent equation over $\hat{\mathbf{v}}$ that, forgetting time indices, leads to define the following LAMP structured matrix, from which we need to compute the top eigenvector:

$$\Gamma^{uv}_p = \frac{1}{\Delta} \left( (a-b)\mathrm{I}_p + b\frac{WW^\intercal}{k} + c\frac{\mathbb{1}_p \mathbb{1}_k^\intercal}{k}\frac{W^\intercal}{\sqrt{k}} \right) \times \left( \frac{1}{a+\frac{\Delta}{d}}\frac{Y^\intercal Y}{p} - d\beta\mathrm{I}_p \right) , \tag{5.44}$$

with

$$a \equiv \rho_v , \quad c \equiv \frac{1}{2}\rho_z^{-3}\mathbb{E}_{P_z}\left[z^3\right]\mathbb{E}_{Q^0_{\text{out}}}[vx^2]\mathbb{E}_{Q^0_{\text{out}}}[vx], \quad b \equiv \rho_z^{-1}\mathbb{E}_{Q^0_{\text{out}}}[vx]^2 , \quad d \equiv \rho_u . \tag{5.45}$$

**Applications:** Consider a gaussian $P_z, P_u = \mathcal{N}(0,1)$ or binary $P_z, P_u = \frac{1}{2}(\delta(z-1)+\delta(z+1))$ prior, for which $\rho_z = \rho_u = 1$. For a noiseless channel $P_{\text{out}}(v|x) = \delta(v-\varphi(x))$, we obtain the following simple and explicit coefficients:

- Linear, $\varphi(x) = x$: $(a,b,c,d) = (1,1,0,1)$
- Sign, $\varphi(x) = \text{sgn}(x)$: $(a,b,c,d) = (1,2/\pi,0,1)$

### 5.3 State evolution equations of LAMP and PCA - linear case

In this section we describe how to obtain the limiting behaviour of the LAMP spectral method for the Wigner model in the large size limit $p \to \infty$. We will show that in the linear case, mean squared errors of LAMP and PCA are directly obtained from the optimal overlap performed by AMP or its state evolution. Recall that the numerical simulations of LAMP and PCA are compared with their state evolution in Fig. 3, with green and red lines respectively.

**LAMP:** For the noiseless linear channel $P_{\text{out}}(v|x) = \delta(v-x)$, the set of eqs. (5.15-5.17) are already linear, and do not require linearizing as above. Hence the LAMP spectral method flows directly from the AMP eqs. (4.3.2). As a consequence, this means that the state evolution equations

associated to the spectral method are simply dictated by the set of AMP state evolution equations from sec. 5.1. However, it is worth stressing that the LAMP MSE is not given by the AMP mean squared error, as LAMP returns a normalized estimator. We now compute the overlaps and mean squared error performed by this spectral algorithm.

Recall that $m_v$ and $q_v$ are the parameters defined in eq. (4.7), that respectively measure the overlap between the ground truth $\mathbf{v}^\star$ and the estimator $\hat{\mathbf{v}}$, and the norm of the estimator. In eq. (3.49), the MSE is given by:

$$\text{MSE}_v = \rho_v + \mathbb{E}_{\mathbf{v}^\star} \lim_{p \to \infty} \frac{1}{p} \|\hat{\mathbf{v}}\|_2^2 - 2\mathbb{E}_{\mathbf{v}^\star} \lim_{p \to \infty} \frac{1}{p} \hat{\mathbf{v}}^\mathsf{T} \mathbf{v}^\star \tag{5.46}$$

$$= \rho_v + q_v - 2m_v \,, \tag{5.47}$$

However the LAMP spectral method computes the normalized top eigenvector of the structured matrix $\Gamma_p$. Hence the norm of the LAMP estimator is $\|\hat{\mathbf{v}}\|_{\text{LAMP}}^2 = q_{v,\text{LAMP}} = 1$, while the Bayes-optimal AMP estimator is not normalized with $\|\hat{\mathbf{v}}\|_{\text{AMP}}^2 = q_{v,\text{AMP}}^\star = m_{v,\text{AMP}}^\star \neq 1$, solutions of eq. (5.1). As the non-normalized LAMP estimator follows AMP state evolutions in the linear case, the overlap with the ground truth is thus given by:

$$m_{v,\text{LAMP}} \equiv \mathbb{E}_{\mathbf{v}^\star} \lim_{p \to \infty} \frac{1}{p} \hat{\mathbf{v}}_{\text{LAMP}}^\mathsf{T} \mathbf{v}^\star = \mathbb{E}_{\mathbf{v}^\star} \lim_{p \to \infty} \frac{1}{p} \left( \frac{\hat{\mathbf{v}}_{\text{AMP}}}{\|\hat{\mathbf{v}}\|_{\text{AMP}}} \right)^\mathsf{T} \mathbf{v}^\star \tag{5.48}$$

$$= \frac{m_{v,\text{AMP}}^\star}{\left( q_{v,\text{AMP}}^\star \right)^{1/2}} = \left( m_{v,\text{AMP}}^\star \right)^{1/2} \,. \tag{5.49}$$

Finally the mean squared error performed by the LAMP method is easily obtained from the optimal overlap reached by the AMP algorithm and yields

$$\text{MSE}_{v,\text{LAMP}} = \rho_v + 1 - 2 \left( q_{v,\text{AMP}}^\star \right)^{1/2} \,. \tag{5.50}$$

**PCA:**  Similarly, in the noiseless linear channel case, we note that at $\alpha = 0$, LAMP reduces exactly to PCA, i.e. it consists in finding the top eigenvector of $Y$, instead $\Gamma_p$. As LAMP follows AMP in this case, we can simply state that the mean squared error performed by PCA is computed using the optimal overlap reached by AMP at $\alpha = 0$:

$$\text{MSE}_{v,\text{PCA}} = \rho_v + 1 - 2 \left( q_{v,\text{AMP}}^\star|_{\alpha=0} \right)^{1/2} \,. \tag{5.51}$$

# 6 Transition from state evolution - stability

In this section we derive sufficient conditions for the existence of the uninformative fixed point $(q_v, \hat{q}_z, q_z) = (0, 0, 0)$ from the state evolution eqs. (15). In the case $(0, 0, 0)$ is a fixed point, we derive its stability, obtaining the Jacobian in eq. (2.42). Its eigenvalues determine the regions for which $(0, 0, 0)$ is stable and unstable, and therefore the critical point $\Delta_c$ where the transition occurs.

For the purpose of our analysis we define the following shorthand notation for the update functions,

$$\mathbf{f}(r, t, s) \equiv \begin{pmatrix} f_1(r, s) \\ f_2(r, s) \\ f_3(t) \end{pmatrix} \tag{6.1}$$

where $(f_1, f_2, f_3)$ are explicitly given by

$$f_1(r, s) = 2\partial_r \Psi_{\text{out}}(r, s) = \mathbb{E}_{\xi, \eta} \left[ \frac{\left( \int dv \, e^{-\frac{r}{2}v^2 + \sqrt{r}v\xi} \int \frac{dx}{\sqrt{2\pi(\rho_z - s)}} e^{-\frac{1}{2}\frac{(x - \sqrt{s}\eta)^2}{\rho_z - s}} P_{\text{out}}(v|x)v \right)^2}{\int dv \, e^{-\frac{r}{2}v^2 + \sqrt{r}v\xi} \int \frac{dx}{\sqrt{2\pi(\rho_z - s)}} e^{-\frac{1}{2}\frac{(x - \sqrt{s}\eta)^2}{\rho_z - s}} P_{\text{out}}(v|x)} \right]$$

$$f_2(r, s) = 2\alpha\partial_s \Psi_{\text{out}}(r, s) = \alpha\mathbb{E}_{\xi, \eta} \left[ \frac{\left( \int dv \, e^{-\frac{r}{2}v^2 + \sqrt{r}v\xi} \int \frac{dx}{\sqrt{2\pi(\rho_z - s)}} e^{-\frac{1}{2}\frac{(x - \sqrt{s}\eta)^2}{\rho_z - s}} P_{\text{out}}(v|x)(x - \sqrt{s}\eta) \right)^2}{\int dv \, e^{-\frac{r}{2}v^2 + \sqrt{r}v\xi} \int \frac{dx}{\sqrt{2\pi(\rho_z - s)}} e^{-\frac{1}{2}\frac{(x - \sqrt{s}\eta)^2}{\rho_z - s}} P_{\text{out}}(v|x)} \right]$$

$$f_3(t) = 2\partial_t \Psi_z(t) = \mathbb{E}_{\xi} \left[ \frac{\left( \int dx \, P_z(z) e^{-\frac{t}{2}z^2 + \sqrt{t}z\xi} z \right)^2}{\int dx \, P_z(z) e^{-\frac{t}{2}z^2 + \sqrt{t}\xi z}} \right] \tag{6.2}$$

In terms of these, the right-hand side of the state evolution equations is given by evaluating $(r, t, s) = \left( \frac{q_v}{\Delta}, \hat{q}_z, q_z \right)$.

## 6.1 Conditions for fixed point

Note that the denominator in the first two state evolution equations is actually constant at $r = 0$,

$$\int dv \int \frac{dx}{\sqrt{2\pi\rho_z}} e^{-\frac{1}{2\rho_z}x^2} P_{\text{out}}(v|x) = \int \frac{dx}{\sqrt{2\pi\rho_z}} e^{-\frac{1}{2\rho_z}x^2} \left( \int dv \, P_{\text{out}}(v|x) = \int \frac{dx}{\sqrt{2\pi\rho_z}} e^{-\frac{1}{2\rho_z}x^2} \right) = 1. \tag{6.3}$$

And in particular, this means that

$$\begin{aligned}
f_2(0, s) &= \mathbb{E}_{\xi, \eta} \left( \int dv \int \frac{dx}{\sqrt{2\pi\rho_z}} e^{-\frac{1}{2\rho_z}x^2} P_{\text{out}}(v|x) \left( x - \sqrt{s}\eta \right) \right)^2 \\
&= \mathbb{E}_{\xi, \eta} \left( \int \frac{dx}{\sqrt{2\pi\rho_z}} e^{-\frac{1}{2\rho_z}x^2} \left( x - \sqrt{s}\eta \right) \int dv \, P_{\text{out}}(v|x) \right)^2 \\
&= \mathbb{E}_{\xi, \eta} \left( \int \frac{dx}{\sqrt{2\pi\rho_z}} e^{-\frac{1}{2\rho_z}x^2} \left( x - \sqrt{s}\eta \right) \right)^2 = 0
\end{aligned} \tag{6.4}$$

for any value of $s \in \mathbb{R}$. In terms of the overlaps, this means that if $q_u$ is a fixed point, we necessarily have $\hat{q}_z = 0$. What is the implication for $q_z$? We need to look at $f_3(\hat{q}_z = 0)$, which is simply given by

$$f_3(0) = \mathbb{E}_{\xi} \left( \int dx \, P_z z \right)^2. \tag{6.5}$$

This means that if $q_u = 0$ and $P_z$ has zero mean, then $q_z = 0$. It remains to check what is a sufficient condition for $q_u = 0$ to be a fixed point. This is the case if

$$f_1(0, 0) = \mathbb{E}_{\xi, \eta} \left( \int dv \int \frac{dx}{\sqrt{2\pi\rho_z}} e^{-\frac{1}{2\rho_z}x^2} P_{\text{out}}(v|x)v \right)^2 \stackrel{!}{=} 0 \tag{6.6}$$

implying

$$\int \mathrm{d}v \int \frac{\mathrm{d}x}{\sqrt{2\pi\rho_z}} e^{-\frac{1}{2\rho_z}x^2} P_{\text{out}}(v|x)v = \int \frac{\mathrm{d}x}{\sqrt{2\pi\rho_z}} e^{-\frac{1}{2\rho_z}x^2} \left( \int \mathrm{d}v \, P_{\text{out}}(v|x)v \right) \stackrel{!}{=} 0 \qquad (6.7)$$

Therefore a set of sufficient conditions for $(q_u, \hat{q}_z, q_z) = (0,0,0)$ to be a fixed point of the state evolution equations are

$$\mathbb{E}_{P_z} z = \int \mathrm{d}x \, P_z(z)z = 0 \qquad (6.8)$$

$$\mathbb{E}_{Q_{\text{out}}^0} v = \int \mathrm{d}v \int \frac{\mathrm{d}x}{\sqrt{2\pi\rho_z}} e^{-\frac{1}{2\rho_z}x^2} P_{\text{out}}(v|x)v = 0 \qquad (6.9)$$

note that the last condition is equivalent to requiring the function $m(x) = \mathbb{E}_{P_{\text{out}}} v$ to be odd.

## 6.2 Stability analysis

We now study the stability of the fixed point $(r, t, s) = (0, 0, 0)$, which is determined by the linearisation of the state evolution equations. But before, to help in the analysis we introduce notation.

**Some notation**   It will be useful to introduce the following notation for the denoising functions in eq. (1.16) evaluated at the overlaps:

$$Q_{\text{out}}^{(r,s)}(v, x; \xi, \eta) = \frac{1}{\mathcal{Z}_{\text{out}}^{(r,s)}(\xi, \eta)} e^{-\frac{r}{2}u^2 + \sqrt{r}\xi u} \frac{1}{\sqrt{2\pi(\rho_z - s)}} e^{-\frac{1}{2}\frac{(x - \sqrt{s}\eta)^2}{\rho_z - s}} P_{\text{out}}(v|x) \qquad (6.10)$$

$$Q_z^t(z; \xi) = \frac{1}{\mathcal{Z}_z^t(\xi)} e^{-\frac{t}{2}z^2 + \sqrt{t}\xi z} P_z(z) \qquad (6.11)$$

where $\mathcal{Z}_{\text{out}}^{(r,s)}$ and $\mathcal{Z}_z$ are the normalisation of the distributions, given explicitly by

$$\mathcal{Z}_{\text{out}}^{(r,s)}(\xi, \eta) = \int \mathrm{d}v \, e^{-\frac{r}{2}v^2 + \sqrt{r}v\xi} \int \frac{\mathrm{d}x}{\sqrt{2\pi(\rho_z - s)}} e^{-\frac{1}{2}\frac{(x - \sqrt{s}\eta)^2}{\rho_z - s}} P_{\text{out}}(v|x)$$

$$\mathcal{Z}_z^t(\xi) = \int \mathrm{d}x \, Q_z^t(z; \xi) = \int \mathrm{d}x \, P_z(z) e^{-\frac{t}{2}z^2 + \sqrt{t}\xi z} \qquad (6.12)$$

Note that $Q_{\text{out}}$ is a family of joint distributions over $(v, x)$, indexed by $r, s \in [0, 1]$. It will be useful to have in mind the following particular cases,

$$Q_{\text{out}}^{(0,s)}(v, x; \eta) = \frac{1}{\sqrt{2\pi(\rho_z - s)}} e^{-\frac{1}{2}\frac{(x - \sqrt{s}\eta)^2}{\rho_z - s}} P_{\text{out}}(v|x) \qquad (6.13)$$

$$Q_{\text{out}}^{(r,0)}(v, x; \xi) = \frac{1}{\mathcal{Z}_{\text{out}}^{(r,0)}(\xi, \eta)} e^{-\frac{r}{2}v^2 + \sqrt{r}v\xi} \frac{1}{\sqrt{2\pi\rho_z}} e^{-\frac{1}{2\rho_z}x^2} \qquad (6.14)$$

where we have used that $\mathcal{Z}_{\text{out}}^{(0,s)}(\eta, \xi) = 1$ (as shown above). It is also useful to define short hands to the associated distributions when we evaluate both $(r, s) = (0, 0)$,

$$Q_{\text{out}}^0(v, x) = Q_{\text{out}}^{(0,0)}(v, x; \xi, \eta) = \frac{1}{\sqrt{2\pi\rho_z}} e^{-\frac{1}{2\rho_z}x^2} P_{\text{out}}(v|x) \qquad (6.15)$$

while $Q_z^0(z; \xi) = P_z(z)$. Note that they are indeed independent of the noises, and that in particular we have $\mathcal{Z}_z^0(\xi) = 1$.

In this notation the condition in eq. (6.9) simply reads that $v$ has mean zero with respect to the $Q_{\text{out}}^0$,

$$\mathbb{E}_{Q_{\text{out}}^0} v = 0 \qquad (6.16)$$

**Expansion around the fixed point**

We now suppose $(r, t, s) = (0, 0, 0)$ is a fixed point of the state evolution equations, i.e. that the conditions in eqs. (6.8) and (6.9) hold. We are interested in the leading order expansion of the update functions $(f_1, f_2, f_3)$ around this point.

**Expansion of $f_1$:** Since $(f_1, f_2)$ are functions of $(r, s)$ only, we look them separately first. Instead of expanding around $(r, s) = (0, 0)$ together, we first expand around $r = 0$ keeping $s$ fixed. This allow us to take the average over $\xi$ explicitly simplifying the expansion considerably,

$$f_1(r, s) \underset{r \ll 1}{=} \mathbb{E}_\eta \left\{ \left( \mathbb{E}_{Q_{\text{out}}^{(0,s)}} v \right)^2 + \left[ \left( \mathbb{E}_{Q_{\text{out}}^{(0,s)}} v \right)^4 + \left( \mathbb{E}_{Q_{\text{out}}^{(0,s)}} v^2 \right)^2 - 2 \left( \mathbb{E}_{Q_{\text{out}}^{(0,s)}} v \right)^2 \mathbb{E}_{Q_{\text{out}}^{(0,s)}} v^2 \right] r + O\left( r^{3/2} \right) \right\}$$
(6.17)

We can now focus on the leading order expansion around $s = 0$. Note we have,

$$\mathbb{E}_{Q_{\text{out}}^{(0,s)}} v = \int \mathrm{d}v \int \frac{\mathrm{d}x}{\sqrt{2\pi(\rho_z - s)}} e^{-\frac{1}{2} \frac{(x - \sqrt{s}\eta)^2}{\rho_z - s}} P_{\text{out}}(v|x)\, v$$

$$\underset{s \ll 1}{=} \mathbb{E}_{\rho_0^v} v + \frac{\sqrt{s}\eta}{\rho_z} \mathbb{E}_{Q_{\text{out}}^0} vx - \frac{s}{2} \frac{\eta^2 - 1}{\rho_z^2} \left( \rho_z \mathbb{E}_{Q_{\text{out}}^0} v - \mathbb{E}_{Q_{\text{out}}^0} x^2 v \right) + O\left( s^{3/2} \right) \tag{6.18}$$

$$= \frac{\sqrt{s}\eta}{\rho_z} \mathbb{E}_{Q_{\text{out}}^0} vx + \frac{s}{2} \frac{\eta^2 - 1}{\rho_z^2} \mathbb{E}_{Q_{\text{out}}^0} x^2 v + O\left( s^{3/2} \right) \tag{6.19}$$

where we used the consistency condition in eq. (6.16) that ensures $(r, s) = (0, 0)$ is indeed a fixed point. Moreover, the leading order term in the expansion of $\mathbb{E}_{Q_{\text{out}}^{(0,s)}} v$ is $O(s^{1/2})$, therefore $\left( \mathbb{E}_{Q_{\text{out}}^{(0,s)}} v \right)^2 \sim O(s)$ and $\left( \mathbb{E}_{Q_{\text{out}}^{(0,s)}} v \right)^4 \sim O\left( s^2 \right)$. Expanding now eq. (6.17) to leading order in $y$ gives

$$f_1(r, s) \underset{r, s \ll 1}{=} \mathbb{E}_\eta \left[ \frac{s}{\rho_z^2} \eta^2 \left( \mathbb{E}_{Q_{\text{out}}^0} vx \right)^2 + r \left( \mathbb{E}_{\rho_0^v} v^2 \right)^2 + O\left( r^{3/2}, s^{3/2} \right) \right]$$

$$= \frac{s}{\rho_z^2} \left( \mathbb{E}_{Q_{\text{out}}^0} vx \right)^2 + r \left( \mathbb{E}_{\rho_0^v} v^2 \right)^2 + O\left( r^{3/2}, s^{3/2} \right) \tag{6.20}$$

From this expansion we read the first two entries of the Jacobian,

$$\partial_r f_1|_{(0,0)} = \left( \mathbb{E}_{Q_{\text{out}}^0} v^2 \right)^2 \qquad\qquad \partial_s f_1|_{(0,0)} = \frac{1}{\rho_z^2} \left( \mathbb{E}_{Q_{\text{out}}^0} vx \right)^2 \tag{6.21}$$

**Expansion of $f_2$:** For $f_2$, we start by expanding with respect to $s$, allowing us to take the average with respect to $\eta$ explicitly,

$$f_2(r, s) \underset{s \ll 1}{=} \alpha \mathbb{E}_\xi \left\{ \left( \mathbb{E}_{Q_{\text{out}}^{(r,0)}} x \right)^2 + \frac{s}{2\rho_z^2} \left[ 2 \left( \mathbb{E}_{Q_{\text{out}}^{(r,0)}} x \right)^4 - 4 \left( \mathbb{E}_{Q_{\text{out}}^{(r,0)}} x \right)^2 \mathbb{E}_{Q_{\text{out}}^{(r,0)}} x^2 + 2 \left( \mathbb{E}_{Q_{\text{out}}^{(r,0)}} x^2 - \rho_z \right)^2 \right] \right\}$$
(6.22)

We can now focus on the leading order expansion around $r = 0$. Note that

$$\mathbb{E}_{Q_{\text{out}}^{(r,0)}} x \underset{r \ll 1}{=} \mathbb{E}_{Q_{\text{out}}^0} x + \sqrt{r}\xi \mathbb{E}_{Q_{\text{out}}^0} xv + \frac{r}{2} (\xi^2 - 1) \mathbb{E}_{Q_{\text{out}}^0} xv^2 + O\left( r^{3/2} \right) \tag{6.23}$$

$$= \sqrt{r}\xi \mathbb{E}_{Q_{\text{out}}^0} xv + \frac{r}{2} (\xi^2 - 1) \mathbb{E}_{Q_{\text{out}}^0} xv^2 + O\left( r^{3/2} \right) \tag{6.24}$$

since

$$\mathbb{E}_{Q_{\text{out}}^0} x = \int \mathrm{d}v \int \frac{\mathrm{d}x}{\sqrt{2\pi\rho_z}} e^{-\frac{1}{2\rho_z} x^2} P_{\text{out}}(v|x) x = \int \frac{\mathrm{d}x}{\sqrt{2\pi\rho_z}} e^{-\frac{1}{2\rho_z} x^2} x = 0. \tag{6.25}$$

Therefore the leading order term is of order $O(r^{1/2})$, and $\left( \mathbb{E}_{Q_{\text{out}}^0} x \right)^2 \sim O(s)$, $\left( \mathbb{E}_{Q_{\text{out}}^0} x \right)^4 \sim O(s^2)$. Expanding now eq. (6.22) in $r \ll 1$,

$$f_2(r, s) \underset{x, s \ll 1}{=} \alpha \mathbb{E}_\xi \left[ x\xi^2 \left( \mathbb{E}_{Q_{\text{out}}^0} vx \right)^2 + \frac{s}{\rho_z^2} \left( \mathbb{E}_{Q_{\text{out}}^0} x^2 - \rho_z \right)^2 \right] + O\left( r^{3/2}, s^{3/2} \right) \tag{6.26}$$

$$= r\alpha \left( \mathbb{E}_{Q_{\text{out}}^0} vx \right)^2 + \frac{s}{\rho_z^2} \alpha \left( \mathbb{E}_{Q_{\text{out}}^0} x^2 - \rho_z \right)^2 + O\left( r^{3/2}, s^{3/2} \right) \tag{6.27}$$

From this expansion we can read the second two entries of the Jacobian,

$$\partial_r f_2|_{(0,0)} = \alpha \left( \mathbb{E}_{Q_{\text{out}}^0} vx \right)^2 \qquad\qquad \partial_s f_2|_{(0,0)} = \frac{\alpha}{\rho_z^2} \left( \mathbb{E}_{Q_{\text{out}}^0} x^2 - \rho_z \right)^2 \tag{6.28}$$

**Expansion of $f_3$:** Note that $f_3$ is independent of $(r, s)$, so it can be treated separately. Expanding in $t \ll 1$ gives

$$f_3(t) = \mathbb{E}_\xi \left[ \frac{1}{\mathcal{Z}_z^t} \left( \int \mathrm{d}x \, P_z(z) e^{-\frac{t}{2}z^2 + \sqrt{t}z\xi} z \right)^2 \right] \underset{t \ll 1}{=} \left( \mathbb{E}_{P_z} z^2 \right)^2 t + O(t^{3/2}) \tag{6.29}$$

where we have used the consistency condition in eq. (6.8). Therefore

$$\partial_t f_3|_{t=0} = \left( \mathbb{E}_{P_z} z^2 \right)^2 \tag{6.30}$$

**Bringing the overlaps back**

In our problem, we have

$$r = \frac{q_u}{\Delta} \qquad\qquad t = \hat{q}_z \qquad\qquad s = q_z \tag{6.31}$$

and therefore the partial derivatives have to be re-scaled,

$$\partial_r = \Delta \partial_{q_u} \qquad\qquad \partial_t = \partial_{\hat{q}_z} \qquad\qquad \partial_s = \partial_{q_z} \tag{6.32}$$

And therefore the Jacobian of the problem is

$$\mathbf{df}(0,0,0) = \begin{pmatrix} \frac{1}{\Delta} \left( \mathbb{E}_{Q_{\text{out}}^0} v^2 \right)^2 & 0 & \frac{1}{\rho_z^2} \left( \mathbb{E}_{Q_{\text{out}}^0} vx \right)^2 \\ \frac{\alpha}{\Delta} \left( \mathbb{E}_{Q_{\text{out}}^0} vx \right)^2 & 0 & \frac{\alpha}{\rho_z^2} \left( \mathbb{E}_{Q_{\text{out}}^0} x^2 - \rho_z \right)^2 \\ 0 & \left( \mathbb{E}_{P_z} z^2 \right)^2 & 0 \end{pmatrix} \tag{6.33}$$

## 6.3 Jacobian for the $\mathbf{uv^\mathsf{T}}$ model

The main difference in the Wishart model is that the state evolution is given in terms of four variables $(p, r, t, s) \equiv \left( \frac{q_u}{\Delta}, \beta \frac{q_v}{\Delta}, q_z, \hat{q}_z \right)$, with the update functions given by

$$\mathbf{f}(p, r, t, s) = \begin{pmatrix} f_0(r) \\ f_1(p, s) \\ f_2(p, s) \\ f_3(t) \end{pmatrix} = 2 \begin{pmatrix} \partial_r \Psi_u(r) \\ \partial_p \Psi_{\text{out}}(p, s) \\ \alpha \partial_s \Psi_{\text{out}}(p, s) \\ \partial_t \Psi_z(t) \end{pmatrix}. \tag{6.34}$$

Note that $(f_1, f_2, f_3)$ are exactly as before, with the only difference that $(f_1, f_2)$ are now evaluated at $p$ instead of $r$. The only new function is $f_0$, which depends only on $r$. This means that the new column in the Jacobian is orthogonal to all the other columns, with a single non-zero entry given by $\partial_r f_0|_{r=0}$. An easy expansion of $f_0$ to first order together with the definitions of $(p, r, t, s)$ yield

$$\mathbf{df}(0,0,0,0) = \begin{pmatrix} 0 & \frac{1}{\Delta} \left( \mathbb{E}_{P_u} u^2 \right)^2 & 0 & 0 \\ \frac{\beta}{\Delta} \left( \mathbb{E}_{Q_{\text{out}}^0} v^2 \right)^2 & 0 & 0 & \frac{1}{\rho_z^2} \left( \mathbb{E}_{Q_{\text{out}}^0} vx \right)^2 \\ \frac{\beta\alpha}{\Delta} \left( \mathbb{E}_{Q_{\text{out}}^0} vx \right)^2 & 0 & 0 & \frac{\alpha}{\rho_z^2} \left( \mathbb{E}_{Q_{\text{out}}^0} x^2 - \rho_z \right)^2 \\ 0 & 0 & \left( \mathbb{E}_{P_z} z^2 \right)^2 & 0 \end{pmatrix}. \tag{6.35}$$

## 6.4 Transition points for specific activations

The transition point $\Delta_c$ is defined as the point in which the uninformative point goes from being stable to unstable. The stability is determined in terms of the eigenvalues of the Jacobian: a fixed point is stable when the eigenvalues are smaller than one, and is unstable when the leading eigenvalue becomes greater than one.

It is instructive to look at $\Delta_c$ in specific cases. We let $P_u = P_z = \mathcal{N}(0, 1)$ together with $P_{\text{out}}(v|x) = \delta (v - \varphi(x))$ and look at different (odd) activation functions $\varphi$.

**Linear activation:** Let $\varphi(x) = x$. In this case the transition is $\Delta_c = \alpha + 1$ in the Wigner model ($\mathbf{vv^\mathsf{T}}$) and $\Delta_c = \sqrt{\beta(\alpha + 1)}$ in the Wishart model ($\mathbf{uv^\mathsf{T}}$)

**Sign activation:** Let $\varphi(x) = \mathrm{sgn}(x)$. In this case the transition is $\Delta_c = 1 + \frac{4}{\pi^2}\alpha$ in the Wigner model ($\mathbf{vv^\mathsf{T}}$) and $\Delta_c = \sqrt{\beta \left( 1 + \frac{4}{\pi^2}\alpha \right)}$ in the Wishart model ($\mathbf{uv^\mathsf{T}}$).

# 7 Random matrix analysis of the transition

In this section, we describe how we can derive the value $\Delta_c$ at which a transition appears in the recovery for a linear activation function, for both the symmetric $\mathbf{v}\mathbf{v}^\mathsf{T}$ and non-symmetric $\mathbf{u}\mathbf{v}^\mathsf{T}$ case, purely from a random matrix theory analysis. This transition is in essence similar to the celebrated Baik-Ben Arous-Péché (BBP) transition of the largest eigenvalue of a spiked Wishart (or Wigner) matrix [18].

## 7.1 A reminder on the Stieltjes transform

Let $\mathbb{C}_+ = \{z \in \mathbb{C}, \text{ Im } z > 0\}$. For any probability measure $\nu$ on $\mathbb{R}$, and any $z \in \mathbb{C}\backslash\text{supp }\nu$, we can define the Stieltjes transform of $\nu$ as:

$$g_\nu(z) \equiv \mathbb{E}_\nu \frac{1}{X - z}.$$

Note that $g_\nu(z)$ is a one-to-one mapping of $\mathbb{C}_+$ on itself. The Stieltjes transform has proven to be a very useful tool from random matrix theory. One of its important features, that we will use to compute the bulk density (see Fig. (3) of the main material) is the Stieltjes-Perron inversion formula, that we state here (see Theorem X.6.1 of [19]):

**Theorem 7.1** (Stieltjes-Perron). *Assume that $\nu$ has a continuous density on $\mathbb{R}$ with respect to the Lebesgue measure. Then:*

$$\forall x \in \mathbb{R}, \quad \frac{\mathrm{d}\nu}{\mathrm{d}x} = \lim_{\epsilon \to 0^+} \frac{1}{\pi}\text{Im } g_\nu(x + i\epsilon).$$

Informally, one has to think that the knowledge of the Stieltjes transform above the real line uniquely determines the measure $\nu$. The Stieltjes transform is particulaly useful in random matrix theory. Consider a (random) symmetric matrix $M$ of size $n$, with real eigenvalues $\{\lambda_i\}$. Then the empirical spectral measure of $M$ is defined as:

$$\nu_n \equiv \frac{1}{n}\sum_{i=1}^n \delta_{\lambda_i}. \tag{7.1}$$

For some random matrix ensembles, the (random) probability measure $\nu_n$ will converge almost surely and in the weak sense to a deterministic probability measure $\nu$ as $n \to \infty$. In this case, we will call $\nu$ the *asymptotic spectral measure* of $M$.

## 7.2 The symmetric $\mathbf{v}\mathbf{v}^\mathsf{T}$ linear case

In this setting, the stationary AMP equations can be reduced on the vector $\hat{\mathbf{v}}$ as:

$$\hat{\mathbf{v}} = \left[\frac{1}{k}WW^\mathsf{T}\right]\left[\frac{1}{\sqrt{\Delta p}}\xi + \frac{1}{\Delta}\frac{\mathbf{v}\mathbf{v}^\mathsf{T}}{p} - \frac{1}{\Delta}\mathrm{I}_p\right]\hat{\mathbf{v}}. \tag{7.2}$$

We assume in the following that $\rho_v = 1$ to simplify the analysis (in this linear problem, it does not imply any loss of generality). Here $\xi/\sqrt{p}$ is a matrix from the Gaussian Orthogonal Ensemble, i.e. $\xi$ is a real symmetric matrix with entries drawn independently from a Gaussian distribution with zero mean and variance $\mathbb{E}\,\xi_{ij}^2 = (1 + \delta_{ij})$. We denote:

$$\Gamma_p^{vv} \equiv \left[\frac{1}{k}WW^\mathsf{T}\right]\left[\frac{1}{\sqrt{\Delta p}}\xi + \frac{1}{\Delta}\frac{\mathbf{v}\mathbf{v}^\mathsf{T}}{p} - \frac{1}{\Delta}\mathrm{I}_p\right]. \tag{7.3}$$

From the state evolution analysis we expect that the eigenvector of $\Gamma_p^{vv}$ associated to its largest eigenvalue has a non-zero overlap with $\mathbf{v}$ in the large $p$ limit as soon as $\Delta < \Delta_c(\alpha) \equiv 1 + \alpha$. In this section, we show this fact using only random matrix theory.

Informally, we first demonstrate that the supremum of the support of the asymptotic spectral measure of $\Gamma_p^{vv}$ touches 1 exactly for $\Delta = \Delta_c(\alpha)$. Then, for $\Delta \leq \Delta_c(\alpha)$, the largest eigenvalue of $\Gamma_p^{vv}$ will converge to 1, which is separated from the bulk of the asymptotic spectral density. The corresponding eigenvector is also positively correlated with $\mathbf{v}$. This gives more detail to the mechanisms of the transition. We show first the following characterization of the asymptotic spectral density of $\Gamma_p^{vv}$:

**Theorem 7.2.** *For any $\alpha, \Delta > 0$, as $p \to +\infty$, the spectral measure of $\Gamma_p^{vv}$ converges almost surely and in the weak sense to a well-defined and compactly supported probability measure $\mu(\alpha, \Delta)$, and we denote $\operatorname{supp} \mu$ its support. We separate two cases:*

(*i*) *If $\Delta \leq \frac{1}{4}$, then $\operatorname{supp} \mu \subseteq \mathbb{R}_-$.*

(*ii*) *Assume now $\Delta > \frac{1}{4}$ and denote $z_1(\Delta) \equiv -\Delta^{-1} + 2\Delta^{-1/2} > 0$. Let $\rho_\Delta$ be the probability measure on $\mathbb{R}$ with density*

$$
\rho_\Delta(\mathrm{d}t) = \frac{\sqrt{\Delta}}{2\pi} \sqrt{4 - \Delta \left(t + \frac{1}{\Delta}\right)^2} \, \mathbb{1}\left\{\left|t + \frac{1}{\Delta}\right| \leq \frac{2}{\sqrt{\Delta}}\right\} \, \mathrm{d}t. \tag{7.4}
$$

*Note that the supremum of the support of $\rho_\Delta$ is $z_1(\Delta)$. The following equation admits a unique solution for $s \in (-z_1(\Delta)^{-1}, 0)$:*

$$
\alpha \int \rho_\Delta(\mathrm{d}t) \left(\frac{st}{1 + st}\right)^2 = 1. \tag{7.5}
$$

*We denote this solution as $s_{\mathrm{edge}}(\alpha, \Delta)$ (or simply $s_{\mathrm{edge}}$). The supremum of the support of $\mu(\alpha, \Delta)$ is denoted $\lambda_{\max}(\alpha, \Delta)$ (or simply $\lambda_{\max}$). It is given by:*

$$
\lambda_{\max} = \begin{cases} -\dfrac{1}{s_{\mathrm{edge}}} + \alpha \displaystyle\int \rho_\Delta(\mathrm{d}t) \dfrac{t}{1 + s_{\mathrm{edge}}t} & \text{if } \alpha \leq 1, \\[3ex] \max\left(0, -\dfrac{1}{s_{\mathrm{edge}}} + \alpha \displaystyle\int \rho_\Delta(\mathrm{d}t) \dfrac{t}{1 + s_{\mathrm{edge}}t}\right) & \text{if } \alpha > 1. \end{cases} \tag{7.6}
$$

Before proving Theorem 7.2, we state a very interesting corollary:

**Corollary 7.1.** *Let $\alpha > 0$. As a function of $\Delta$, $\lambda_{\max}$ (see Theorem 7.2) has a unique global maximum, reached exactly at the point $\Delta = \Delta_c(\alpha) = 1 + \alpha$. Moreover, $\lambda_{\max}(\alpha, \Delta_c(\alpha)) = 1$.*

We can then state the transition result. Its method of proof is very much inspired by [20] [3].

**Theorem 7.3.** *Let $\alpha, \Delta > 0$. Let us denote $\lambda_1 \geq \lambda_2$ the first and second eigenvalues of $\Gamma_p^{vv}$. Then we have:*

- *If $\Delta \geq \Delta_c(\alpha)$, then as $p \to \infty$ we have $\lambda_1 \underset{a.s.}{\to} \lambda_{\max}$ and $\lambda_2 \underset{a.s.}{\to} \lambda_{\max}$.*

- *If $\Delta \leq \Delta_c(\alpha)$, then as $p \to \infty$ we have $\lambda_1 \underset{a.s.}{\to} 1$ and $\lambda_2 \underset{a.s.}{\to} \lambda_{\max}$.*

*Moreover, let us denote $\tilde{\boldsymbol{v}}$ an eigenvector of $\Gamma_p^{vv}$ with eigenvalue $\lambda_1$, normalized such that $\|\tilde{\boldsymbol{v}}\|^2 = p$. Then:*

$$
\frac{1}{p^2} |\tilde{\boldsymbol{v}}^\mathsf{T} \boldsymbol{v}|^2 \underset{a.s.}{\to} \epsilon(\Delta). \tag{7.7}
$$

*The function $\epsilon(\Delta)$ satisfies the following properties: $\epsilon(\Delta) = 0$ for all $\Delta \geq \Delta_c(\alpha)$, $\epsilon(\Delta) > 0$ for all $\Delta < \Delta_c(\alpha)$ and $\lim_{\Delta \to 0} \epsilon(\Delta) = 1$.*

Our method of proof for Theorem 7.3 allows us to compute numerically the squared correlation $\epsilon(\Delta)$. It is given, for all $\Delta < \Delta_c(\alpha)$, as

$$
\epsilon(\Delta) = \frac{1}{\alpha} \frac{\left[S^{(2)}(1)\right]^2}{S^{(1,2)}(1)}.
$$

The $S^{(1,2)}$ and $S^{(2)}$ functions are defined in Lemma 7.2, and formulas are also given that allow to compute them numerically. A non-trivial consistency check is to verify that $\epsilon(\Delta)$ coincides with the variable $q_v$ given by the mutual information analysis of Theorem 1 of the main material. We show numerically that they indeed coincide in Fig. 2.

Figure 2: The function $\epsilon(\Delta)$ computed in the linear case by Theorem 1 of the main material (information theoretic analysis) and Theorem 7.3 (random matrix analysis) ($\alpha = 2$).

***Remark*** (The nature of the transition). As was already noticed in some previous works (see for instance a related remark in [20]), the existence of a transition in the largest eigenvalue and the corresponding eigenvector for a large matrix of the type $M + \theta P$ (with $P$ of finite rank and $\theta > 0$) depends on the decay of the asymptotic spectral density of $M$ at the right edge of its bulk. For a power-law decay, there can be either no transition, a transition in the largest eigenvalue and the corresponding eigenvector, or a transition in the largest eigenvalue but not in the corresponding eigenvector. The situation in our setting is somewhat more involved, as both the bulk and the spike depend on the parameter $\Delta$, and they are not independent (they are correlated via the matrix $W$). However, this intuition remains true: if we do not show and use it explicitly, the decay of the density of $\mu(\alpha, \Delta)$ at the right edge is of the type $(\lambda_{\max} - \lambda)^{1/2}$, which is the hidden feature that is responsible for a transition both in the largest eigenvalue and the corresponding eigenvector, which is what we show in Theorem 7.2.

### 7.3 The non-symmetric uvᵀ linear case

The analysis is very similar to the one of the symmetric case of Section 7.2. The counterpart to the matrix of eq. (7.3) is here:

$$\Gamma_p^{uv} \equiv \frac{1}{\Delta} \frac{WW^\mathsf{T}}{k} \times \left( \frac{1}{1+\Delta} \frac{y^\mathsf{T} y}{p} - \beta \, \mathrm{I}_p \right) \in \mathbb{R}^{p \times p}. \tag{7.8}$$

Recall that we have here $\alpha = \frac{p}{k}$ and $\beta = \frac{n}{p}$. $W \in \mathbb{R}^{p \times k}$ is an i.i.d. standard Gaussian matrix, and the matrix $y \in \mathbb{R}^{n \times p}$ is constructed as:

$$y = \sqrt{\Delta} \xi + \frac{\mathbf{u}\mathbf{v}^\mathsf{T}}{\sqrt{p}}. \tag{7.9}$$

Here, $\xi \in \mathbb{R}^{n \times p}$ is also an i.i.d. standard Gaussian matrix, independent of $W$. As it will be useful for stating the theorem, we recall the Marchenko-Pastur probability measure with ratio $\beta$, denoted $\rho_{\mathrm{MP}, \beta}$ [22]:

$$\begin{cases} \lambda_+(\beta) = \left( 1 + \frac{1}{\sqrt{\beta}} \right)^2, & \text{(7.10a)} \\[2ex] \lambda_-(\beta) = \left( 1 - \frac{1}{\sqrt{\beta}} \right)^2, & \text{(7.10b)} \\[2ex] \frac{\mathrm{d}\rho_{\mathrm{MP}, \beta}}{\mathrm{d}t} \equiv (1 - \beta)\, \delta(t) + \frac{\beta}{2\pi} \frac{\sqrt{[\lambda_+(\beta) - t]\,[t - \lambda_-(\beta)]}}{t} \mathbb{1}_{t \in (\lambda_-(\beta), \lambda_+(\beta))}. & \text{(7.10c)} \end{cases}$$

We can now state the couterpart to Theorem 7.2 in the **uvᵀ** setting:

**Theorem 7.4.** *For any $\alpha, \beta, \Delta > 0$, the spectral measure of $\Gamma_p^{uv}$ converges almost surely and in the weak sense to a well-defined and compactly supported measure $\mu(\Delta, \alpha, \beta)$. We denote $\mathrm{supp}\,\mu$ its*

*support. We introduce a function $z_1$ and a probability measure $\rho_{\beta,\Delta}$ as follows:*

$$z_1(\beta, \Delta) \equiv \frac{-\beta + \Delta + 2\Delta\sqrt{\beta}}{\Delta(1 + \Delta)},$$

$$\frac{\mathrm{d}\rho_{\beta,\Delta}}{\mathrm{d}t} \equiv \frac{1+\Delta}{\beta} \frac{\mathrm{d}\rho_{\mathrm{MP},\beta}}{\mathrm{d}t} \left( \frac{1+\Delta}{\beta} t + \frac{1+\Delta}{\Delta} \right). \tag{7.11}$$

*Note that $z_1(\beta, \Delta)$ is the supremum of the support of $\rho_{\beta,\Delta}$. Let finally*

$$\Delta_{\mathrm{pos}}(\beta) \equiv \frac{\beta}{1 + 2\sqrt{\beta}}. \tag{7.12}$$

*We separate two cases:*

(i) *If $\Delta \leq \Delta_{\mathrm{pos}}(\beta)$, then $z_1(\beta, \Delta) \leq 0$ and $\operatorname{supp} \mu \subseteq \mathbb{R}_-$.*

(ii) *Assume now $\Delta > \Delta_{\mathrm{pos}}(\beta)$. Then $z_1(\beta, \Delta) > 0$. The following equation admits a unique solution for $s \in (-z_1(\beta, \Delta)^{-1}, 0)$:*

$$\alpha \int \rho_{\beta,\Delta}(\mathrm{d}t) \left( \frac{st}{1 + st} \right)^2 = 1. \tag{7.13}$$

*We denote this solution as $s_{\mathrm{edge}}(\alpha, \beta, \Delta)$ (or simply $s_{\mathrm{edge}}$). We denote $\lambda_{\max}(\alpha, \beta, \Delta)$ (or only $\lambda_{\max}$) the supremum of the support of $\mu(\Delta, \alpha, \beta)$. Then we have:*

$$\lambda_{\max} = \begin{cases} -\dfrac{1}{s_{\mathrm{edge}}} + \alpha \displaystyle\int \rho_{\beta,\Delta}(\mathrm{d}t) \dfrac{t}{1 + s_{\mathrm{edge}}t} & \text{if } \alpha \leq 1, \\[2ex] \max\left( 0, -\dfrac{1}{s_{\mathrm{edge}}} + \alpha \displaystyle\int \rho_{\beta,\Delta}(\mathrm{d}t) \dfrac{t}{1 + s_{\mathrm{edge}}t} \right) & \text{if } \alpha > 1. \end{cases} \tag{7.14}$$

We can state the corresponding corollary to this theorem:

**Corollary 7.2.** *Let $\alpha, \beta > 0$. Seen as a function of $\Delta$, $\lambda_{\max}$ (see Theorem 7.2) has a unique global maximum, attained exactly at the point $\Delta_c(\alpha, \beta) \equiv \sqrt{\beta(1 + \alpha)}$. Moreover,*

$$\lambda_{\max}(\alpha, \beta, \Delta_c(\alpha, \beta)) = 1.$$

We can then describe the complete transition. Proving this transition would follow the same main lines as the proof of the transition in the $\mathbf{v}\mathbf{v}^\intercal$ case (Theorem 7.3), but would be significantly heavier. This is left for future work, so we state the transition in this setting as a conjecture:

**Conjecture 7.1.** *Let $\alpha, \beta, \Delta > 0$. Let us denote $\lambda_1 \geq \lambda_2$ the first and second eigenvalues of $\Gamma_p^{uv}$. Then we have:*

- *If $\Delta \geq \Delta_c(\alpha, \beta)$, then as $p \to \infty$ we have $\lambda_1 \underset{a.s.}{\to} \lambda_{\max}$ and $\lambda_2 \underset{a.s.}{\to} \lambda_{\max}$.*

- *If $\Delta \leq \Delta_c(\alpha, \beta)$, then as $p \to \infty$ we have $\lambda_1 \underset{a.s.}{\to} 1$ and $\lambda_2 \underset{a.s.}{\to} \lambda_{\max}$.*

*Let us denote $\tilde{\mathbf{v}}$ an eigenvector of $\Gamma_p^{uv}$ with eigenvalue $\lambda_1$, normalized such that $\|\tilde{\mathbf{v}}\|^2 = p$. Then:*

$$\frac{1}{p^2} |\tilde{\mathbf{v}}^\intercal \mathbf{v}|^2 \underset{a.s.}{\to} \epsilon(\Delta). \tag{7.15}$$

*It satisfies $\epsilon(\Delta) = 0$ for all $\Delta \geq \Delta_c(\alpha, \beta)$, $\epsilon(\Delta) > 0$ for all $\Delta < \Delta_c(\alpha, \beta)$ and $\lim_{\Delta \to 0} \epsilon(\Delta) = 1$.*

## 7.4 Proof of Theorem 7.2 and Corollary 7.1

### 7.4.1 Proof of Theorem 7.2

*Proof of Theorem 7.2 (ii).* We begin by treating the more involved case $(ii)$, that is we assume $\Delta > \frac{1}{4}$. Note first that by basic linear algebra, the spectrum of $\Gamma_p^{vv}$ is, up to 0 eigenvalues, the same as the spectrum of the following matrix $\Gamma_k^{vv}$:

$$\Gamma_k^{vv} \equiv \frac{1}{k} W^\intercal \left[ \frac{1}{\sqrt{\Delta p}} \xi + \frac{1}{\Delta} \frac{\mathbf{v}\mathbf{v}^\intercal}{p} - \frac{1}{\Delta} \mathrm{I}_p \right] W \in \mathbb{R}^{k \times k}, \tag{7.16}$$

More precisely, if $p \geq k$ (so $\alpha \geq 1$) we have $\mathrm{Sp}\,(\Gamma_p^{vv}) = \mathrm{Sp}\,(\Gamma_k^{vv}) \cup \{0\}^{p-k}$, and conversely if $k > p$. These additional zero eigenvalues in the case $\alpha > 1$ explain the $\max(0, \cdot)$ term in the conclusion of Theorem 7.2.

For the remainder of the proof we can thus consider $\Gamma_k^{vv}$ instead of $\Gamma_p^{vv}$ given the remark above. Moreover, for simplicity we will drop the $vv$ exponent in those matrices, and just denote them $\Gamma_k, \Gamma_p$. The bulk of $\Gamma_k$ can be studied using standard random matrix theory results. Such matrices were first studied by Marchenko and Pastur in a seminal work [22], which was generalized (and made rigorous) later in [21]. Note finally that by the celebrated results of Wigner [23], the spectral distribution of the matrix $\xi/\sqrt{\Delta p} - \mathrm{I}_p/\Delta$ converges in law (and almost surely) as $p \to \infty$ to $\rho_\Delta$, given by eq. (7.4). We can then use Theorem 1.1 of [21], that we recall here for our setting:

**Theorem 7.5** (Silverstein-Bai). *Let $p, k \to \infty$ with $p/k \to \alpha > 0$. Let $W \in \mathbb{R}^{p \times k}$ be an i.i.d. Gaussian matrix, whose elements come from the standard Gaussian distribution $\mathcal{N}(0, 1)$. Let $T_p \in \mathbb{R}^{p \times p}$ be a random symmetric matrix, independent of $W$, such that the empirical spectral distribution of $T_p$ converges (almost surely) in law to a measure $\rho_T$. Then, almost surely, the empirical spectral distribution of $B_k \equiv \frac{1}{k} W^\intercal T_p W$ converges in law to a (nonrandom) measure $\mu_B$, whose Stieltjes transform satisfies, for every $z \in \mathbb{C}_+$:*

$$g_{\mu_B}(z) = -\left[ z - \alpha \int \nu_T(\mathrm{d}t) \frac{t}{1 + t g_{\mu_B}(z)} \right]^{-1}. \tag{7.17}$$

*Moreover, for every $z \in \mathbb{C}_+$, there is a unique solution to eq. (7.17) such that $g_{\mu_B}(z) \in \mathbb{C}_+$. This equation thus characterizes unambiguously the measure $\mu_B$.*

Applying Theorem 7.5 to our setting shows that we can define $\nu(\alpha, \Delta)$ as the limit eigenvalue distribution of $\Gamma_k$, and we denote $g_\nu(z)$ its Stieltjes transform. From the remarks above, $\mu(\alpha, \Delta)$ and $\nu(\alpha, \Delta)$ only differ by the addition of a delta distribution. For instance, if $\alpha \geq 1$:

$$\mu(\alpha, \Delta) = \alpha \nu(\alpha, \Delta) + (1 - \alpha)\delta_0. \tag{7.18}$$

The main quantity of interest to us is $z_{\mathrm{edge}}$, defined as the supremum of the support of $\nu(\alpha, \Delta)$. If $z_{\mathrm{edge}} \geq 0$, then it will also be the supremum of the support of $\mu(\alpha, \Delta)$, and thus equal to $\lambda_{\max}$. Theorem 7.5 shows that for every $z \in \mathbb{C}_+ \cup (\mathbb{R}\backslash\mathrm{supp}\,\nu)$, $g_\nu(z)$ is the only solution in $\mathbb{C}_+ \cup \mathbb{R}$ to the following equation:

$$g_\nu(z) = -\left[ z - \alpha \int \rho_\Delta(\mathrm{d}t) \frac{t}{1 + t g_\nu(z)} \right]^{-1}. \tag{7.19}$$

The validity of the equation for $\mathbb{R}\backslash\mathrm{supp}\,\nu$ (and not only on $\mathbb{C}_+$) follows from the continuity of $g_\nu(z)$ on $\mathbb{C}_+ \cup (\mathbb{R}\backslash\mathrm{supp}\,\nu)$, a generic property of the Stieltjes transform. It is easy to see that $g_\nu$ induces a strictly increasing diffeomorphism $g_\nu : (z_{\mathrm{edge}}, +\infty) \to (\lim_{z \to z_{\mathrm{edge}}^+} g_\nu(z), 0)$, so that we can define its inverse $g_\nu^{-1}$ and from eq. (7.19), it satisfies for every $s \in (\lim_{z \to z_{\mathrm{edge}}^+} g_\mu(z), 0)$:

$$g_\nu^{-1}(s) = -\frac{1}{s} + \alpha \int \rho_\Delta(\mathrm{d}t) \frac{t}{1 + st}. \tag{7.20}$$

**Remark**  Note that this can be written in terms of the $\mathcal{R}$-transform of $\nu$ (an useful tool of free probability):

$$\mathcal{R}_\nu(s) \equiv g_\nu^{-1}(-s) - \frac{1}{s} = \alpha \int \rho_\Delta(\mathrm{d}t) \frac{t}{1 - st}.$$

In order to compute $z_{\mathrm{edge}}$ from eq. (7.19), we use a result of Section 4 of [21], also stated for instance in [24], that describes the form of the support of $\nu(\alpha, \Delta)$. It can be stated in the following way. Recall that since $\Delta > \frac{1}{4}$, $z_1(\Delta) > 0$ is the maximum of the support of $\rho_\Delta$. Let $s_{\mathrm{edge}}$ be the unique solution in $(-z_1(\Delta)^{-1}, 0)$ of the equation $(g_\nu^{-1})'(s) = 0$, that is by eq. (7.20):

$$\alpha \int \rho_\Delta(\mathrm{d}t) \left( \frac{st}{1 + st} \right)^2 = 1. \tag{7.21}$$

Indeed, it is straighforward to show that the left-hand side of eq. (7.21) tends to $0$ as $s \to 0^-$, tends to $+\infty$ as $s \to -z_1(\Delta)^{-1}$, and is a strictly decreasing and continuous function of $s$. Then (see for instance eq. (2.13) and eq. (2.14) of [24]) $z_{\mathrm{edge}}$ is given by

$$z_{\mathrm{edge}} = \lim_{s \to s_{\mathrm{edge}}^+} g_\nu^{-1}(s),$$

$$= -\frac{1}{s_{\mathrm{edge}}} + \alpha \int \rho_\Delta(\mathrm{d}t) \frac{t}{1 + s_{\mathrm{edge}}t}. \tag{7.22}$$

This ends the proof of $(ii)$. ∎

Let us make a final remark that will be useful in our future analysis. Note that $z_1(\Delta) > 1$ for all $\Delta > 1$. Moreover, for all $\Delta > 1$, we have by an explicit computation:

$$\alpha \int \rho_\Delta(\mathrm{d}t) \left( \frac{t}{1 - t} \right)^2 = \frac{\alpha}{\Delta - 1}.$$

By the argument above, this yields the following result, that we state as a lemma:

**Lemma 7.1.** *Assume $\Delta > 1$. Then:*

(i) *If $\Delta < \Delta_c(\alpha)$, then $s_{\mathrm{edge}} > -1$.*

(ii) *If $\Delta = \Delta_c(\alpha)$, then $s_{\mathrm{edge}} = -1$.*

(iii) *If $\Delta > \Delta_c(\alpha)$, then $s_{\mathrm{edge}} < -1$.*

*Proof of Theorem 7.2, (i).* Assume now $\Delta \leq \frac{1}{4}$. Then the support of $\rho_\Delta$ is a subset of $\mathbb{R}_-$. Since $0 \in \mathbb{R}_-$, we can use again the remark we made in the proof of $(ii)$ to study $\Gamma_k$ instead of $\Gamma_p$. Moreover, Theorem 7.5 still applies here so that we have the Silverstein equation (7.20) for every $s \in \mathbb{C}_+$:

$$g_\nu^{-1}(s) = -\frac{1}{s} + \alpha \int \rho_\Delta(\mathrm{d}t) \frac{t}{1 + st}.$$

By the Stieltjes-Perron inversion Theorem 7.1, it is enough to check that for every $z > 0$, there exists a unique $s < 0$ such that $g_\nu^{-1}(s) = z$. Indeed, this will yield $s = g_\nu(z) \in \mathbb{R}$. In particular, $\lim_{\epsilon \to 0^+} \mathrm{Im}\, g_\nu(z + i\epsilon) = 0$ for every $z > 0$, which will imply $\mathrm{supp}(\nu) \subseteq \mathbb{R}_-$ and thus $\mathrm{supp}(\mu) \subseteq \mathbb{R}_-$.

Therefore, let $z > 0$. From eq. (7.20) and the fact that $\mathrm{supp}(\rho_\Delta) \subseteq \mathbb{R}_-$, we easily obtain:

$$\lim_{s \to -\infty} g_\nu^{-1}(s) = 0,$$

$$\lim_{s \to 0^-} g_\nu^{-1}(s) = +\infty.$$

Moreover, $g_\nu^{-1}(s)$ is a strictly increasing continuous function of $s$, so that the existence and unicity of $s = g_\nu(z) < 0$ is immediate, which ends the proof. ∎

### 7.4.2 Proof of Corollary 7.1

*Proof.* Let us make a few remarks:

- By Theorem 7.2, we know that if $\Delta \leq \frac{1}{4}$, then $\lambda_{\max} \leq 0$.

- It is trivial by the form of $\Gamma_p$ that, as $\Delta \to +\infty$, $\lambda_{\max} \to 0$.

Let $z_{\mathrm{edge}} = -\frac{1}{s_{\mathrm{edge}}} + \alpha \int \rho_\Delta(\mathrm{d}t) \frac{t}{1 + s_{\mathrm{edge}}t}$. Then we know that $\lambda_{\max} = z_{\mathrm{edge}}$ if $\alpha \leq 1$ and $\lambda_{\max} = \max(0, z_{\mathrm{edge}})$ if $\alpha > 1$. In particular, by the remark above, $z_{\mathrm{edge}} \leq 0$ for $\Delta = \frac{1}{4}$ and

$z_{\text{edge}} \to 0^+$ as $\Delta \to \infty$. It is easy to see that $z_{\text{edge}}$ is a continuous and differentiable function of $\Delta$, so that if we show the two following facts for any $\Delta \geq \frac{1}{4}$:

$$\frac{\mathrm{d}z_{\text{edge}}}{\mathrm{d}\Delta} = 0 \Leftrightarrow \Delta = \Delta_c(\alpha) = 1 + \alpha, \tag{7.23}$$

$$z_{\text{edge}}(\Delta_c(\alpha)) = 1, \tag{7.24}$$

this would end the proof as $z_{\text{edge}}$ would necessarily have a unique global maximum, located in $\Delta = \Delta_c(\alpha)$, in which we have $\lambda_{\max} = 1$. We thus prove eq. (7.23) and eq. (7.24) in the following.

**Proof of eq. (7.23)**     By the chain rule:

$$\frac{\mathrm{d}z_{\text{edge}}}{\mathrm{d}\Delta} = \frac{\partial z_{\text{edge}}}{\partial \Delta} + \frac{\partial s_{\text{edge}}}{\partial \Delta} \frac{\partial z_{\text{edge}}}{\partial s_{\text{edge}}},$$

$$= \frac{\partial z_{\text{edge}}}{\partial \Delta},$$

using the very definition of $s_{\text{edge}}$, eq. (7.21), as $z_{\text{edge}} = g_\nu^{-1}(s_{\text{edge}})$. Given the explicit form of $\rho_\Delta$, one can compute easily:

$$\frac{\partial z_{\text{edge}}}{\partial \Delta} = -\alpha \frac{s_{\text{edge}} + 2s_{\text{edge}}^2 - \Delta + \sqrt{s_{\text{edge}}^2 - 2s_{\text{edge}}(1 + 2s_{\text{edge}})\Delta + \Delta^2}}{2s_{\text{edge}}^3 \sqrt{s_{\text{edge}}^2 - 2s_{\text{edge}}(1 + 2s_{\text{edge}})\Delta + \Delta^2}}.$$

It is then simple analysis to see that since $s_{\text{edge}} < 0$, $\frac{\partial z_{\text{edge}}}{\partial \Delta} = 0$ is equivalent to $s_{\text{edge}} = -1$ and $\Delta > 1$. Recall that $s_{\text{edge}}$ is originally defined as a solution to eq. (7.21):

$$\alpha \int \rho_\Delta(\mathrm{d}t) \left( \frac{s_{\text{edge}}t}{1 + s_{\text{edge}}t} \right)^2 = 1.$$

Inserting $s_{\text{edge}} = -1$ into this equation and using the explicit form of $\rho_\Delta$ given by eq. (7.4), and using moreover that $\Delta > 1$, this reduces to:

$$\frac{\alpha}{\Delta - 1} = 1,$$

which is equivalent to $\Delta = \Delta_c(\alpha) = 1 + \alpha$.

**Proof of eq. (7.24)**     By Lemma 7.1, we know that for $\Delta = \Delta_c(\alpha)$ we have $s_{\text{edge}} = -1$. Given eq. (7.4), it is then straightforward to compute:

$$z_{\text{edge}}(\Delta_c(\alpha)) = -1 + \alpha \int \rho_{\Delta_c(\alpha)}(\mathrm{d}t) \frac{t}{1 - t},$$

$$= 1.$$

∎

## 7.5   Proof of Theorem 7.3

### 7.5.1   Transition of the largest eigenvalue

This part is a detailed outline of the proof. Some parts of the calculation are not fully rigorous, however they can be justified more precisely by following exactly the lines of [20] and [25]. We will emphasize when such refinements have to be made. Recall that we have by eq. (7.3) the following decomposition of $\Gamma_p^{vv}$ (that we denote $\Gamma_p$ for simplicity):

$$\Gamma_p = \underbrace{\left[ \frac{1}{k} WW^{\mathsf{T}} \right] \left[ \frac{1}{\sqrt{\Delta p}} \xi - \frac{1}{\Delta} \mathrm{I}_p \right]}_{\Gamma_p^{(0)}} + \underbrace{\frac{1}{\Delta} \frac{WW^{\mathsf{T}}}{k} \frac{\mathbf{v}\mathbf{v}^{\mathsf{T}}}{p}}_{\text{rank 1 perturbation}}. \tag{7.25}$$

Theorem 7.2 and Corollary. 7.1, along with their respective proofs, already describe in great detail the limit eigenvalue distribution of $\Gamma_p^{(0)}$. We first note that for any $\lambda \in \mathbb{R}$ that is not an eigenvalue of $\Gamma_p^{(0)}$ one can write:

$$\det(\lambda \mathrm{I}_p - \Gamma_p) = \det\left(\lambda \mathrm{I}_p - \Gamma_p^{(0)}\right) \det\left( \mathrm{I}_p - \left(\lambda \mathrm{I}_p - \Gamma_p^{(0)}\right)^{-1} \frac{1}{\Delta} \frac{WW^{\mathsf{T}}}{k} \frac{\mathbf{v}\mathbf{v}^{\mathsf{T}}}{p} \right).$$

In particular, this implies immediately that $\lambda$ is an eigenvalue of $\Gamma_p$ and not an eigenvalue of $\Gamma_p^{(0)}$ if and only if $1$ is an eigenvalue of $\left(\lambda \mathrm{I}_p - \Gamma_p^{(0)}\right)^{-1} \frac{1}{\Delta} \frac{WW^\intercal}{k} \frac{\mathbf{v}\mathbf{v}^\intercal}{p}$. Since this is a rank-one matrix, its only non-zero eigenvalue is equal to its trace, so it is equivalent to:

$$1 = \mathrm{Tr}\left[\left(\lambda \mathrm{I}_p - \Gamma_p^{(0)}\right)^{-1} \frac{1}{\Delta} \frac{WW^\intercal}{k} \frac{\mathbf{v}\mathbf{v}^\intercal}{p}\right]. \tag{7.26}$$

Recall that by definition, $\mathbf{v}$ is constructed as $\mathbf{v} = W\mathbf{z}/\sqrt{k}$, with $\mathbf{z}$ a standard Gaussian i.i.d. vector in $\mathbb{R}^k$, independent of $W$. For any matrix $A$, we have the classical concentration $\frac{1}{k}\mathbf{z}^\intercal A\mathbf{z} = \frac{1}{k}\mathrm{Tr}A$ with high probability as $k \to \infty$. In eq. (7.26), this yields at leading order as $p \to \infty$:

$$\Delta = \frac{1}{p}\mathrm{Tr}\left[\left(\lambda \mathrm{I}_p - \Gamma_p^{(0)}\right)^{-1} \left(\frac{WW^\intercal}{k}\right)^2\right]. \tag{7.27}$$

We will prefer to use $k \times k$ matrices. We use the simple linear algebra identity, for any $p \times p$ symmetric matrix $A$, and any integer $q \geq 1$:

$$\mathrm{Tr}\left[\left(\lambda \mathrm{I}_p - \frac{WW^\intercal}{k}A\right)^{-1}\left(\frac{WW^\intercal}{k}\right)^q\right] = \mathrm{Tr}\left[\left(\lambda \mathrm{I}_k - \frac{1}{k}W^\intercal AW\right)^{-1}\left(\frac{W^\intercal W}{k}\right)^q\right].$$

This can be derived for instance by expanding both sides in powers of $\lambda^{-1}$ and using the cyclicity of the trace. Finally, we can state that the eigenvalues of $\Gamma_p$ that are outside of the spectrum of $\Gamma_p^{(0)}$ must satisfy, as $k \to \infty$:

$$\alpha\Delta = \frac{1}{k}\mathrm{Tr}\left[\left(\lambda \mathrm{I}_k - \Gamma_k^{(0)}\right)^{-1}\left(\frac{W^\intercal W}{k}\right)^2\right], \tag{7.28}$$

with

$$\Gamma_k^{(0)} \equiv \frac{1}{k}W^\intercal\left[\frac{1}{\sqrt{\Delta p}}\xi - \frac{1}{\Delta}\mathrm{I}_p\right]W.$$

We will now make use of two important lemmas, at the core of our analysis. They will also prove to be useful in the eigenvector correlation analysis.

**Lemma 7.2.** *Recall that $\nu$ is the limit eigenvalue distribution of $\Gamma_k^{(0)}$, that the supremum of its support is $\lambda_{\max}$, and its Stieltjes transform is $g_\nu$. For every integer $r \geq 0$, we define:*

$$S_k^{(r)}(\lambda) \equiv \frac{1}{k}\mathrm{Tr}\left[\left(\Gamma_k^{(0)} - \lambda \mathrm{I}_k\right)^{-1}\left(\frac{W^\intercal W}{k}\right)^r\right].$$

*For $r \in \{0,1,2,3\}$[4] and every $\lambda > \lambda_{\max}$, as $k \to \infty$ $S_k^{(r)}(\lambda)$ converges almost surely to a well defined limit $S^{(r)}(\lambda)$. This limit is given by:*

$$\begin{cases}
S^{(0)}(\lambda) &= g_\nu(\lambda), \\
S^{(1)}(\lambda) &= g_\nu(\lambda)\left[\alpha - (1 + \lambda g_\nu(\lambda))\right], \\
S^{(2)}(\lambda) &= g_\nu(\lambda)\left[\alpha(1+\alpha) - (1+2\alpha)(1+\lambda g_\nu(\lambda)) + (1+\lambda g_\nu(\lambda))^2\right], \\
S^{(3)}(\lambda) &= g_\nu(\lambda)\left[(\alpha + 3\alpha^2 + \alpha^3) - (1 + 5\alpha + 3\alpha^2)(1 + \lambda g_\nu(\lambda))\right. \\
&\qquad \left. +(2 + 3\alpha)(1 + \lambda g_\nu(\lambda))^2 - (1 + \lambda g_\nu(\lambda))^3\right].
\end{cases} \tag{7.29}$$

*We define similarly for every integer $r, q \geq 0$:*

$$S_k^{(r,q)}(\lambda) \equiv \frac{1}{k}\mathrm{Tr}\left[\left(\Gamma_k^{(0)} - \lambda \mathrm{I}_k\right)^{-1}\left(\frac{W^\intercal W}{k}\right)^r\left(\Gamma_k^{(0)} - \lambda \mathrm{I}_k\right)^{-1}\left(\frac{W^\intercal W}{k}\right)^q\right].$$

Note that $S_k^{(r,q)} = S_k^{(q,r)}$ and that $S_k^{(r,0)}(\lambda) = \partial_z S_k^{(r)}(\lambda)$. For every $\lambda > \lambda_{\max}$, $S_k^{(1,1)}(\lambda)$ and $S_k^{(1,2)}(\lambda)$ converge almost surely (as $k \to \infty$) to well-defined limits, that satisfy the following equations:

$$S^{(1,1)}(\lambda) = g_\nu(\lambda) S^{(2)}(\lambda) - [1 + \lambda g_\nu(\lambda)] \partial_\lambda S^{(1)}(\lambda)$$

$$+ \alpha g_\nu(\lambda) \left[ g_\nu(\lambda) + S^{(1)}(\lambda) \right] \int \frac{\rho_\Delta(\mathrm{d}t) t}{(1 + t g_\nu(\lambda))^2} \left[ t \partial_\lambda S^{(1)}(\lambda) - g_\nu(\lambda) \right],$$

$$S^{(1,2)}(\lambda) = g_\nu(\lambda) S^{(3)}(\lambda) - [1 + \lambda g_\nu(\lambda)] \left[ S^{(1,1)}(\lambda) + (1 + \alpha) \partial_\lambda S^{(1)}(\lambda) \right]$$

$$+ \alpha g_\nu(\lambda) \left[ (1 + \alpha) g_\nu(\lambda) + S^{(1)}(\lambda) + S^{(2)}(\lambda) \right] \int \frac{\rho_\Delta(\mathrm{d}t) t}{(1 + t g_\nu(\lambda))^2} \left[ t \partial_\lambda S^{(1)}(\lambda) - g_\nu(\lambda) \right].$$

**Lemma 7.3.** *Let $\alpha, \Delta > 0$. We focus mainly on $S^{(2)}(\lambda)$. We have:*

(i) *For every $r$, $S^{(r)}(\lambda)$ is a strictly increasing function of $\lambda$, and $\lim_{\lambda \to \infty} S^{(r)}(\lambda) = 0$.*

(ii) *For every $\lambda > \lambda_{\max}$, $S^{(2)}(\lambda) = -\alpha\Delta$ if and only if $\Delta \leq \Delta_c(\alpha)$ and $\lambda = 1$.*

(iii) *For every $\Delta > \Delta_c(\alpha)$, $\lim_{\lambda \to \lambda_{\max}} S^{(2)}(\lambda) \in (-\alpha\Delta, 0)$ (it is well defined by monotonicity of $S^{(2)}(\lambda)$).*

Let us see how item $(ii)$ of Lemma 7.3 and eq. (7.28) end the proof of the eigenvalue transition. First, note that by the celebrated Weyl's interlacing inequalities [26], we have:

$$\liminf_{p \to \infty} \lambda_1 \geq \lambda_{\max},$$

$$\limsup_{p \to \infty} \lambda_2 \leq \lambda_{\max}.$$

This implies that because the perturbation of the matrix is of rank one, *at most one* outlier eigenvalue will exist in the limit $p \to \infty$. By eq. (7.28), this outlier $\lambda_1$ exists if and only if it satisfies, in the large $p \to \infty$ limit, the equation $S^{(2)}(\lambda_1) = -\alpha\Delta$. By item $(ii)$ of Lemma 7.3, this is the case only for $\lambda_1 = 1$ and $\Delta \leq \Delta_c(\alpha)$, which ends the proof. A completely rigorous treatement of these arguments requires to state more precisely concentration results. Such a treatment has been made in [20] in a very close case (from which all the arguments transpose), and we refer to it for more details. We finally describe the proofs of the lemmas in the following.

*Proof of Lemma. 7.2.* The essence of the computation originates from the derivation of Theorem 7.5 in [21]. Note that $S_k^{(0)}(\lambda)$ converges a.s. to the Stieltjes transform $g_\nu(\lambda)$ as $k \to \infty$ by Theorem 7.5. For every $1 \leq i \leq p$, $w_i$ denotes the $i$-th row of $W$. We denote $y = \frac{1}{\sqrt{\Delta p}}\xi - \frac{1}{\Delta}\mathrm{I}_p$. Since $W$ is independent of $y$, we can denote $y_1, \cdots, y_p$ the eigenvalues of $y$, and their empirical distribution converges a.s. to $\rho_\Delta$ as we know. We have in distribution:

$$\Gamma_k^{(0)} = \frac{1}{k} W^\mathsf{T} y W \stackrel{d}{=} \frac{\alpha}{p} \sum_{i=1}^{p} y_i \, w_i \, w_i^\mathsf{T}.$$

For every $i$, we denote:

$$\Gamma_{k,i}^{(0)} \equiv= \frac{\alpha}{p} \sum_{j(\neq i)}^{p} y_j \, w_j \, w_j^\mathsf{T}.$$

Note that $\Gamma_{k,i}^{(0)}$ is independent of $w_i$. We start from the (trivial) decomposition, for every $\lambda$:

$$-\frac{1}{\lambda} = \left( \Gamma_k^{(0)} - \lambda\mathrm{I}_k \right)^{-1} - \frac{1}{\lambda} \frac{W^\mathsf{T} y W}{k} \left( \Gamma_k^{(0)} - \lambda\mathrm{I}_k \right)^{-1}. \qquad (7.30)$$

We will make use of the Sherman-Morrison formula that gives the inverse of a matrix perturbed by a rank-one change:

$$(B + \tau\omega\omega^\mathsf{T})^{-1} = B^{-1} - \frac{1}{1 + \tau\omega^\mathsf{T} B^{-1}\omega} B^{-1}\omega\omega^\mathsf{T} B^{-1}, \qquad (7.31)$$

$$\omega^\mathsf{T} (B + \tau\omega\omega^\mathsf{T})^{-1} = \frac{1}{1 + \tau\omega^\mathsf{T} B^{-1}\omega}\omega^\mathsf{T} B^{-1}. \qquad (7.32)$$

Using it in eq. (7.30) yields:

$$-\frac{1}{\lambda} = \left(\Gamma_k^{(0)} - \lambda I_k\right)^{-1} - \frac{\alpha}{\lambda}\frac{1}{p}\sum_{i=1}^{p} y_i \frac{w_i}{1 + \frac{y_i}{k}w_i^\mathsf{T}(\Gamma_{k,i}^{(0)} - \lambda I_k)^{-1}w_i} w_i^\mathsf{T}\left(\Gamma_{k,i}^{(0)} - \lambda I_k\right)^{-1}. \quad (7.33)$$

Taking the trace of eq. (7.33), using the independence of $w_i$ and $\Gamma_{k,i}^{(0)}$, and the concentration $\frac{1}{k}w_i^\mathsf{T}Aw_i = \frac{1}{k}\mathrm{Tr}A$ with high probability for large $k$, we obtain the following equation:

$$-\frac{1}{\lambda} = g_\nu(\lambda) - g_\nu(\lambda)\frac{\alpha}{\lambda}\int \rho_\Delta(dt)\frac{t}{1 + tg_\nu(\lambda)}. \quad (7.34)$$

This is exactly the identity in Theorem 7.5 ! In the following, we will use very similar identities. A completely rigorous derivation of these would, however, require many technicalities to ensure in particular the concentration of all the involved quantities. It would exactly follow the proof of [21], and thus we do not repeat all the technicalities here. We can multiply eq. (7.33) by $\frac{W^\mathsf{T}W}{k}$, and take the trace:

$$-\frac{1}{\lambda}\frac{1}{k}\mathrm{Tr}\left[\frac{WW^\mathsf{T}}{k}\right] = S_k^{(1)}(\lambda) - \frac{\alpha}{\lambda}\frac{1}{p}\sum_i y_i \frac{\frac{w_i^\mathsf{T}}{\sqrt{k}}\left(\Gamma_{k,i}^{(0)} - \lambda I_k\right)^{-1}\left(\frac{1}{k}\sum_{j(\neq i)} w_j w_j^\mathsf{T} + \frac{1}{k}w_i w_i^\mathsf{T}\right)\frac{w_i}{\sqrt{k}}}{1 + \frac{y_i}{k}w_i^\mathsf{T}(\Gamma_{k,i}^{(0)} - \lambda I_k)^{-1}w_i}.$$

In the large $p, k$ limit, this implies that $S_k^{(1)}(\lambda)$ converges to a well-defined limit $S^{(1)}(\lambda)$, and this limit satisfies:

$$-\frac{\alpha}{\lambda} = S^{(1)}(\lambda) - \frac{\alpha}{\lambda}\left[\int \rho_\Delta(dt)\frac{t}{1 + tg_\nu(\lambda)}\right]\left(g_\nu(\lambda) + S^{(1)}(\lambda)\right)$$

Using finally eq. (7.34), it is equivalent to:

$$S^{(1)}(\lambda) = g_\nu(\lambda)\left[\alpha - (1 + \lambda g_\nu(\lambda))\right].$$

Multiplying eq. (7.33) by $\left(\frac{W^\mathsf{T}W}{k}\right)^2$ or $\left(\frac{W^\mathsf{T}W}{k}\right)^3$ yields, by the same analysis:

$$S^{(2)}(\lambda) = g_\nu(\lambda)\left[\alpha(1 + \alpha) - (1 + 2\alpha)(1 + \lambda g_\nu(\lambda)) + (1 + \lambda g_\nu(\lambda))^2\right],$$
$$S^{(3)}(\lambda) = g_\nu(\lambda)\left[(\alpha + 3\alpha^2 + \alpha^3) - (1 + 5\alpha + 3\alpha^2)(1 + \lambda g_\nu(\lambda))\right.$$
$$\left. + (2 + 3\alpha)(1 + \lambda g_\nu(\lambda))^2 - (1 + \lambda g_\nu(\lambda))^3\right].$$

The convergence of $S_k^{(1,1)}(\lambda)$ and $S_k^{(1,2)}(\lambda)$ follows from the same analysis, as well as the equations they satisfy. We detail the derivation of the equation on $S^{(1,1)}(\lambda)$ and leave the derivation of the second equation for the reader. We multiply eq. (7.33) by $\frac{W^\mathsf{T}W}{k}$. To simplify the calculations, we make use of concentrations, and denote $F_i \equiv \frac{W^\mathsf{T}W}{k} - \frac{1}{k}w_i w_i^\mathsf{T}$, which is independent of $w_i$. We obtain at leading order as $p \to \infty$:

$$-\frac{W^\mathsf{T}W}{k\lambda} = \left(\Gamma_k^{(0)} - \lambda I_k\right)^{-1}\frac{W^\mathsf{T}W}{k} - \frac{\alpha}{\lambda}\frac{1}{p}\sum_{i=1}^{p}\frac{y_i}{1 + y_i g_\nu(\lambda)}w_i w_i^\mathsf{T}\left(\Gamma_{k,i}^{(0)} - \lambda I_k\right)^{-1}F_i$$

$$-\frac{\alpha}{\lambda}\frac{1}{p}\sum_{i=1}^{p}\frac{y_i g_\nu(\lambda)}{1 + y_i g_\nu(\lambda)}w_i w_i^\mathsf{T}.$$

We multiply this equation by $(\Gamma_k^{(0)} - \lambda I_k)^{-1}$ and we use Sherman-Morrison formula eq. (7.31):

$$\left(\Gamma_k^{(0)} - \lambda I_k\right)^{-1} = \left(\Gamma_{k,i}^{(0)} - \lambda I_k\right)^{-1} - \left(\Gamma_{k,i}^{(0)} - \lambda I_k\right)^{-1}\frac{y_i w_i w_i^\mathsf{T}}{1 + y_i g_\nu(\lambda)}\left(\Gamma_{k,i}^{(0)} - \lambda I_k\right)^{-1}.$$

Using again the concentration of $\frac{1}{k}w^\mathsf{T} Aw$ on $\frac{1}{k}\mathrm{Tr}[A]$, this yields the cumbersome expression:

$$-\frac{W^\mathsf{T} W}{k\lambda}\left(\Gamma_k^{(0)} - \lambda\mathrm{I}_k\right)^{-1} = \left(\Gamma_k^{(0)} - \lambda\mathrm{I}_k\right)^{-1}\frac{W^\mathsf{T} W}{k}\left(\Gamma_k^{(0)} - \lambda\mathrm{I}_k\right)^{-1} \tag{7.35}$$

$$-\frac{\alpha}{\lambda}\frac{1}{p}\sum_{i=1}^p \frac{y_i}{1 + y_i g_\nu(\lambda)} w_i w_i^\mathsf{T}\left(\Gamma_{k,i}^{(0)} - \lambda\mathrm{I}_k\right)^{-1} F_i \left(\Gamma_{k,i}^{(0)} - \lambda\mathrm{I}_k\right)^{-1}$$

$$+\frac{\partial_\lambda S^{(1)}(\lambda)}{\lambda}\frac{\alpha}{p}\sum_{i=1}^p \frac{y_i^2}{(1 + y_i g_\nu(\lambda))^2} w_i w_i^\mathsf{T}\left(\Gamma_{k,i}^{(0)} - \lambda\mathrm{I}_k\right)^{-1}$$

$$-\frac{\alpha}{\lambda}\frac{1}{p}\sum_{i=1}^p \frac{y_i g_\nu(\lambda)}{(1 + y_i g_\nu(\lambda))^2} w_i w_i^\mathsf{T}\left(\Gamma_{k,i}^{(0)} - \lambda\mathrm{I}_k\right)^{-1}.$$

We finally multiply this equation by $\frac{W^\mathsf{T} W}{k}$ and take its trace. Using again the concentrations, we reach:

$$-\frac{S^{(2)}(\lambda)}{\lambda} = S^{(11)}(\lambda) - \frac{\alpha}{\lambda p}\sum_{i=1}^p \frac{y_i}{1 + y_i g_\nu(\lambda)}\left[S^{(11)}(\lambda) + \partial_\lambda S^{(1)}(\lambda)\right]$$

$$+\frac{\partial_\lambda S^{(1)}(\lambda)}{\lambda}\frac{\alpha}{p}\sum_{i=1}^p \frac{y_i^2}{(1 + y_i g_\nu(\lambda))^2}\left[g_\nu(\lambda) + S^{(1)}(\lambda)\right]$$

$$-\frac{\alpha}{\lambda}\frac{1}{p}\sum_{i=1}^p \frac{y_i g_\nu(\lambda)}{(1 + y_i g_\nu(\lambda))^2}\left[g_\nu(\lambda) + S^{(1)}(\lambda)\right].$$

We now take the limit $p \to \infty$ in the sum over $i$ and use Theorem 7.5 in the form:

$$\frac{\alpha}{\lambda}\int \rho_\Delta(\mathrm{d}t)\frac{t}{1 + tg_\nu(\lambda)} = 1 + \frac{1}{\lambda g_\nu(\lambda)}.$$

Inserting this into eq. (7.35) along with some trivial algebra yields:

$$S^{(1,1)}(\lambda) = g_\nu(\lambda)S^{(2)}(\lambda) - [1 + \lambda g_\nu(\lambda)]\,\partial_\lambda S^{(1)}(\lambda)$$

$$+\alpha g_\nu(\lambda)\left[g_\nu(\lambda) + S^{(1)}(\lambda)\right]\int \frac{\rho_\Delta(\mathrm{d}t)t}{(1 + tg_\nu(\lambda))^2}\left[t\,\partial_\lambda S^{(1)}(\lambda) - g_\nu(\lambda)\right],$$

which is what we aimed to show. Performing the same analysis for $S^{(1,2)}(\lambda)$ ends the proof. ∎

*Proof of Lemma 7.3.* Point $(i)$ is trivial by definition of $S_k^{(r)}(\lambda)$ and the almost sure convergence proven in Lemma 7.2. We turn to points $(ii)$ and $(iii)$. Let us denote the following function:

$$T^{(2)}(s) \equiv s\left[\alpha(1 + \alpha) - (1 + 2\alpha)\left(1 + sg_\nu^{-1}(s)\right) + \left(1 + sg_\nu^{-1}(s)\right)^2\right].$$

By Lemma 7.2, we have $T^{(2)}(s) = S^{(2)}(g_\nu^{-1}(s))$ so $T^{(2)}(s) < 0$ for $s \in (s_{\mathrm{edge}}, 0)$ by negativity of $S^{(2)}(\lambda)$ (as the trace of a negative matrix). Therefore, point $(ii)$ is equivalent to:

$$\forall s \in (s_{\mathrm{edge}}, 0), \quad T^{(2)}(s) = -\alpha\Delta \Leftrightarrow s = g_\nu(1) \text{ and } \Delta \leq \Delta_c(\alpha), \tag{7.36}$$

while point $(iii)$ means that for every $\Delta > \Delta_c(\alpha)$,

$$\forall s \in (s_{\mathrm{edge}}, 0), \quad T^{(2)}(s) > -\alpha\Delta. \tag{7.37}$$

The condition $s > s_{\mathrm{edge}}$ arises naturally as the counterpart of $z \geq \lambda_{\max}$. Recall that by Corollary 7.1, we have $\lambda_{\max} \leq 1$ for all $\Delta$. As $g_\nu^{-1}(s)$ is here completely explicit by eq. (7.20), and recalling the form of $\rho_\Delta$ in eq. (7.4), it is easy to show by an explicit computation the following identity:

$$\forall s \neq -1, \quad T^{(2)}(s) = -\alpha\Delta + \alpha\left[g_\nu^{-1}(s) - 1\right]\frac{s - \Delta - 2s\Delta + \sqrt{s^2 - 2s(1+s)\Delta + \Delta^2}}{2(1+s)},$$

$$T^{(2)}(-1) = \begin{cases} -\alpha(1+\alpha) & \text{if } \Delta \geq 1, \\ -\alpha\Delta(1 + \alpha\Delta) & \text{if } \Delta \leq 1. \end{cases}$$

It is then easy to see that the only possible solution to $T(s) = -\alpha\Delta$ with $s \in (s_{\text{edge}}, 0)$ is $s = g_\nu(1)$, if $g_\nu(1) \neq -1$. However, by Lemma 7.1, for any $\Delta > \Delta_c(\alpha)$ we have $s_{\text{edge}} < -1$. Moreover, in this case, one computes very easily (all expressions are explicit) $g_\nu^{-1}(-1) = 1$. Given the identity above, there is therefore no solution to $T^{(2)}(s) = -\alpha\Delta$ in $(s_{\text{edge}}, 0)$. By continuity of $T^{(2)}(s)$, and since $\lim_{s \to 0} T^{(2)}(s) = 0$, this implies $T^{(2)}(s) > -\alpha\Delta$ for $s \in (s_{\text{edge}}, 0)$, which proves point $(iii)$.

Assume now $\Delta \leq \Delta_c(\alpha)$. Note that the case $\Delta = \Delta_c(\alpha)$ is easy, as $s_{\text{edge}} = -1$ is the unique solution to $T^{(2)}(s) = -\alpha(1 + \alpha)$. For $\Delta < \Delta_c(\alpha)$, by Lemma 7.1 we obtain $-1 < s_{\text{edge}}$. In particular, $g_\nu(1) > s_{\text{edge}} > -1$, and we thus have that $s = g_\nu(1)$ is a solution (and the only one) to $T^{(2)}(s) = -\alpha\Delta$ by the identity shown above. This shows $(ii)$ and ends the proof of Lemma 7.3. ∎

### 7.5.2 Correlation of the leading eigenvector

We now turn to the study of the leading eigenvector. Let $\tilde{\mathbf{v}}$ be an eigenvector associated with the largest eigenvalue $\lambda_1$, normalized such that $\|\tilde{\mathbf{v}}\|^2 = p$. Then we have:

$$(\lambda_1 I_p - \Gamma_p^{(0)})\tilde{\mathbf{v}} = \frac{1}{\Delta}\frac{WW^\intercal}{k}\frac{\mathbf{v}^\intercal\tilde{\mathbf{v}}}{p}\mathbf{v}. \tag{7.38}$$

By normalization of $\tilde{\mathbf{v}}$, we obtain:

$$\tilde{\mathbf{v}} = \sqrt{p}\frac{\left(\lambda_1 I_p - \Gamma_p^{(0)}\right)^{-1}\frac{WW^\intercal}{k}\mathbf{v}}{\sqrt{\mathbf{v}^\intercal\frac{WW^\intercal}{k}\left(\lambda_1 I_p - \left(\Gamma_p^{(0)}\right)^\intercal\right)^{-1}\left(\lambda_1 I_p - \Gamma_p^{(0)}\right)^{-1}\frac{WW^\intercal}{k}\mathbf{v}}},$$

and therefore:

$$\frac{1}{p^2}\left|\tilde{\mathbf{v}}^T\mathbf{v}\right|^2 = \frac{1}{p}\frac{\left[\mathbf{v}^\intercal\left(\lambda_1 I_p - \Gamma_p^{(0)}\right)^{-1}\frac{WW^\intercal}{k}\mathbf{v}\right]^2}{\mathbf{v}^\intercal\frac{WW^\intercal}{k}\left(\lambda_1 I_p - \left(\Gamma_p^{(0)}\right)^\intercal\right)^{-1}\left(\lambda_1 I_p - \Gamma_p^{(0)}\right)^{-1}\frac{WW^\intercal}{k}\mathbf{v}}. \tag{7.39}$$

Using $\mathbf{v} = \frac{W}{\sqrt{k}}\mathbf{z}$ and the concentration of $\frac{1}{k}\mathbf{z}^\intercal A\mathbf{z}$ on $\frac{1}{k}\text{Tr } A$, we reach that as $p, k \to \infty$, we have:

$$\frac{1}{p^2}\left|\tilde{\mathbf{v}}^T\mathbf{v}\right|^2 \sim \frac{\left[\frac{1}{p}\text{Tr}\left\{\left(\lambda_1 I_p - \Gamma_p^{(0)}\right)^{-1}\left(\frac{WW^\intercal}{k}\right)^2\right\}\right]^2}{\frac{1}{p}\text{Tr}\left\{\left(\lambda_1 I_p - \left(\Gamma_p^{(0)}\right)^\intercal\right)^{-1}\left(\lambda_1 I_p - \Gamma_p^{(0)}\right)^{-1}\left(\frac{WW^\intercal}{k}\right)^3\right\}}. \tag{7.40}$$

The numerator is equal to $[\alpha^{-1}S_k^{(2)}(\lambda_1)]^2$, using the $S^{(r)}$ functions that we introduced in Lemma 7.2. Let us compute the denominator. Recall that we can write $\Gamma_p^{(0)} = WW^\intercal M/k$, with a symmetric matrix $M$ that is independent of $W$. For any $z$ large enough, we can expand:

$$\text{Tr}\left\{\left(zI_p - \left(\Gamma_p^{(0)}\right)^\intercal\right)^{-1}\left(zI_p - \Gamma_p^{(0)}\right)^{-1}\left(\frac{WW^\intercal}{k}\right)^3\right\},$$

$$= \sum_{a=0}^\infty\sum_{b=0}^\infty z^{-a-b-2}\text{Tr}\left\{\left(M\frac{WW^\intercal}{k}\right)^a\left(\frac{WW^\intercal}{k}M\right)^b\left(\frac{WW^\intercal}{k}\right)^3\right\},$$

$$\overset{(a)}{=} \sum_{a=0}^\infty\sum_{b=0}^\infty z^{-a-b-2}\text{Tr}\left\{\left(\frac{W^\intercal MW}{k}\right)^a\frac{W^\intercal W}{k}\left(\frac{W^\intercal MW}{k}\right)^b\left(\frac{W^\intercal W}{k}\right)^2\right\},$$

$$= \text{Tr}\left\{\left(zI_k - \Gamma_k^{(0)}\right)^{-1}\frac{W^\intercal W}{k}\left(zI_k - \Gamma_k^{(0)}\right)^{-1}\left(\frac{W^\intercal W}{k}\right)^2\right\},$$

$$= kS_k^{(1,2)}(z),$$

where in $(a)$ we used the cyclicity of the trace. Given Corollary 7.1, we know $\liminf_{p\to\infty} \lambda_1 \geq \lambda_{\max}$, so we can use the above calculation to write:

$$\epsilon(\Delta) = \lim_{\lambda\to\lambda_1} \lim_{k\to\infty} \frac{1}{\alpha} \frac{\left[S_k^{(2)}(\lambda)\right]^2}{S_k^{(1,2)}(\lambda)}. \tag{7.41}$$

As in the eigenvalue transition proof, to make this fully rigorous one would need to use more precisely the concentration results, and follow exactly the lines of [20]. We now use the transition of the leading eigenvalue (Corollary 7.1), that gives us the value of $\lambda_1$.

- For $\Delta < \Delta_c(\alpha)$, we know that $\lambda_1$ converges almost surely to 1. Consequently, we have in this case:

$$\epsilon(\Delta) = \frac{1}{\alpha} \frac{\left[S^{(2)}(1)\right]^2}{S^{(1,2)}(1)}.$$

By Lemma 7.3, we know that $S^{(2)}(1) = -\alpha\Delta$. Moreover, by Corollary 7.1 $\lambda_{\max} < 1$. This implies that $S^{(1,2)}(1) \in (0, +\infty)$. Indeed, 1 is out of the bulk of $\nu(\alpha, \Delta)$, so $g_\nu(1) \in (-\infty, 0)$ and by the relations shown in Lemma 7.2, all the transforms $S^{(r)}(1)$ and $S^{(r,q)}(1)$ will be finite. Note that $S^{(1,2)}(1) > 0$ by positivity of the matrices involved. This implies that for every $\Delta < \Delta_c(\alpha)$, $\epsilon(\Delta) > 0$.

- For $\Delta = \Delta_c(\alpha)$, we have $\lambda_{\max} = 1$ and $\lim_{\lambda\to 1} S^{(2)}(\lambda) = -\alpha\Delta$ as we have shown. For every $r, q$, let us define the functions $T^{(r)}$ and $T^{(r,q)}$ by $S^{(r)}(\lambda) = T^{(r)}[g_\nu(\lambda)]$ and $S^{(r,q)}(\lambda) = T^{(r,q)}[g_\nu(\lambda)]$. By Lemma 7.2 and the chain rule, we have:

$$\forall s \in (s_{\mathrm{edge}}, 0), \tag{7.42}$$

$$T^{(1,2)}(s) = sT^{(3)}(s) - \left[1 + sg_\nu^{-1}(s)\right] \left[T^{(1,1)}(s) + (1+\alpha)\frac{\partial_s T^{(1)}(s)}{\partial_s g_\nu^{-1}(s)}\right]$$

$$+ \alpha s \left[(1+\alpha)s + T^{(1)}(s) + T^{(2)}(s)\right] \int \frac{\rho_\Delta(\mathrm{d}t)t}{(1+ts)^2} \left[t\frac{\partial_s T^{(1)}(s)}{\partial_s g_\nu^{-1}(s)} - s\right].$$

Recall that $g_\nu^{-1}(s)$ is explicit by eq. (7.20) and $s_{\mathrm{edge}} = \lim_{\lambda\to\lambda_{\max}} g_\nu(\lambda)$. It moreover satisfies (cf Theorem 7.2) $\partial_s g_\nu^{-1}(s_{\mathrm{edge}}) = 0$. For $\Delta = \Delta_c(\alpha)$, by Lemma 7.1 we have $g_\nu(1) = -1 = s_{\mathrm{edge}}$. It is then only trivial algebra to verify from eq. (7.42) and the remaining relations of Lemma 7.2 that $T^{(1,2)}(-1) = +\infty$, which implies $\epsilon(\Delta_c(\alpha)) = 0$.

- We investigate here the $\Delta \to 0$ limit. In this limit, we know from eq. (7.41) and the analysis in the case $\Delta < \Delta_c(\alpha)$ above that

$$\lim_{\Delta\to 0} \epsilon(\Delta) = \lim_{\Delta\to 0} \frac{\alpha\Delta^2}{S^{(1,2)}(1)}.$$

It is again heavy but straightforward algebra to verify from eq. (7.42) and the remaining relations of Lemma 7.2 that as $\Delta \to 0$ and for any $s \in (s_{\mathrm{edge}}, 0)$:

$$T^{(1,2)}(s) = \alpha\Delta^2 + \mathcal{O}(\Delta^3).$$

This yields $\lim_{\Delta\to 0} \epsilon(\Delta) = 1$.

- Finally, we consider $\Delta > \Delta_c(\alpha)$. By eq. (7.41) and item $(iii)$ of Lemma 7.3, to obtain $\epsilon(\Delta) = 0$ we only need to prove that $\lim_{\lambda\to\lambda_{\max}} S^{(1,2)}(\lambda) = +\infty$. Equivalently, we must show $\lim_{s\to s_{\mathrm{edge}}} T^{(1,2)}(s) = +\infty$. Recall that $\partial_s g_\nu^{-1}(s_{\mathrm{edge}}) = 0$ and that since $s_{\mathrm{edge}}$ is finite, all $T^{(r)}(s_{\mathrm{edge}})$ for $r = 0, 1, 2, 3$ are finite as well by Lemma 7.2. It thus only remains to check that $\lim_{s\to s_{\mathrm{edge}}} T^{(1,2)}(s)\partial_s g_\nu^{-1}(s) > 0$. This would imply that $\lim_{s\to s_{\mathrm{edge}}} T^{(1,2)}(s) = +\infty$. We put this statement as a lemma, actually stronger than what we need:

**Lemma 7.4.** *For every $\alpha > 0$ and $\Delta > 1$, we have*

$$\liminf_{s\to s_{\mathrm{edge}}} T^{(1,2)}(s)\partial_s g_\nu^{-1}(s) > 0.$$

We prove this for every $\Delta > 1$, while only the case $\Delta > 1 + \alpha$ is needed in our analysis. As already argued, this lemma ends the proof.

*Proof of Lemma 7.4.* The idea is to lower bound $S^{(1,2)}(\lambda)$ by $\partial_\lambda g_\nu(\lambda)$, for every $\lambda > \lambda_{\max}$. We separate three cases:

- First, assume $\alpha > 1$. Then $W^\intercal W / k$ is full rank. In particular, by the classical results of [22], its lowest eigenvalue, denoted $\zeta_{\min}$ converges almost surely to $(1 - \alpha^{-1/2})^2$, the left edge of the Marchenko-Pastur distribution. Moreover, for any two symmetric positive square matrices $A$ and $B$, we know that $\mathrm{Tr}\,[AB] \geq 0$. Indeed, there exists a positive square root of $A$, and $\mathrm{Tr}\,[AB] = \mathrm{Tr}[A^{1/2} B A^{1/2}] \geq 0$. This implies immediately that if $a_0$ is the smallest eigenvalue of $A$, then $\mathrm{Tr}\,[AB] \geq a_0 \mathrm{Tr}\,[B]$, as $A - a_0 I$ is positive. We can use this to write, for any $\lambda > \lambda_{\max}$:

$$
\begin{aligned}
S_k^{(1,2)}(\lambda) &= \frac{1}{k} \mathrm{Tr} \left[ \left( \Gamma_k^{(0)} - \lambda \mathrm{I}_k \right)^{-1} \left( \frac{W^\intercal W}{k} \right) \left( \Gamma_k^{(0)} - \lambda \mathrm{I}_k \right)^{-1} \left( \frac{W^\intercal W}{k} \right)^2 \right], \\
&\geq \zeta_{\min}^2 \frac{1}{k} \mathrm{Tr} \left[ \left( \Gamma_k^{(0)} - \lambda \mathrm{I}_k \right)^{-1} \left( \frac{W^\intercal W}{k} \right) \left( \Gamma_k^{(0)} - \lambda \mathrm{I}_k \right)^{-1} \right], \\
&\geq \zeta_{\min}^3 \frac{1}{k} \mathrm{Tr} \left[ \left( \Gamma_k^{(0)} - \lambda \mathrm{I}_k \right)^{-2} \right].
\end{aligned}
$$

Taking the limit $k \to \infty$ in this last inequality, we obtain:

$$
S^{(1,2)}(\lambda) \geq \left( 1 - \alpha^{-1/2} \right)^6 \partial_\lambda g_\nu(\lambda). \tag{7.43}
$$

Taking the limit $\lambda \to \lambda_{\max}$ (or equivalently $s \to s_{\mathrm{edge}}$) yields

$$
\liminf_{s \to s_{\mathrm{edge}}} T^{(1,2)}(s) \partial_s g_\nu^{-1}(s) \geq \left( 1 - \alpha^{-1/2} \right)^6 > 0. \tag{7.44}
$$

- Now assume $\alpha < 1$. We do the same reasoning, as $W W^\intercal / k$ is now full rank, and it smallest eigenvalue, also denoted $\zeta_{\min}$, converges a.s. as $k \to \infty$ to $(1 - \sqrt{\alpha})^2$. We know (see the beginning of the current proof of the eigenvector correlation) that we can rewrite $S_k^{(1,2)}(\lambda)$ as the trace of a $p \times p$ matrix:

$$
\begin{aligned}
S_k^{(1,2)}(\lambda) &= \frac{1}{k} \mathrm{Tr} \left[ \left( \left( \Gamma_k^{(0)} \right)^\intercal - \lambda \mathrm{I}_k \right)^{-1} \left( \Gamma_k^{(0)} - \lambda \mathrm{I}_k \right)^{-1} \left( \frac{W W^\intercal}{k} \right)^3 \right], \\
&\geq \zeta_{\min}^3 \frac{1}{k} \mathrm{Tr} \left[ \left( \left( \Gamma_k^{(0)} \right)^\intercal - \lambda \mathrm{I}_k \right)^{-1} \left( \Gamma_k^{(0)} - \lambda \mathrm{I}_k \right)^{-1} \right], \\
&\geq \zeta_{\min}^3 \frac{1}{k} \mathrm{Tr} \left[ \left( \Gamma_k^{(0)} - \lambda \mathrm{I}_k \right)^{-2} \right],
\end{aligned}
$$

in which the last inequality comes from $\mathrm{Tr}\,[A A^\intercal] \geq \mathrm{Tr}\,[A^2]$ for any positive square matrix $A$. Once again, taking the limit $k \to \infty$, and then the limit $\lambda \to \lambda_{\max}$, this yields

$$
\liminf_{s \to s_{\mathrm{edge}}} T^{(1,2)}(s) \partial_s g_\nu^{-1}(s) \geq \left( 1 - \alpha^{1/2} \right)^6 > 0. \tag{7.45}
$$

- Finally, we treat the $\alpha = 1$ case. In this case, we can not use easy bounds as in the two previous cases as the support of the Marchenko-Pastur distribution touches 0. However, recall that everything is explicit here : $\rho_\Delta$ is given by eq. (7.4), $g_\nu^{-1}(s)$ is given by eq. (7.20) and Lemma 7.2 gives all the $T^{(r)}$ and $T^{(r,q)}$ in terms of $g_\nu^{-1}$ and $\rho_\Delta$. We can moreover use what we proved in Theorem 7.2:

$$
\partial_s g_\nu^{-1}(s_{\mathrm{edge}}) = \frac{1}{s^2} - \alpha \int \rho_\Delta(\mathrm{d}t) \frac{t^2}{(1 + t s_{\mathrm{edge}})^2} = 0.
$$

This can be used to simplify the term $\partial_s T^{(1)}(s)$ and the term $\int \rho_\Delta(dt) \frac{t^2}{(1+ts)^2}$. Some heavy but straightforward algebra yields from these relations that the following limit is finite, and is given by:

$$\lim_{s \to s_{\text{edge}}} T^{(1,2)}(s) \, \partial_s g_\nu^{-1}(s) = h(s_{\text{edge}}),$$

with

$$h(s) = \frac{h_1(s)^2 \times h_2(s)}{4s^6},$$
$$h_1(s) = -\Delta + \sqrt{\Delta^2 + s^2 - 2\Delta(2s+1)s} + s,$$
$$h_2(s) = 3\Delta - 3\sqrt{\Delta^2 + s^2 - 2\Delta(2s+1)s} + s(4s-3),$$

It is then very simple algebra (solving quadratic equations and using $\Delta > 1$) to see that there is no real negative solution to $h(s) = 0$, and that $h(s) > 0$ for all $s \in (-\infty, 0)$. This implies that $h(s_{\text{edge}}) > 0$, which ends the proof.

■

All together, this ends the proof of Theorem 7.3.

## 7.6 Proof of Theorem 7.4 and Corollary 7.2

### 7.6.1 Proof of Theorem 7.4

*Proof.* The proof is very similar to the proof of Theorem 7.2, and we will only point out the main differences. The proof of $(i)$ is exactly the same as the proof of the point $(i)$ of Theorem 7.2, once one notices that for $\Delta \leq \Delta_{\text{pos}}(\beta)$, the support of $\rho_{\beta,\Delta}$ is a subset of $\mathbb{R}_-$. We thus turn to the proof of $(ii)$. Again, the spectrum of $\Gamma_p^{uv}$, given by eq. (7.8) is, up to 0 eigenvalues, the same as the spectrum of $\Gamma_k^{uv}$, defined as follows:

$$\Gamma_k^{uv} \equiv \frac{1}{\Delta} \frac{1}{k} W^\intercal \left( \frac{1}{1+\Delta} \frac{y^\intercal y}{p} - \beta \, \mathrm{I}_p \right) W \in \mathbb{R}^{k \times k}. \tag{7.46}$$

We drop for simplicity the $uv$ exponents in these matrices. Once again, we can apply the Silverstein equation of Theorem 7.5 and the same arguments that we used in the proof of Theorem 7.2 completely transpose here. One notices that, by the classical Marchenko-Pastur results [22], the spectral distribution of $y^\intercal y/(p\Delta(1+\Delta)) - (\beta/\Delta)\, \mathrm{I}_p$ converges almost surely and in law to $\rho_{\beta,\Delta}$, before repeating the exact arguments of the proof of Theorem 7.2. This ends the proof of Thm. 7.4. ■

### 7.6.2 Proof of Corollary 7.2

*Proof.* Let $\alpha, \beta > 0$. We note:

- By Theorem 7.4, we know that if $\Delta = \Delta_{\text{pos}}(\beta)$, then $\lambda_{\max} \leq 0$.

- It is trivial by the form of $\Gamma_p$, see eq. (7.8), that as $\Delta \to +\infty$, $\lambda_{\max} \to 0$.

Let $z_{\text{edge}} = -\frac{1}{s_{\text{edge}}} + \alpha \int \rho_{\beta,\Delta}(dt) \frac{t}{1+s_{\text{edge}}t}$. Then we know that $\lambda_{\max} = z_{\text{edge}}$ if $\alpha \leq 1$ and $\lambda_{\max} = \max(0, z_{\text{edge}})$ if $\alpha > 1$. In particular, by the remark above, $z_{\text{edge}} \leq 0$ for $\Delta \leq \Delta_{\text{pos}}(\beta)$ and $z_{\text{edge}} \to 0^+$ as $\Delta \to \infty$. It is easy to see that $z_{\text{edge}}$ is a continuous and derivable function of $\Delta$, so that if we show the two following facts for any $\Delta \geq \Delta_{\text{pos}}(\beta)$:

$$\frac{dz_{\text{edge}}}{d\Delta} = 0 \Leftrightarrow \Delta = \Delta_c(\alpha, \beta) = \sqrt{\beta(1+\alpha)} \tag{7.47}$$
$$z_{\text{edge}}(\Delta_c(\alpha, \beta)) = 1, \tag{7.48}$$

this would end the proof as $z_{\text{edge}}$ would necessarily have a unique local maximum, located in $\Delta_c(\alpha, \beta)$, in which we have $\lambda_{\max} = 1$. We thus prove eq. (7.47) and eq. (7.48) in the following.

**Proof of eq. (7.47):** By the chain rule,

$$\frac{\mathrm{d}z_{\mathrm{edge}}}{\mathrm{d}\Delta} = \frac{\partial z_{\mathrm{edge}}}{\partial \Delta} + \frac{\partial s_{\mathrm{edge}}}{\partial \Delta}\frac{\partial z_{\mathrm{edge}}}{\partial s_{\mathrm{edge}}},$$

$$= \frac{\partial z_{\mathrm{edge}}}{\partial \Delta},$$

by the very definition of $s_{\mathrm{edge}}$, c.f. Theorem 7.4, since $z_{\mathrm{edge}} = g_\nu^{-1}(s_{\mathrm{edge}})$. Given the explicit form of $\rho_{\beta,\Delta}$, c.f. eq. (7.11), one can compute $z_{\mathrm{edge}}$ as a function of $s_{\mathrm{edge}}$. Its expression is cumbersome, but nevertheless explicit (we write $s$ instead of $s_{\mathrm{edge}}$ to avoid too heavy expressions):

$$z_{\mathrm{edge}} = \frac{-\alpha\Delta(\Delta+1) + \alpha\sqrt{\Delta^2(\Delta+1)^2 + s^2\left(\beta^2 - 2\beta\Delta(2\Delta+1) + \Delta^2\right) - 2\Delta(\Delta+1)s(\beta-\Delta)}}{2s^2(\beta s - \Delta)}$$

$$+ \frac{2(\alpha-1)\beta s^2 + \alpha s(\beta-\Delta) + 2\Delta s}{2s^2(\beta s - \Delta)}.$$

From this expression, it is simple analysis to verify that the only $s_{\mathrm{edge}} \in (-z_1(\beta,\Delta)^{-1}, 0)$ that satisfies $\frac{\partial z_{\mathrm{edge}}}{\partial \Delta} = 0$ is $s_{\mathrm{edge}} = -1$, and only if $\Delta > \sqrt{\beta}$. Recall that $s_{\mathrm{edge}}$ is defined as the solution to:

$$\alpha \int \rho_{\beta,\Delta}(\mathrm{d}t)\left(\frac{s_{\mathrm{edge}}t}{1 + s_{\mathrm{edge}}t}\right)^2 = 1.$$

Inserting $s_{\mathrm{edge}} = -1$ into this equation and using the explicit form of $\rho_{\beta,\Delta}$ of eq. (7.11) and that $\Delta > \sqrt{\beta}$, this reduces to:

$$\frac{\alpha\beta}{\Delta^2 - \beta} = 1,$$

which is equivalent to $\Delta = \Delta_c(\alpha,\beta) = \sqrt{\beta(1+\alpha)}$.

**Proof of eq. (7.48):** Given the computation above, we know that for $\Delta = \Delta_c(\alpha,\beta)$ we have $s_{\mathrm{edge}} = -1$. Given eq. (7.11), it is straightforward to compute:

$$z_{\mathrm{edge}}(\Delta_c(\alpha,\beta)) = -1 + \alpha \int \rho_{\Delta_c(\alpha,\beta)}(\mathrm{d}t)\frac{t}{1-t},$$

$$= 1.$$

$\blacksquare$

## 7.7 A note on non-linear activation functions

We consider here a non-linear activation function, in the spiked Wigner model or the spiked Wishart model. In these models, the spectral method with a non-linear activation function consists in taking the largest eigenvalue and the corresponding eigenvector of the matrix $\Gamma_p^{uu}$ (for the spiked Wigner model) or $\Gamma_p^{uv}$ (for the spiked Wishart model). These matrices are given by:

$$\Gamma_p^{uu} = \frac{1}{\Delta}\left((a-b)\mathrm{I}_p + b\frac{WW^\intercal}{k} + c\frac{\mathbb{1}_p\mathbb{1}_k^\intercal}{k}\frac{W^\intercal}{\sqrt{k}}\right) \times \left(\frac{Y}{\sqrt{p}} - a\mathbb{1}_M\right),$$

$$\Gamma_p^{uv} = \frac{1}{\Delta}\left((a-b)\mathrm{I}_p + b\frac{WW^\intercal}{k} + c\frac{\mathbb{1}_p\mathbb{1}_k^\intercal}{k}\frac{W^\intercal}{\sqrt{k}}\right) \times \left(\frac{1}{a+\frac{\Delta}{d}}\frac{Y^\intercal Y}{p} - d\beta\mathrm{I}_p\right)$$

In these equations, $a, b, c$ are coefficients that depend on the non-linearity. In the linear case, $c = 0$ and $a = b = 1$. Let us now assume for instance a non-linearity such that $a, b \neq 0$ and $c = 0$. Both $\Gamma_p^{uv}$ and $\Gamma_p^{uu}$ can be represented as

$$\Gamma_p = \left[(a-b)\mathrm{I}_p + b\frac{WW^\intercal}{k}\right]M, \tag{7.49}$$

in which $M$ is a symmetric (non necessarily positive or negative) matrix, independent of $W$. In order to perform the same analysis we made in the case of a linear activation function, we need in particular to be able to characterize the bulk of such matrices. Although this might be doable with more refined techniques, this does not seem to come as a direct consequence of the analysis of Silverstein and Bai [22, 25]. Indeed, one cannot write that the eigenvalues of $\Gamma_p$ are identical, up to $0$ eigenvalues, to the ones of a matrix of the type

$$\frac{1}{k} W^{\intercal} M' W,$$

which are the types of matrices covered by the analysis of Bai and Silverstein. Moreover, it is not immediate to use results of free probability [27] in this context. Indeed, $\Gamma_p$ in eq. (7.49) is the product of two matrices that are asymptotically free, but $M$ is not positive, which prevents a priori the use of the classical results on the $S$-transform of a product of two asymptotically free matrices. Writing $\Gamma_p$ as the sum of $(a - b)M$ and $b(WW^{\intercal})M/k$ does not yield any obvious results either, as these two matrices are not asymptotically free. For this reason, and although there might exist techniques to study the bulk of the matrix of eq. (7.49) and the transition in its largest eigenvalue, this is left for future work.

# 8 Phase diagrams of the Wishart model

Despite we illustrated the main part mostly with the Wigner model, in this section we present phase diagrams for the Wishart model. We show in particular a heat map of $\text{MMSE}_v$ as a function of the noise to signal ratio $\Delta/\rho_v^2$ for linear, sign and relu activation functions in Fig. 3. The white dashed line marks the critical threshold $\Delta_c$, given in the Wishart model by eq. (6.4), while the the dotted line shows the critical threshold of reconstruction for PCA.

Besides we show also the mean squared error as a function of the noise variance for larger values of $\alpha$ in Fig. 4. The $\text{MMSE}_v$ has been obtained solving the state evolution equations eq. (4.39), that show as well an unique stable fixed point for the large range of values that we studied, initializing with either informative or random conditions.

Figure 3: Spiked Wishart model: $\text{MMSE}_v$ on the spike as a function of noise to signal ratio $\Delta/\rho_v^2$, and generative prior (4) with compression ratio $\alpha$ for linear (left), sign (center), and relu (right) activations at $\beta = 1$. Dashed white lines mark the phase transitions $\Delta_c$, matched by both the AMP and LAMP algorithms. Dotted white line marks the phase transition of canonical PCA.

Figure 4: Spiked Wishart model: $\text{MMSE}_v$ as a function of noise $\Delta$ for a wide range of compression ratios $\alpha = 0, 1, 10, 100, 1000$, for linear (left), sign (center), and relu (right) activations, at $\beta = 1$.

## Footnotes

[1]Note that differently from the replica calculation in sec. 2, to write down the factor graph and derive the associated AMP algorithm we need to fix beforehand the structure of the prior distribution.

[2]These variables appear as well in the replica computation through Dirac delta Fourier representation.

[3]Note that while all the calculations are justified, refinements would be needed in order to be completely rigorous. These refinements would follow exactly some proofs of [21] and [20], so we will refer to them when necessary.

[4]The almost sure convergence could probably be extended to all $r \in \mathbb{N}^\star$ but we will only use these values of $r$ in the following.