[Reviews · NeurIPS 2019]

Reviewer 1



This paper investigates the matrix decomposition under the assumption that the spiked vector comes from a generated model. In particular, a single layer generating model with a linear/non-linear activation is considered. The authors study the phase transition on when the underlying spiked vector can be recovered, and shows that there is no algorithmic gap with generative-model priors, which is different from the sparse model. In addition, a new spectral method based on approximate messaging is proposed. The authors shows that this algorithm can reach the statistically optimal threshold. In general, this manuscript is well-written. The contribution is sufficiently significant. I recommend publication. On the other hand, I feel the 1-layer generative model may be overly simple. Some discussion on multi-layer networks as a prior is expected. Update after reading the rebuttal letter: The authors well addressed my question on how to generalize the model to the multi-layers case by providing additional experimental results. On the other hand, I do expect more interesting results on how this Bayes settings can give us more insight on practical problems that beyond information theoretical bound and algorithmic gap. My score is the same and I recommend publication.

Reviewer 2



The paper provides interesting and technically sound results that extend results for sparse models to ones where the spikes are generated with a (random) generative neural network: - The paper finds that contrary to sparse models, known algorithms obtain asymptotically optimal performance even if k is small. - The paper designs a spectral method, an approximate message passing algorithm, that reaches the statistical optimal threshold. - The paper provides information theoretic results on the estimation of the spikes, for a multilayer network, extending previous results for the single layer network. The paper writes in Section 3 that the AMP algorithm asymptotically reaches optimal performance, and that the analysis from the previous section on fixpoints applies. In the previous section the fix points are determined numerically in Figure 1. How can the paper conclude that the AMP reaches optimal performance in general, would that not only apply to the regime studied numerically? For the

Reviewer 3



This paper is well-written and well cited. In terms of techniques, the techniques used in this paper is mostly established tools developed in recent years. Though, for this complicated model, applying these established tools is already challenging. In terms of contribution, this model provides another example towards understanding the statistical-computational gap in teacher-student scenario: the behavior of this model is different from sparse PCA. Though, it is arguable that this different phenomenon is interesting enough. Overall, this is a good paper, so I vote for acceptance. -- The question was that, whether this result is interesting enough. The authors' reply sounds reasonable.

[Author Response · NeurIPS 2019]

We thank the referees for their interest in our paper and for their valuable comments that help us to make the paper clearer.

**Answer to referee 1:** We analyzed the multi-layer case beyond what is reported in the submitted paper. We have results for an arbitrary number of layers with sign, relu and linear activation functions. The main conclusions presented in the paper also apply to these cases. Notably (i) we did not observe any algorithmic gap, and (ii) the LAMP spectral method eq. (18) again reaches the same threshold as multi-layer AMP.

Equations to get the optimal error in the multi-layer case are in page 10-11 of the SM. Similarly to the discussion in Section 4 of the SM, the multi-layer AMP algorithm is obtained by combining the one in eq. (4.1) (for the low-rank layer) and the ML-AMP in the same 'plug and play' spirit discussed around eq. (4.4) of the SM. We also repeated the analysis of Section 6 of the SM for the multi-layer case, and obtained the corresponding threshold for an arbitrary number of layers and generic activation. For example, the threshold for a $L$-layer generative prior with sign activations is $\Delta_c = 1 + \sum_{l=1}^{L} \prod_{k=0}^{l-1} \frac{4}{\pi^2} \tilde{\alpha}_{L-k}$, where $\tilde{\alpha}_l = k_{l+1}/k_l$ is the aspect ratio of the weights matrix $W^{(l)}$ of layer $l$.

In the figure on the right we plot the recovery error as a function of the noise for a 3-layer prior with linear activations, and for a 2-layer prior with sign activations. We observe very much the same picture as in Fig. 3 in the main paper. We see that the Bayes optimal errors are continuous and hence do not present the algorithmic gap associated with a discontinuous phase transitions. We compare to the performance of the canonical PCA and the LAMP spectral method eq. (18) confirming (up to finite size effects) our theoretical finding that the LAMP spectral method achieves the optimal threshold. We will incorporate these results, plus a related discussion, into the final version of the paper.

Figure 1: Error as a function of noise. **a)** Three layers generative model with $(\tilde{\alpha}_1, \tilde{\alpha}_2, \tilde{\alpha}_3) = (1, 1, 1)$ using linear activations ($k_1 = 10^4$) **b)** Two layers generative model with $(\tilde{\alpha}_1, \tilde{\alpha}_2) = (1, 1)$ using sign activations ($k_1 = 2.10^4$). The vertical lines show the PCA and the optimal threshold respectively.

**Answer to referee 2:** Our claims of optimality of AMP are indeed limited to the cases investigated numerically. We will adjust the wording so that this is not misleading and extend the corresponding discussion. We do not claim AMP will reach optimal performance *in full generality*. One can engineer a situation, for instance with a very shifted relu on the last layer, and a very large intermediate layer, so that the spike **v** becomes effectively sparse with weakly correlated, almost independent, components, thus recovering the classical algorithmic gap. What is striking, however, is that the algorithmic gap disappears in all the first-to-come-in-mind cases that we have investigated. To clarify, the assumptions of this result are: the data was created using the spiked matrix model and the spike generated from a neural network with independent weight matrices and i.i.d. Gaussian entries. AMP optimality is achieved when the Bayes optimal error as a function of the noise is a continuous curve. This was the case in all the scenarios for which we solved the corresponding equations numerically. We will make a statement collecting all the assumptions in the final version.

We will work to improve readability of the final version. We consider that building on previous works (e.g. we use the strategy of [38], but the focus of that work is entirely different from the present one), putting the detailed (and lengthy) proofs in the appendix, and thus not being able to fit all the relevant material in the 8 pages, is standard for NeurIPS though.

**Answer to referee 3:** Incorporating the structure of the signals (both sparsity and generative modelling) allows to perform signal processing tasks more efficiently from the information theoretic point of view. The disappointment for sparse PCA (for $\Theta(1)$ sparsity) is that such improvement is, as far as we know, not algorithmically tractable, i.e. the naive PCA threshold is not improved when taking sparsity into account, and the computational-statistical gap exists. The fact that the gap disappears when sparsity is replaced by a generative model is important because it gives back the hope that the structure can be exploited not only information-theoretically but also tractably.

Whether the results of our paper translate to practical situations is currently under investigation. The improvement observed with LAMP over PCA on the fashion-MNIST is promising, and we hope to report soon even larger improvements for spiked matrix estimation using trained GAN priors as has been done in previous works, e.g. [5,8,9,10] for compressed sensing and denoising. We will add a related clarification into the final version.

[Meta-Review · NeurIPS 2019]

The paper provides an analysis of phase transition of spiked matrix models with generative-model priors, as opposed to the sparse prior, of previous work, and it looks at spectral and approximate message algorithm for this model. Well written. Several difference with sparse priors are noted, and it's an interesting contribution to statistical-computational gaps and student-teacher modeling.